
# EUREC$^4$A

Bjorn Stevens[1], Sandrine Bony[2], David Farrell[3], Felix Ament[4,1], Alan Blyth[5], Christopher Fairall[6], Johannes Karstensen[7], Patricia K Quinn[8], Sabrina Speich[9], Claudia Acquistapace[10], Franziska Aemisegger[11], Anna Lea Albright[2], Hugo Bellenger[2], Eberhard Bodenschatz[12], Kathy-Ann Caesar[3], Rebecca Chewitt-Lucas[3], Gijs de Boer[13,6], Julien Delanoë[14], Leif Denby[15], Florian Ewald[16], Benjamin Fildier[9], Marvin Forde[3], Geet George[1], Silke Gross[16], Martin Hagen[16], Andrea Hausold[16], Karen J. Heywood[17], Lutz Hirsch[1], Marek Jacob[10], Friedhelm Jansen[1], Stefan Kinne[1], Daniel Klocke[18], Tobias Kölling[19,1], Heike Konow[4], Marie Lothon[20], Wiebke Mohr[21], Ann Kristin Naumann[1,22], Louise Nuijens[23], Léa Olivier[24], Robert Pincus[13,6], Mira Pöhlker[25], Gilles Reverdin[24], Gregory Roberts[26,27], Sabrina Schnitt[10], Hauke Schulz[1], A. Pier Siebesma[23], Claudia Christine Stephan[1], Peter Sullivan[28], Ludovic Touzé-Peiffer[2], Jessica Vial[2], Raphaela Vogel[2], Paquita Zuidema[29], Nicola Alexander[3], Lyndon Alves[30], Sophian Arixi[26], Hamish Asmath[31], Gholamhossein Bagheri[12], Katharina Baier[1], Adriana Bailey[28], Dariusz Baranowski[32], Alexandre Baron[33], Sébastien Barrau[26], Paul A. Barrett[34], Frédéric Batier[35], Andreas Behrendt[36], Arne Bendinger[7], Florent Beucher[26], Sebastien Bigorre[37], Edmund Blades[38], Peter Blossey[39], Olivier Bock[40], Steven Böing[15], Pierre Bosser[41], Denis Bourras[42], Pascale Bouruet-Aubertot[24], Keith Bower[43], Pierre Branellec[44], Hubert Branger[45], Michal Brennek[46], Alan Brewer[47], Pierre-Etienne Brilouet[20], Björn Brügmann[1], Stefan A. Buehler[4], Elmo Burke[48], Ralph Burton[5], Radiance Calmer[13], Jean-Christophe Canonici[49], Xavier Carton[50], Gregory Jr. Cato[51], Jude Andre Charles[52], Patrick Chazette[33], Yanxu Chen[9], Michal T. Chilinski[46], Thomas Choularton[43], Patrick Chuang[53], Shamal Clarke[54], Hugh Coe[43], Céline Cornet[55], Pierre Coutris[56], Fleur Couvreux[26], Susanne Crewell[10], Timothy Cronin[57], Zhiqiang Cui[15], Yannis Cuypers[24], Alton Daley[3], Gillian M Damerell[17], Thibaut Dauhut[1], Hartwig Deneke[58], Jean-Philippe Desbios[49], Steffen Dörner[25], Sebastian Donner[25], Vincent Douet[59], Kyla Drushka[60], Marina Dütsch[61,62], André Ehrlich[63], Kerry Emanuel[57], Alexandros Emmanouilidis[63], Jean-Claude Etienne[26], Sheryl Etienne-Leblanc[64], Ghislain Faure[26], Graham Feingold[47], Luca Ferrero[65], Andreas Fix[16], Cyrille Flamant[66], Piotr Jacek Flatau[27], Gregory R. Foltz[67], Linda Forster[19], Iulian Furtuna[68], Alan Gadian[15], Joseph Galewsky[69], Martin Gallagher[43], Peter Gallimore[43], Cassandra Gaston[29], Chelle Gentemann[70], Nicolas Geyskens[71], Andreas Giez[16], John Gollop[72], Isabelle Gouirand[73], Christophe Gourbeyre[56], Dörte de Graaf[1], Geiske E. de Groot[23], Robert Grosz[46], Johannes Güttler[12], Manuel Gutleben[16], Kashawn Hall[3], George Harris[74], Kevin C. Helfer[23], Dean Henze[75], Calvert Herbert[74], Bruna Holanda[25], Antonio Ibanez-Landeta[12], Janet Intrieri[76], Suneil Iyer[60], Fabrice Julien[26], Heike Kalesse[63], Jan Kazil[13,47], Alexander Kellman[72], Abiel T. Kidane[21], Ulrike Kirchner[1], Marcus Klingebiel[1], Mareike Körner[7], Leslie Ann Kremper[25], Jan Kretzschmar[63], Ovid Krüger[25], Wojciech Kumala[46], Armin Kurz[16], Pierre L'Hégaret[77], Matthieu Labaste[24], Tom Lachlan-Cope[78], Arlene Laing[79], Peter Landschützer[1], Theresa Lang[22,1], Diego Lange[36], Ingo Lange[4], Clément Laplace[80], Gauke Lavik[21], Rémi Laxenaire[81], Caroline Le Bihan[44], Mason Leandro[53], Nathalie Lefevre[24], Marius Lena[68], Donald Lenschow[28], Qiang Li[16], Gary Lloyd[43], Sebastian Los[69], Niccolò Losi[82], Oscar Lovell[83], Christopher Luneau[84], Przemyslaw Makuch[85], Szymon Malinowski[46], Gaston Manta[9], Eleni Marinou[16,86], Nicholas Marsden[43], Sebastien Masson[24], Nicolas Maury[26], Bernhard Mayer[19], Margarette Mayers-Als[3], Christophe Mazel[87], Wayne McGeary[88,3], James C. McWilliams[89], Mario Mech[10], Melina Mehlmann[7], Agostino Niyonkuru Meroni[90], Theresa Mieslinger[4,1], Andreas Minikin[16], Peter Minnett[29], Gregor Möller[19], Yanmichel Morfa Avalos[1], Caroline Muller[9], Ionela Musat[2], Anna Napoli[90], Almuth Neuberger[1], Christophe Noisel[24], David Noone[91], Freja Nordsiek[12], Jakub L. Nowak[46], Lothar Oswald[16], Douglas J Parker[15], Carolyn Peck[92], Renaud Person[24,93], Miriam Philippi[21], Albert Plueddemann[37], Christopher Pöhlker[25], Veronika Pörtge[19], Ulrich Pöschl[25], Lawrence Pologne[3], Michał Posyniak[32], Marc Prange[4], Estefanía Quiñones Meléndez[75], Jule Radtke[22,1], Karim Ramage[59], Jens Reimann[16], Lionel Renault[94,89], Klaus Reus[7], Ashford Reyes[3], Joachim Ribbe[95], Maximilian Ringel[1], Markus Ritschel[1], Cesar B Rocha[96], Nicolas Rochetin[9], Johannes Röttenbacher[63], Callum Rollo[17], Haley Royer[29], Pauline Sadoulet[26], Leo Saffin[15], Sanola Sandiford[3], Irina Sandu[97], Michael Schäfer[63], Vera Schemann[10], Imke Schirmacher[4], Oliver Schlenczek[12], Jerome Schmidt[98], Marcel Schröder[12], Alfons Schwarzenboeck[56], Andrea Sealy[3], Christoph J Senff[13,47], Ilya Serikov[1], Samkeyat Shohan[63], Elizabeth Siddle[17], Alexander Smirnov[99], Florian Späth[36], Branden Spooner[3], M. Katharina Stolla[1], Wojciech Szkółka[32], Simon P. de Szoeke[75], Stéphane Tarot[44], Eleni Tetoni[16], Elizabeth Thompson[6], Jim Thomson[60], Lorenzo Tomassini[34], Julien Totems[33], Alma





Anna Ubele[25], Leonie Villiger[11], Jan von Arx[21], Thomas Wagner[25], Andi Walther[100], Ben Webber[17], Manfred Wendisch[63], Shanice Whitehall[3], Anton Wiltshire[83], Allison A. Wing[101], Martin Wirth[16], Jonathan Wiskandt[7], Kevin Wolf[63], Ludwig Worbes[1], Ethan Wright[81], Volker Wulfmeyer[36], Shanea Young[102], Chidong Zhang[8], Dongxiao Zhang[103,8], Florian Ziemen[104], Tobias Zinner[19], and Martin Zöger[16]

[1]Max Planck Institute for Meteorology, Hamburg, Germany
[2]LMD/IPSL, Sorbonne Université, CNRS, Paris, France
[3]Caribbean Institute for Meteorology and Hydrology, Barbados
[4]Universität Hamburg, Hamburg, Germany
[5]National Centre for Atmospheric Science, University of Leeds, UK
[6]NOAA Physical Sciences Laboratory, Boulder, CO, USA
[7]GEOMAR Helmholtz Centre for Ocean Research Kiel, Kiel, Germany
[8]NOAA PMEL, Seattle, WA, USA
[9]LMD/IPSL, École Normale Supérieure, CNRS, Paris, France
[10]Institute for Geophysics and Meteorology, University of Cologne, Cologne, Germany
[11]Institute for Atmospheric and Climate Science, ETH Zurich, Zurich, Switzerland
[12]Max Planck Institute for Dynamics and Self-Organization, Göttingen, Germany
[13]Cooperative Institute for Research In Environmental Sciences, University of Colorado, Boulder, CO, USA
[14]LATMOS/IPSL, Université Paris-Saclay, UVSQ, Guyancourt, France
[15]University of Leeds, Leeds, UK
[16]Deutsches Zentrum für Luft- und Raumfahrt, Oberpfaffenhofen, Germany
[17]Centre for Ocean and Atmospheric Sciences, School of Environmental Sciences, University of East Anglia, Norwich, UK
[18]DWD Hans-Ertel-Zentrum für Wetterforschung, Offenbach, Germany
[19]Ludwig-Maximilians-Universität in Munich, Munich, Germany
[20]Laboratoire d'Aérologie, University of Toulouse, CNRS, Toulouse, France
[21]Max Planck Institute for Marine Microbiology, Bremen, Germany
[22]Meteorological Institute, Center for Earth System Research and Sustainability, Universität Hamburg, Hamburg, Germany
[23]Delft University of Technology, Delft, The Netherlands
[24]Sorbonne Université, CNRS, IRD, MNHN, UMR7159 LOCEAN/IPSL, Paris, France
[25]Max Planck Institute for Chemistry, Mainz, Germany
[26]CNRM, University of Toulouse, Météo-France, CNRS, Toulouse, France
[27]Scripps Institution of Oceanography, University of California San Diego, San Diego, CA, USA
[28]National Center for Atmospheric Research, Boulder, CO, USA
[29]University of Miami, Miami, FL, USA
[30]Hydrometeorological Service, Guyana
[31]Institute of Marine Affairs, Trinidad and Tobago
[32]Institute of Geophysics, Polish Academy of Sciences, Warsaw, Poland
[33]LSCE/IPSL, CNRS-CEA-UVSQ, University Paris-Saclay, Gif sur Yvette, France
[34]Met Office, Exeter, UK
[35]Frédéric Batier Photography, Berlin, Germany
[36]Institute of Physics and Meteorology, University of Hohenheim, Stuttgart, Germany
[37]Woods Hole Oceanographic Institution, Woods Hole, MA, USA
[38]Queen Elizabeth Hospital, Barbados
[39]Department of Atmospheric Sciences, University of Washington, Seattle, WA, USA
[40]IPGP, Paris, France
[41]ENSTA Bretagne, Lab-STICC, CNRS, Brest, France
[42]Aix Marseille Université, Université de Toulon, CNRS, IRD, MIO UM 110, Marseille, France
[43]University of Manchester, Manchester, UK
[44]Ifremer, Brest, France
[45]Irphe, CNRS/Amu/Ecm, Marseille, France
[46]University of Warsaw, Warsaw, Poland
[47]NOAA Chemical Sciences Laboratory, Boulder, CO, USA
[48]St. Christopher Air & Sea Ports Authority, Basseterre, St. Kitts and Nevis





[49]SAFIRE, Météo-France, CNRS, CNES, Cugnaux, France

[50]LOPS/IUEM, Université de Bretagne Occidentale, CNRS, Brest, France

[51]Argyle Meteorological Services, St. Vincent & The Grenadines

[52]Grenada Meteorological Services, Grenada

[53]University of California Santa Cruz, Santa Cruz, CA, USA

[54]Cayman Islands National Weather Service, Cayman Islands

[55]LOA, Univ. Lille, CNRS, Lille, France

[56]LAMP, Université Clermont Auvergne, CNRS, Clermont-Ferrand, France

[57]Massachusetts Institute of Technology, Cambridge, MA, USA

[58]Leibniz Institute for Tropospheric Research, Leipzig, Germany

[59]IPSL, CNRS, Paris, France

[60]Applied Physics Laboratory, University of Washington, Seattle, WA, USA

[61]Department of Earth and Space Sciences, University of Washington, Seattle, WA, USA

[62]Department of Meteorology and Geophysics, University of Vienna, Vienna, Austria

[63]Leipzig Institute for Meteorology, University of Leipzig, Germany

[64]Meteorological Department St. Maarten

[65]Gemma Center, University of Milano-Bicocca, Milan, Italy

[66]LATMOS/IPSL, Sorbonne Université, CNRS, Paris, France

[67]NOAA Atlantic Oceanographic and Meteorological Laboratory, Miami, FL, USA

[68]Compania Fortuna, Sucy en Brie, France

[69]Department of Earth and Planetary Sciences, University of New Mexico, Albuquerque, NM USA

[70]Farallon Institute, USA

[71]DT-INSU, CNRS, France

[72]Barbados Coast Guard, Barbados

[73]The University of the West Indies, Cave Hill Campus Barbados, Barbados

[74]Regional Security System, Barbados

[75]College of Earth, Ocean and Atmospheric Sciences, Oregon State University, Corvallis, OR, USA

[76]NOAA Earth System Research Laboratory, Boulder, CO, USA

[77]LOPS, Université de Bretagne Occidentale, Brest, France

[78]British Antarctic Survey, Cambridge, UK

[79]Caribbean Meteorological Organization, Trinidad and Tobago

[80]IPSL, Paris, France

[81]Center for Ocean-Atmospheric Prediction Studies, Florida State University, Tallahassee, FL, USA

[82]Milano Bicocca University, Italy

[83]Trinidad and Tobago Meteorological Services, Trinidad and Tobago

[84]OSU Pytheas, Marseille, France

[85]The Institute of Oceanology, Polish Academy of Sciences, Sopot, Poland

[86]National Observatory of Athens, Athens, Greece

[87]Dronexsolution, Toulouse, France

[88]Barbados Meteorological Services, Barbados

[89]Department of Atmospheric and Oceanic Sciences, UCLA, Los Angeles, CA, USA

[90]CIMA Research Foundation, Savona, Italy

[91]University of Auckland, Auckland, NZ

[92]Meteorological Service, Kingston, Jamaica

[93]Sorbonne Université, CNRS, IRD, MNHN, INRAE, ENS, UMS 3455, OSU Ecce Terra, Paris, France

[94]LEGOS, University of Toulouse, IRD, CNRS, CNES, UPS, Toulouse, France

[95]University of Southern Queensland, Toowoomba, Australia

[96]University of Connecticut Avery Point, Groton, CT, USA

[97]European Centre for Medium Range Weather Forecasts, Reading, UK

[98]Naval Research Laboratory, Monterey, CA, USA

[99]Science Systems and Applications, Inc., Lanham, Maryland, USA

[100]University of Wisconsin-Madison, Madison, WI, USA

[101]Department of Earth, Ocean and Atmospheric Science, Florida State University, Tallahassee, FL, USA



[102]National Meteorological Service of Belize, Belize
[103]Cooperative Institute for Climate, Ocean, and Ecosystem Studies, University of Washington, Seattle, WA, USA
[104]Deutsches Klimarechenzentrum GmbH, Hamburg, Germany

**Correspondence:** Bjorn Stevens (bjorn.stevens@mpimet.mpg.de), Sandrine Bony (sandrine.bony@lmd.ipsl.fr)

**Abstract.** The science guiding the EUREC[4]A campaign and its measurements are presented. EUREC[4]A comprised roughly five weeks of measurements in the downstream winter trades of the North Atlantic — eastward and south-eastward of Barbados. Through its ability to characterize processes operating across a wide range of scales, EUREC[4]A marked a turning point in our ability to observationally study factors influencing clouds in the trades, how they will respond to warming, and their

link to other components of the earth system, such as upper-ocean processes or, or the life-cycle of particulate matter. This characterization was made possible by thousands (2500) of sondes distributed to measure circulations on meso (200 km) and larger (500 km) scales, roughly four hundred hours of flight time by four heavily instrumented research aircraft, four global-ocean class research vessels, an advanced ground-based cloud observatory, a flotilla of autonomous or tethered measurement devices operating in the upper ocean (nearly 10000 profiles), lower atmosphere (continuous profiling), and along the air-sea

interface, a network of water stable isotopologue measurements, complemented by special programmes of satellite remote sensing and modeling with a new generation of weather/climate models. In addition to providing an outline of the novel measurements and their composition into a unified and coordinated campaign, the six distinct scientific facets that EUREC[4]A explored — from Brazil Ring Current Eddies to turbulence induced clustering of cloud droplets and its influence on warm-rain formation — are presented along with an overview EUREC[4]A's outreach activities, environmental impact, and guidelines for

scientific practice.

## 1  Introduction

The clouds of the trades are curious creatures. On the one hand fleeting, sensitive to subtle shifts in the wind, to the presence and nature of particulate matter, to small changes in radiant energy transfer, surface temperatures or myriad other factors as they scud along the sky (Siebesma et al., 2020). On the other hand, from the view of the climate and often in our mind's eye,

immutable and substantial (Stevens and Schwartz, 2012) – like Magritte's suspended stone. In terms of climate change, should even a small part of their sensible side express itself with warming, large effects could result. This realization has motivated a great deal of research in recent years (Bony et al., 2015), culminating in a recent field study named EUREC[4]A (*ElUcidating the RolE of Cloud-Circulation Coupling in ClimAte*). EUREC[4]A's measurements, which this paper describes, express the most ambitious effort ever to quantify how cloud-properties co-vary with their atmospheric and oceanic environment across the

enormous (mm to Mm) range of relevant scales.

Initially EUREC[4]A was proposed as a way to test hypothesized cloud-feedback mechanisms thought to explain large differences in model estimates of climate sensitivity, and to provide benchmark measurements for a new generation of models and satellite observations (Bony et al., 2017). To meet these objectives required quantifying different measures of clouds in the trade winds as a function of their large-scale environment. In the past, efforts to use measurements for this purpose – from





BOMEX[1] (Holland and Rasmusson, 1973) to ASTEX (Albrecht et al., 1995) to RICO (Rauber et al., 2007), see also Bannon
(1949) – have been hampered by an inability to constrain the mean vertical motion over larger-scales, and by difficulties in
quantifying something as multifaceted as a field of clouds (Bretherton et al., 1999; Stevens et al., 2001; Siebesma et al., 2003;
vanZanten et al., 2011). EUREC⁴A was made possible by emergence of new methods to measure these quantities, many of
which were developed through experimentation over the past decade in and around the Barbados Cloud Observatory (Stevens

et al., 2016, 2019a). To execute these measurements required a high-flying aircraft (HALO) to characterize the clouds and
cloud environment from above, both with remote sensing and through the distribution of a large number of dropsondes around
the perimeter of a mesoscale (ca 200 km diameter) circle. A second low-flying aircraft (the ATR), with in situ cloud sensors and
sidewards staring active remote sensing, was necessary to ground truth the remote sensing from above, as well as to determine
the distribution of cloudiness and aspects of the environment that could not be measured from above. By making these mea-

surements upwind of the Barbados Cloud Observatory, and adding a research vessel (the R/V Meteor) for additional surface
based remote sensing and surface flux measurements, the environment and its clouds would be yet better constrained.

Quantifying day-to-day variations in both cloudiness and its environment, opened the door to additional questions, greatly
expanding EUREC⁴A's scope. In addition to testing hypothesized cloud feedback mechanisms, EUREC⁴A's experimental plan
was augmented to (i) quantify the relative role of micro and macrophysical factors in rain formation; (ii) quantify different

factors influencing the mass, energy and momentum balances in the sub-cloud layer; (iii) identify processes influencing the
evolution of ocean meso-scale eddies; (iv) measure the influence of ocean heterogeneity, i.e., fronts and eddies, on air-sea
interaction and cloud formation; and (v) provide benchmark measurements for a new generation of both fine-scale coupled
models and satellite retrievals. Complementing these scientific pursuits, EUREC⁴A developed outreach and capacity building
activities that allowed scientists coming from outside the Caribbean to benefit from local expertise and vice versa.

Addressing these additional questions required a substantial expansion of the activities initially planned by the Barbadian-
French-German partnership that first proposed EUREC⁴A. This was accomplished through a union of projects led by additional
investigators. For instance, EUREC⁴A-UK (a UK project), brought a Twin Otter (TO for short) and ground based facilities for
aerosol measurements to advance cloud physics studies; EUREC⁴A-OA secured the service of two additional research vessels
(the R/V L'Atalante and the R/V Maria Sybilla Merian) and various autonomous observing platforms to study ocean processes;

and the Atlantic Tradewind Ocean–Atmosphere Mesoscale Interaction Campaign (ATOMIC) brought an additional research
vessel (the R/V Ronald H. Brown), assorted autonomous systems, and the WP-3D Orion, Miss Piggy, to help augment stud-
ies of air-sea and aerosol-cloud interactions. Additionally nationally funded projects funded a large-scale sounding array, the
installation of a scanning precipitation radar, the deployment of ship-borne kite-stablized helium balloons (CloudKites), a net-
work of water stable isotopologue measurements, as well as a rich assortment of uncrewed aerial and seagoing systems, among

them fixed-wing aircraft, quad copters, drifters, buoys, gliders, and saildrones. Support within the region helped link activities
to operational initiatives, such as a training programme for forecasters from the region, and fund scientific participation from
around the Caribbean. The additional measurement platforms considerably increased the scope of EUREC⁴A, whose opera-

---

[1]Acronyms for field experiments, many instrument, instrument platforms, and institutions, often take the form of a proper name, which if not expanded
in the text are provided in the cited literature or in B describing the instrumentation.



**Figure 1.** The EUREC⁴A study area in the lower trades of the North Atlantic. The zonally oriented band following the directions of the trades between the Northwest Tropical Atlantic Station (NTAS) and the Barbados Cloud Observatory (BCO) is called Tradewind Alley. It encompasses two study areas (A and B). The EUREC⁴A-Circle is defined by the circular airborne sounding array centered at 57.7°W. A third study area (C) followed the southeast to northwest meanders of what we called the Boulevard des Tourbillons. The background shows a negative of the cloud field taken from the 5 February, 2020 MODIS-Terra (ca 1430 UTC) overpass.





tions were coordinated over a large area (roughly $10° \times 10°$, as shown in Fig. 1) within the lower trades near Barbados, making it possible to pursue the additional objectives outlined above and described in more depth below.

This article describes EUREC⁴A in terms of seven different facets as outlined above. To give structure to such a vast undertaking we focus on EUREC⁴A's novel aspects, but strive to describe these in a way that also informs and guides the use of EUREC⁴A data by those who did not have the good fortune to share in its collection. The presentation (§3) of these seven facets is framed by an overview of the general setting of the campaign in § 2, and a discussion of more peripheral, but still important, aspects such as data access and the ecological impacts of our activities in § 4.

## 2    General setting and novel measurements

EUREC⁴A deployed a wide diversity of measurement platforms over two theatres of action. These, the 'Tradewind Alley' and the 'Boulevard des Tourbillons', are illustrated schematically in Fig. 1. Tradewind Alley comprised an extended corridor with its downwind terminus defined by the BCO and extending upwind to the Northwest Tropical Atlantic Station (NTAS $51°$W, $15°$N), an advanced open ocean mooring Weller (2018); Bigorre and Plueddemann (2020) that has been operated continuously since 2001. Measurements aimed at addressing the initial objectives of EUREC⁴A were situated near the western end of the corridor, within the range of low-level scans of the C-band radar on Barbados. The area of overlap between the radar and the ($\sim 200$ km diameter) EUREC⁴A-Circle (marked A in Fig. 1) defined a region of intensive measurements in support of studies of cloud-circulation interactions, cloud physics, and factors influencing the mesoscale patterning of clouds. Additional measurements between the NTAS and $55°$W (Region B in Fig. 1) supported studies of air-sea interaction and provided complementary measurements of the upwind environment, including a characterization of its clouds and aerosols.

The Boulevard des Tourbillons describes the geographic region that hosted intensive measurements to study how air-sea interaction is influenced by mesoscale eddies, sub-mesoscale fronts, and filaments in the ocean (Region C in Fig. 1). Large (ca 300 km) warm eddies – which migrate Northwestward and often envelope Barbados, advecting large fresh-water filaments stripped from the shore of South America – created a laboratory well suited to this purpose. These eddies, known as North Brazil Current (NBC) Rings, form when the retroflecting NBC pinches off around $7°$N. Characterizing these eddies further offered the possibility to expand the upper-air network of radiosondes, and to make contrasting cloud measurements in a potentially different large-scale environment. This situation led EUREC⁴A to develop its measurements following the path of the NBC rings toward Barbados from their place of formation near the point of the NBC retroflection, with a center of action near Region C in Fig. 1. Measurements in the Boulevard des Tourbillons extended the upper-air measurement network, and provided cloud measurements to contrast with similar measurements being made in Tradewind Alley.

### 2.1    Platforms for measuring the lower atmosphere

Aerial measurements were made by research aircraft, uncrewed (i.e., remotely piloted) aerial systems (UASs), and from balloon or parachute borne soundings. These were mostly distributed along Tradewind Alley. Fig. 2 shows the realization of the EUREC⁴A strategy in the form of repeated Box-L flight pattern flown by the ATR (orange) within the EUREC⁴A-Circle

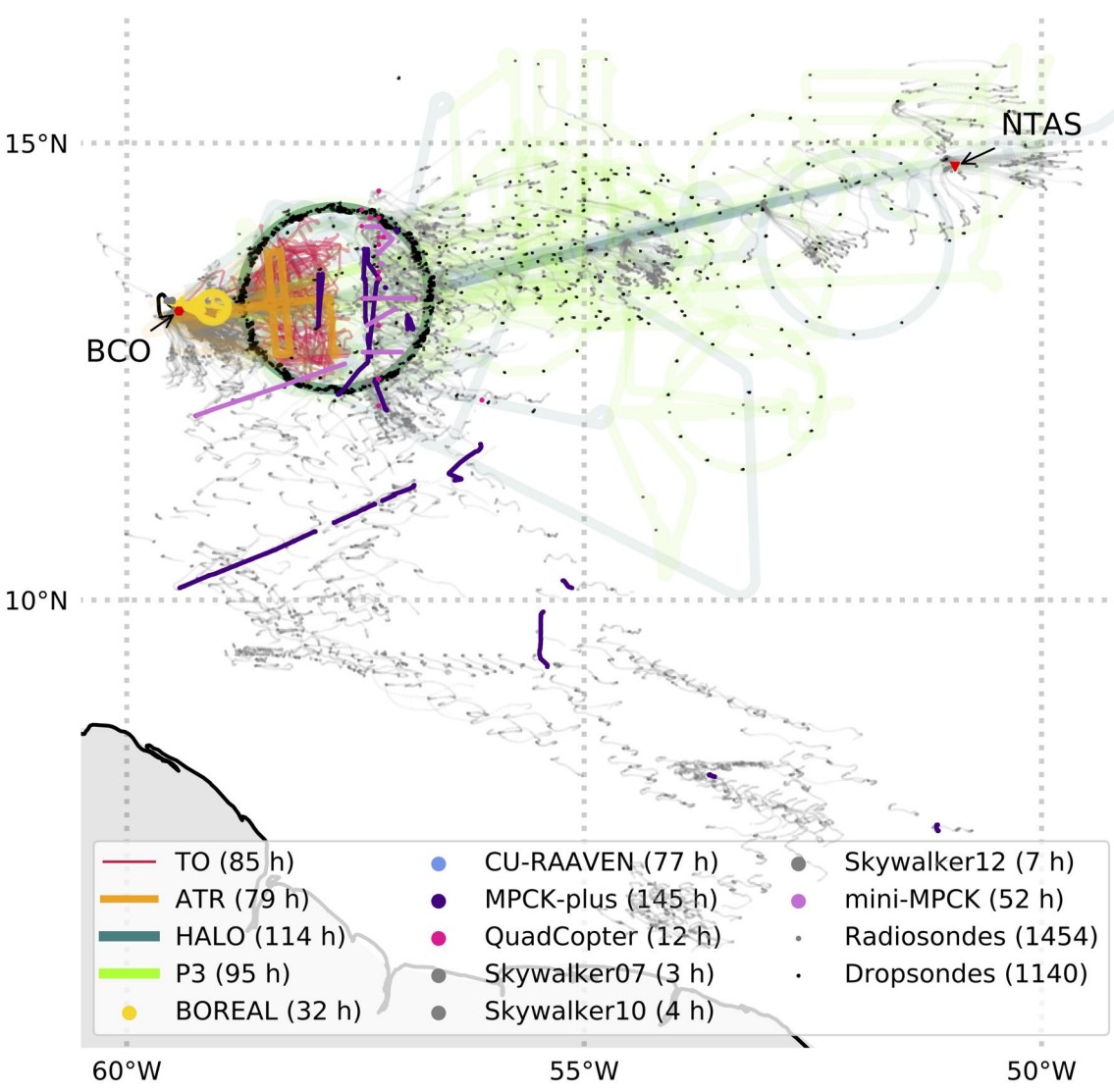

**Figure 2.** Domain of aerial measurements. Flight time for the crewed aircraft is for the period spent east of 59°W and west of 45°W. Radiosonde ascents and descents are counted separately but only when valid measurements are reported. Sonde trajectories are shown by trail of points, which are more pronounced due to their larger horizontal displacement.

(mostly flown by HALO; teal, with black points for dropsondes). Excursions by HALO and flights by the P-3 extended the

area of measurements upwind of the EUREC$^4$A-Circle toward the NTAS. The TO intensively sampled clouds in the area of

ATR operations in the western half of the EUREC$^4$A-Circle. UASs provided extensive measurements of the lower atmosphere,

and because of their more limited range and need to avoid air-space conflicts with other platforms, concentrated in the area

between the EUREC$^4$A-Circle and Barbados.

Different clusters of radiosonde soundings (evident as short traces of gray points) can also be discerned in Fig. 2. Those

soundings originating from the BCO (342) and from the R/V Meteor (362) were launched from relatively fixed positions, with

the R/V Meteor operating between 12.5°N and 14.5°N along the 57.25°W meridian. East of the EUREC$^4$A-Circle, sondes

were launched by the R/V Ronald H. Brown (Ron Brown), which mostly measured air-masses in coordination with the P-3

measurements between the NTAS and the EUREC$^4$A-Circle. The R/V Maria Sybilla Merian (MS-Merian) and R/V L'Atalante

(Atalante) combined to launch 424 sondes in total, as they worked water masses up and down the Boulevard. For most sondes,

measurements were recorded for both the ascent and descent, with descending sondes falling by parachute for all platforms

except the R/V Ron Brown. The synoptic environment encountered during EUREC$^4$A, the radiosonde measurement strategy,

and an analysis of the sonde data are described in more detail by Stephan et al. (2020).

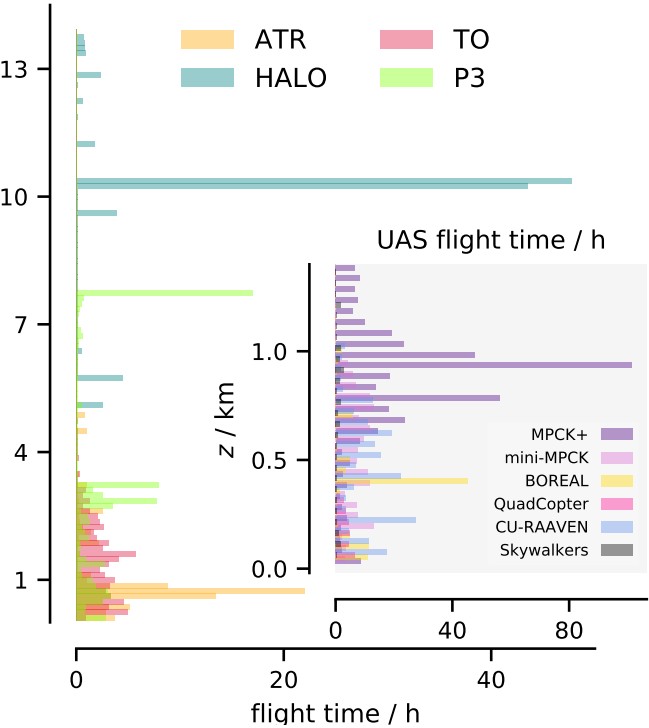

**Figure 3.** Flight time spent at different altitudes by different airborne platforms





HALO, the ATR and most of the UASs emphasized statistical sampling. Hence flight plans did not target specific conditions,
although the ATR flight levels were adjusted slightly based on the estimated height of the boundary layer and cloud field that
was encountered – but this varied relatively little. Measurements from the MPCK+ (a large CloudKite tethered to the R/V
MS-Merian) emphasized the lower cloud layer selecting conditions when clouds seemed favorable. The mini-MPCK was used
more for profiling the boundary layer and the cloud-base region, and was deployed when conditions allowed. The Twin Otter
targeted cloud fields, often flying repeated samples through cloud clusters identified visibly, but also sampled the sub-cloud
layer. The P-3 strategy was more mixed; some flights targeted specific conditions and others were more statistically oriented;
for example, to fill gaps in the HALO and ATR sampling strategy. The different sampling strategies are reflected in Fig. 3
where the measurements of HALO are sharply concentrated at about $10.5\,\mathrm{km}$ and those of the ATR at about $800\,\mathrm{m}$. Fig. 3 also
shows the strong emphasis on sampling the lower atmosphere, with relatively uniform coverage of the lower $3\,\mathrm{km}$. Except for
the Twin-Otter, which was limited to daytime operations, take-off and landing times of the aircraft were staggered, with three
night flights by the P3, to better sample the diurnal cycle. Data papers for the individual platforms are being prepared and will
describe their activities in greater detail.

HALO performed several satellite underpasses as part of planned 'excursions' from its circling flight pattern. These included
one underpass of MISR on 5 February 2020, and another under the core satellite of the Global Precipitation Mission (GPM)
on 11 February, 2020.

## 2.2 Airborne platforms for measuring the upper ocean, and air-sea interface

Four global class research vessels – all equipped with surface meteorological measurements and underway temperature/salinity
sampling devices – and scores of autonomous surface and sub-surface vehicles were deployed along Tradewind Alley and the
Boulevard des Tourbillons. The tracks of the surface vessels are shown in Fig. 4. These tracks, colored by measurements of the
sea-surface temperature, show slightly more variability in water temperatures along the Boulevard des Tourbillons, in contrast
with more steady westward warming of surface temperatures following the trades along Tradewind Alley. The more dynamic
situation along the 'Boulevard', as compared to the situation on the 'Alley', required a different measurement strategy. For the
former, research vessels actively tracked and surveyed mesoscale features, for the latter the sampling was more statistical so as
to better support the airborne measurements and cloud characterization.

Along Tradewind Alley, the R/V Meteor mostly worked along the line of longitude at 57.25°W between 12.4°N and 14.2°N.
The R/V Ron Brown, coordinating its measurements with the P3, was stationed between the NTAS and the Meridional Over-
turning Variability Experiment ($50\,\mathrm{nm}$ northwest of the NTAS, not shown in Fig. 1) moorings in January, and in the region
upwind of the EUREC[4]A-Circle, near 55°W, in February. For both positions, SWIFT buoys were deployed and recovered in
coordination with P-3 Airborne Expendable Bathythermograph (AXBT) soundings. A Saildrone, two Wave Gliders, an Au-
toNaut (Caravela), four underwater gliders, and extensive Conductivity-Temperature-Depth (CTD) casts from the two ships
profiled the upper ocean Fig. 5.

Along the Boulevard des Tourbillons the R/V MS-Merian and the R/V Atalante studied the meso- and submesocale dy-
namics. Both research vessels extensively profiled the ocean's upper km using a wide assortment of instruments, including

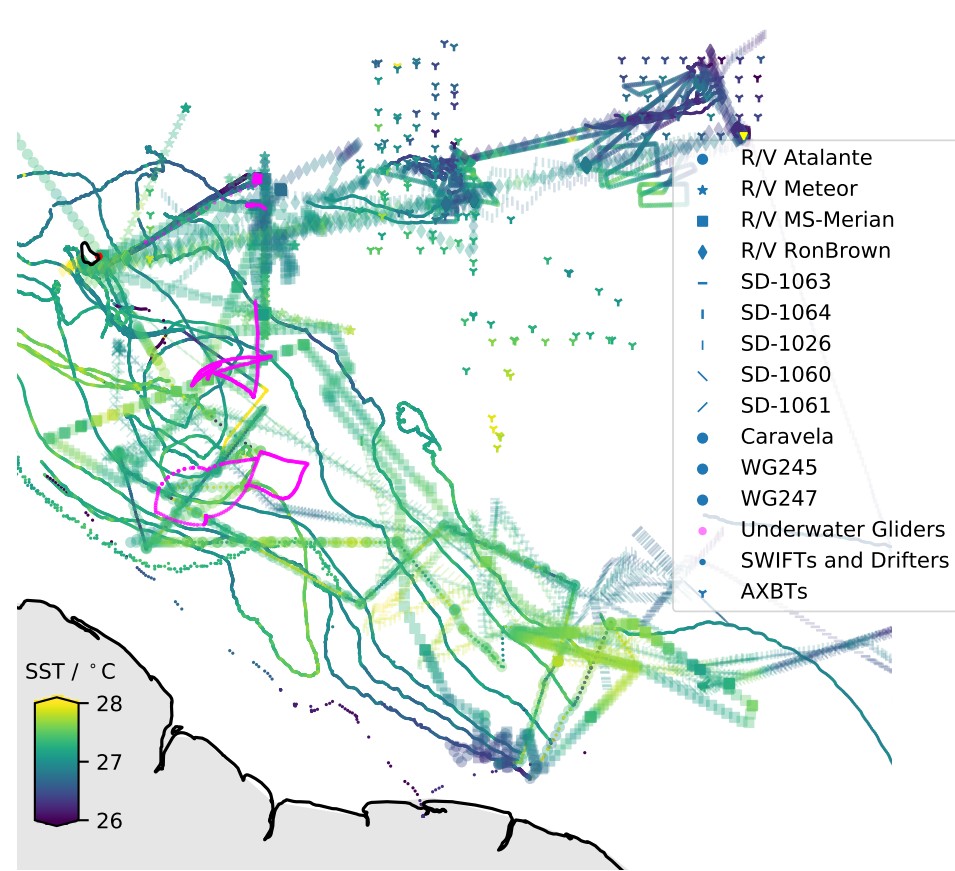

**Figure 4.** Map showing location of measurements by surface and sub-surface platforms

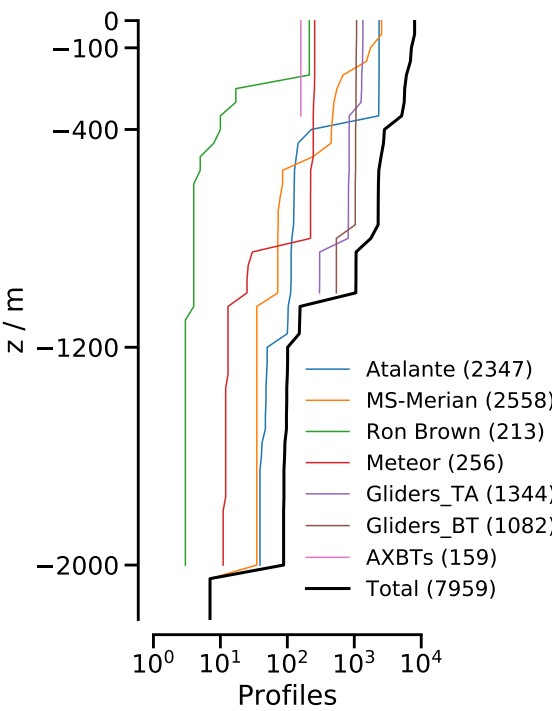

**Figure 5.** Number of profiles to sample a given depth. Ship-based profiling is from CTD casts, underway CTDs, XBTs, and moving vessel profilers. AXBTs were dropped by the P3.

underway CTDs, Moving Vessel Profilers, vertical microstructure profilers (VMP and MSS), Expendable Bathythermographs (XBTs) and Expendable CTDs (XCTDs). Three ocean gliders (one SeaExplorer, two Slocum electric gliders) provided dense

sampling (more than 1300 profiles, most to at least $700\,\mathrm{m}$, Fig. 5) of subsurface structures associated with mesoscale eddies. Of the roughly eight thousand upper ocean profiles performed during EUREC[4]A, nearly three fourths were performed in coordination with the eddy sampling along the 'Boulevard'. Four Saildrones, 22 drifters and four deployments of two air-sea fluxes observing prototypes, OCARINA and PICCOLO substantially expanded the observations at the ocean-atmosphere interface. Five Argo floats equipped with a dissolved oxygen sensor were deployed to allow a Lagrangian monitoring of the ocean surface

and subsurface dynamics during and after the campaign.

    To effectively survey features in the active waters of the Boulevard des Tourbillons the sampling strategy and cruise plan were assessed daily, using information from the past day's measurements, updates from satellite products, weather forecasts, and ocean predictions. Tailored satellite products and model predictions were provided by a variety of groups[2] to help track and follow surface features in near real time.

---

[2]Collecte Localisation Satellite, the Centre Aval de Traitement des Données, Mercator Ocean, and the Center for Ocean-Atmospheric Prediction Studies





## 2.3 Instrument clusters

EUREC[4]A set itself apart from past field studies both through new types of measurements, as performed by individual platforms, but also through the quantity or clustering of certain instruments. Instrument clustering means using similar instruments across a number of platforms so as to improve the statistical characterization of air-masses, and their evolution. The ability to make such measurements, enables estimates of systematic and random measurement errors, giving rise to a different quality of measurement as compared to those made previously, especially in marine environments. Examples are described below and include the use of remote sensing, instruments for measuring stable water isotopologues, and drones. A platform-by-platform listing of the EUREC[4]A instrumentation is provided in B.

### 2.3.1 Remote Sensing

EUREC[4]A included eight cloud-sensitive Doppler (W and Ka band) radars. Four zenith staring instruments were installed at surface sites (BCO, R/V MS-Merian, R/V Meteor, and R/V Ron Brown) and three on aircraft (nadir, zenith on the ATR, HALO and the P3). The ATR flew a second, horizontally staring, Doppler system. Two scanning radars (a C-band system installed on Barbados, and a P-3 X-band tail radar), and three profiling rain radars (one at the BCO, another at the Caribbean Institute for Meteorology and Hydrology (CIMH) and a third on the R/V MS-Merian), measured precipitation. The R/V MS-Merian additionally had an X-Band radar installed for wave characteristics and surface currents over a roughly 2 km footprint around the ship. Fourteen lidars were operated, four of which were advanced (high-spectral resolution, multi-wavelength) Raman or DIAL (Differential Absorption Lidar) systems for profiling water vapor and aerosol/cloud properties. The Raman systems (at the BCO, on the R/V MS-Merian and R/V Meteor) were upward staring surface mounted systems, the DIAL operated in a downward looking model from HALO (Wirth et al., 2009). On the ATR a backscatter UV lidar operated alongside the horizontally staring radar, looking horizontally to provide an innovative plan-form view of cloudiness near cloud base. In total, six wind-lidars and three ceilometers were operated from the BCO and all Research Vessels except for the R/V Atalante. As an example of the synergy the combination of these sensors provides, Fig. 6 shows water vapor flux profiles (Behrendt et al., 2020) estimated from co-located vertically staring Doppler wind-lidar and Raman (water-vapor) lidar measurements from the ARTHUS system (Lange et al., 2019) aboard the R/V MS-Merian. This type of measurement strategy, employing a dense network of remote sensors to both improve sampling and realize synergies, is increasingly emphasized for land-atmosphere interaction studies (e.g., Wulfmeyer et al., 2018), but it is more difficult to realize, and thus uncommon, over the ocean.

More standard, but still unprecedented by virtue of its space-time-frequency coverage, was the contribution of airborne, surface and space-based passive remote sensing to EUREC[4]A. Three 14-channel microwave radiometers operated from surface platforms, and a 25 channel downward staring system operated from HALO (Mech et al., 2014; Schnitt et al., 2017). Handheld sun-photometer measurements were made on all four research vessels and an automated system operated from Ragged Point, near the BCO, provided additional constraints on estimates of aerosol loading (from lidars) and column water vapor (from radiometers). Infrared radiometers for measuring the surface skin temperature were operated on the ATR, HALO, the R/V Ron Brown, the Boreal and CU-RAAVEN UAVs, and on the five Saildrones. For estimating fluxes of radiant energy, broadband

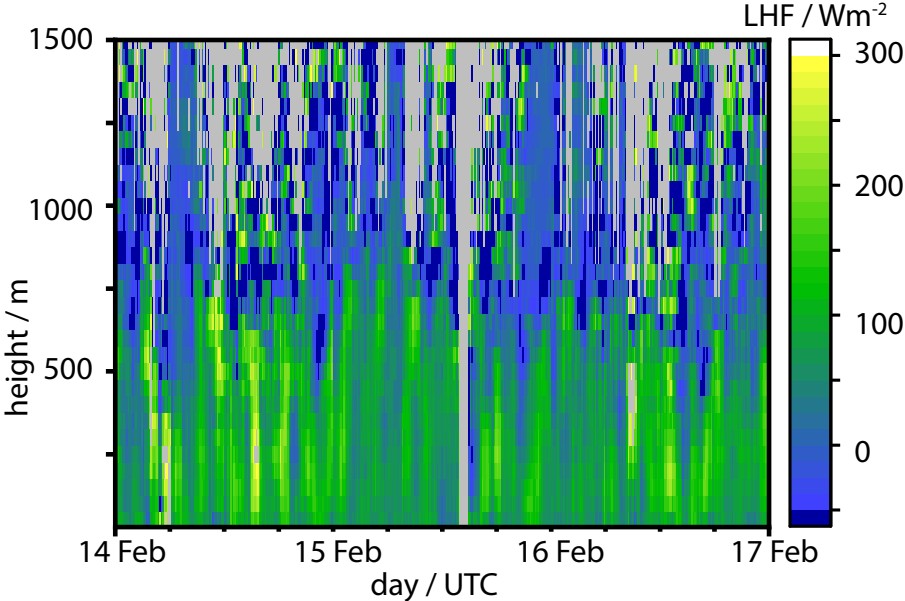

**Figure 6.** Synergy showing ship-based remotely sensed latent heat (enthalpy of vaporization) flux profiles (LHF) from the combination of water-vapor Raman lidar (ARTHUS) and Doppler wind lidar aboard the R/V MS-Merian. The mean value over the three day period is $100\,\mathrm{W\,m^{-2}}$ at $200\,\mathrm{m}$ and the fluxes are positive throughout the sub-cloud layer.

longwave and shortwave radiometers were installed on three of the airborne (zenith and nadir) and surface (zenith) platforms. In addition, HALO and the R/V Meteor hosted high-spectral resolution systems measuring shortwave and near-infrared down-
190 and up-welling radiances (Wendisch et al., 2001). Near-real-time geostationary GOES-East satellite imagery and cloud product retrievals between 19°N-5°S, 49°W-66°W were collected, with finer temporal resolution of every minute (between 14 January and 14 February, with a few data gaps due to the need to support hazardous weather forecasting in other domains) archived over most of this domain. ASTER's high-resolution ($15\,\mathrm{m}$ visible and near infrared, and $90\,\mathrm{m}$ thermal) imager on board of TERRA was activated between 7°N-17°N and 41°W-61°N. It recorded 412 images of $60\,\mathrm{km}\times60\,\mathrm{km}$ in 25 overpasses between
195 11 January and 15 February. These images are complemented by Sentinel-2 data with images at $10\,\mathrm{m}$ resolution in some visible-near-infrared bands and $20\,\mathrm{m}$ resolution in shortwave-infrared bands relevant for cloud microphysical retrievals.

The intensity of remote sensing instrumentation in the vicinity of the EUREC[4]A-Circle will support efforts to, for the first time, observationally close the column energy budget over the ocean; and, to test hypotheses that link precipitation to processes across very different time and space scales.

### 2.3.2 Stable water isotopologues

EUREC[4]A benefited from an unusually complete and spatially extensive network of stable water isotopologue measurements ($H_2^{18}O$, $H_2^{16}O$, and HDO) distributed across multiple platforms. Seven laser spectrometers and five precipitation sampling sys-





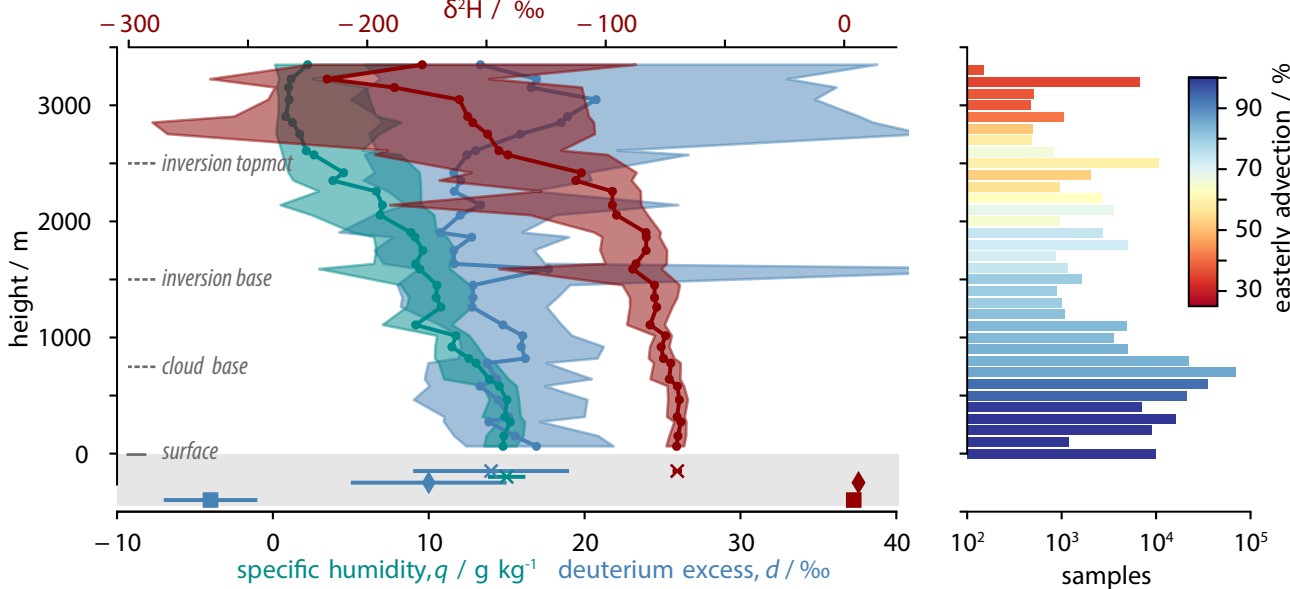

**Figure 7.** Water stable-isotopologues measurements during EUREC[4]A. This shows the sampling from all stable-isotopologue measurements, including samples from near surface waters. Shown on the left is the mean vertical profile of the measured isotopologues, on the right is the frequency of Easterlies.

tems especially designed to avoid post-sampling re-evaporation were deployed. At the BCO, two laser spectrometers provided robust high-frequency measurements of isotopologues in water vapour and 46 event-based precipitation samples were col-

205 lected. Three ships – the R/V Atalante, the R/V Meteor, and the R/V Ron Brown – were similarly equipped, and in addition collected ocean water samples (340 in total) from the underway water line and the CTDs. These samples have been analysed in the laboratory together with fifty ship-board rainfall samples. Two of the high-frequency laser spectrometers were mounted on the ATR and P-3 to measure the vertical distribution of water isotopologues. The airborne measurements also added continuity, sampling air-masses between the BCO and R/V Meteor stations and between the R/V Meteor and the upwind R/V Ron Brown.

The measurements provided very good coverage through the depth of the lower ($3\,\mathrm{km}$) atmosphere. Air-parcel backward trajectories based on three-dimensional wind fields from the operational ECMWF analyses indicate that boundary layer air came almost exclusively from the East, with a more heterogeneous isotropic origin of air-masses sampled above 2500m (Fig. 7, see also Aemisegger et al., 2020). Large-scale context for the in-situ measurements will be provided by retrievals of atmospheric HDO and $H_2^{16}O$ from space-borne instruments.

The size of the network of isotopologue measurements and the degree of coordination among the different measurement sites will enable investigations of the variability of the stable water isotopologues – in space and time, in ocean water, atmospheric vapor, and precipitation following the trades – that were previously not possible.

### 2.3.3 Drones and tethered platforms

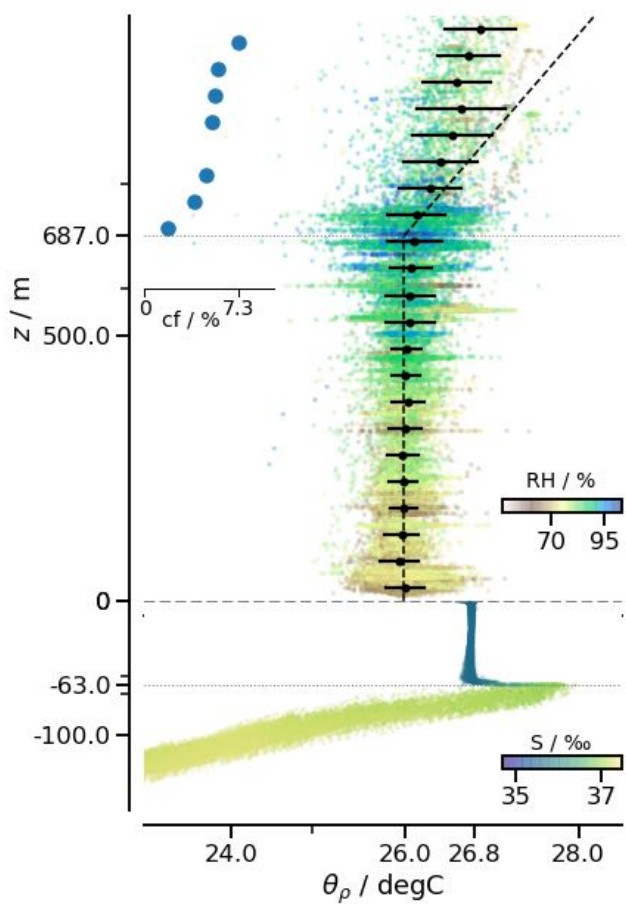

**Figure 8.** Typical ocean and atmospheric boundary layer upwind of the BCO. Points show CU-RAAVEN measurements of the density potential temperature above the surface, and underwater glider measurements of the temperature below the surface normalized to compensate for differences associated with either synoptic variations or from variations in the depth of the sampled boundary layers. Blue dots show profile of cloud fraction from all MPCK profiles. The black dashed line show the isentropic lapse rate of moist air with the measured near surface properties; the slope discontinuity at the lifting condensation level marks the shift from an unsaturated to a saturated isentrope. The temperature difference between the sea-surface and the lower atmosphere is taken from saildrone data.

A diversity of tethered and remotely piloted platforms provided measurements in the lower atmosphere and upper ocean.

Many of these had been used in past field studies, but what set EUREC[4]A apart was its coordinated use of so many platforms. Five fixed wing systems and a quad-copter provided approximately 200 h of open ocean atmospheric profiling, while seven seagliders profiled the underlying ocean porpoising well over a thousand times, mostly between the surface and 700 m. Fig. 8





presents measurements from one of the seagliders, and the CU-RAAVEN which along with the other fixed-wing systems (Boreal and Skywalkers) was flown from Morgan Lewis, a windward beach about 20 km north of the BCO. The measurements

highlight the boundary-layers on either side of the air-sea interface, one (in the atmosphere) extending to about 700 m, and capped by a layer that is stably stratified with respect to unsaturated, but unstable with respect to saturated convection. The typical ocean mixed layer was as impressively well mixed, but over a layer about ten times shallower. Here the measurements document the peculiar situation of salinity maintaining the stratification that caps the downward growth of the ocean mixed layer. Ship-based measurements of the air-sea interface were greatly extended by five sail-drones, three wave-gliders, six

Swift drifters, two autonomous prototype drifters (OCARINA and PICCOLO), and twenty-two drifters. In Fig. 8 the air-sea temperature difference of about $0.8\,\mathrm{K}$ is based on sail-drone data, which also quantifies the role of moisture in driving density differences. During EUREC[4]A more than half of the density difference between the near-surface air, and air saturated at the skin-temperature of the underlying ocean, can be attributed to variations in the specific humidity.

Kite stabilized helium balloons, known as Max Planck Cloud Kites (MPCKs, or CloudKites) made their campaign debut

during EUREC[4]A. Three instrument systems were flown. One large MPCK+ instrument was flown on the R/V MS-Merian on the larger aerostat (115 kg lift, 1.5 km ceiling) to sample clouds. Two smaller mini-MPCK instruments were flown both on the same aerostat and the smaller aerostat on the R/V Meteor (30 kg lift, 1 km ceiling) which focused on boundary layer and cloud-base profiling. Measurements from the CloudKites are used to quantify the cloud coverage in Fig. 8.

## 3  EUREC[4]A's Seven Science Facets

In this section we elaborate on scientific (and social) topics that motivated EUREC[4]A and how the measurements were specifically performed to address them. The presentation aims to emphasize advances as compared to what had been possible in the past, yet not lose sight of the need to also provide a clear sketch of the campaign as a whole. Additional details describing the activities of specific platforms, or groups of platforms, are being described in complementary data papers, and a full listing of the instrumentation deployed is presented in appendices.

### 245  3.1  Testing hypothesized cloud-feedback mechanisms

As described by Bony et al. (2017), EUREC[4]A was conceived as a way to test the hypothesis that enhanced mixing of the lower troposphere desiccates clouds at their base, in ways that warming would enhance (Rieck et al., 2012; Sherwood et al., 2014; Brient et al., 2016; Vial et al., 2016), but the signal of which has not been possible to identify in past measurements (Nuijens et al., 2014). In addition, recent research suggests that clouds in the trades tend to organize in mesoscale patterns

(Stevens et al., 2019b) selected by environmental conditions (Bony et al., 2020). These findings raise the additional question as to whether changes in the mesoscale cloud organization with evolving environmental conditions might play a role in low-cloud feedbacks. To address these questions, EUREC[4]A developed techniques to measure the strength of convective-scale and large-scale vertical motions in the lower troposphere, to estimate the cloud fraction near cloud-base, and to quantify possible drivers


of changes in mesoscale cloud patterns, such as coherent structures within the sub-cloud layer, radiative cooling or air-mass

trajectories, as well as their subsequent influence on cloud properties.

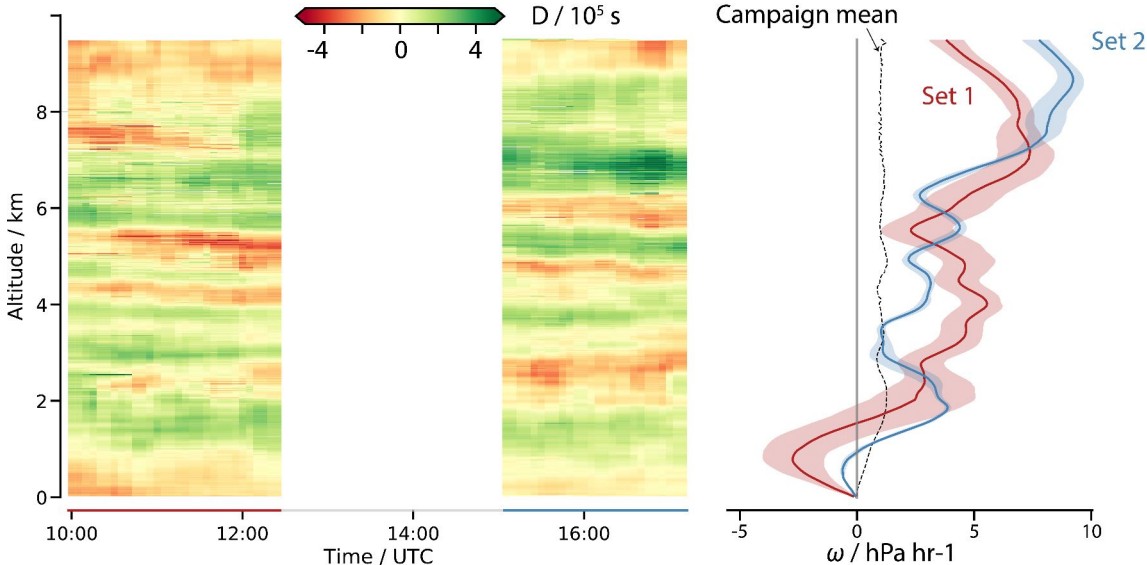

**Figure 9.** Divergence of the horizontal wind from dropsonde measurements (left) and the vertical pressure velocity derived from these (right) for the two sets of circles flown on Feb. 5. Black dashed line on right-most panel denotes vertical pressure velocity averaged over all sondes, from all HALO flights.

To make the desired measurements required HALO and the ATR to fly closely coordinated flight patterns, ideally sampling different phases of the diurnal cycle (Vial et al., 2019). This was realized by HALO circling at an altitude of $10.5\,\mathrm{km}$, three and a half times, over $210\,\mathrm{min}$. Within this period three full sounding circles were defined by a set of twelve dropsonde launches, one for each $30°$ change in heading. The start time of successive sounding circles was offset by fifteen minutes so

as to distribute the sondes through the period of circling. During this time HALO also provided continuous active and passive remote sensing of the cloud field below. Flying $50\,\mathrm{min}$ 'box' patterns just above the estimated cloud base (usually near a height of about $800\,\mathrm{m}$, Fig. 3), the ATR provided additional remote sensing, as well as in-situ turbulence and cloud microphysical measurements. After two-to-three box patterns, the ATR flew two-to-four 'L'-shaped wind-aligned and wind-perpendicular patterns (the 'L' in Fig. 1) at the top, middle and bottom of the sub-cloud layer, before returning to Barbados to refuel for a

second mission. While the ATR was refueling, HALO made an excursion, usually in the direction of the R/V Ron Brown and the NTAS buoy. On all but two occasions the ATR returned to the measurement zone after refueling (about $90\,\mathrm{min}$ later) to execute a second round of sampling, accompanied by HALO returning for another $210\,\mathrm{min}$ tour of the EUREC[4]A-Circle. All told this resulted in eighteen coordinated ($4\,\mathrm{h}$ flight segments), one of which involved the P-3 substituting for HALO on one of its night-time flights.





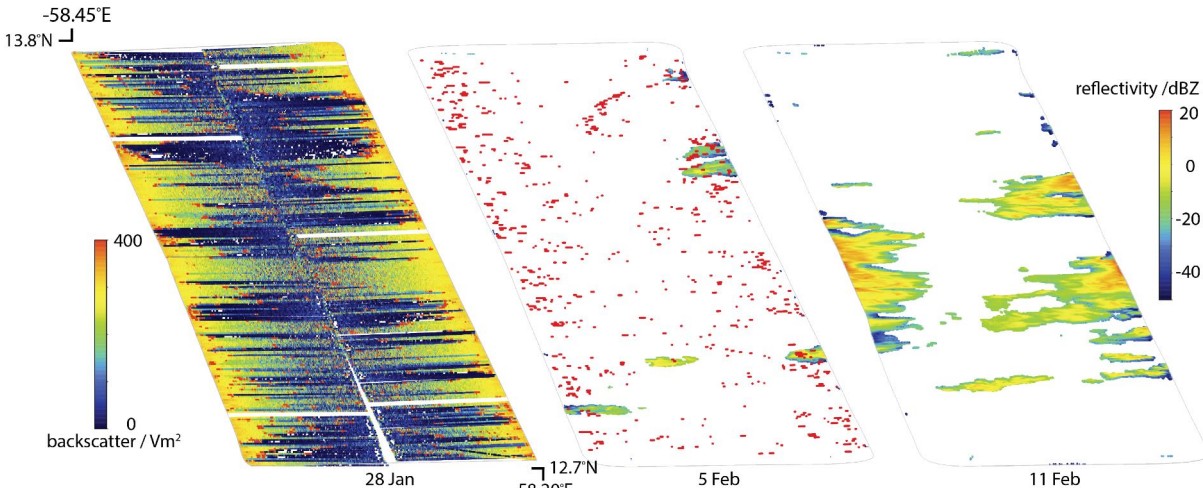

**Figure 10.** Illustration, for January 28, February 5, and February 11, of the cloud field observed at cloud base by the ATR with horizontal lidar and radar measurements. The left panel (28 Jan) shows only lidar data (attenuated backscatter signal corrected for molecular transmission), the right panel (11 Feb) shows only radar-reflectivity data and the middle panel (5 Feb) shows radar reflectivity and lidar cloud mask.

A first target of the flight strategy was the measurement, for each sounding circle, of the vertical profile of mass divergence using dropsondes following Bony and Stevens (2019). In Fig. 9 the vertical pressure velocity, $\omega$, estimated from this divergence is averaged over a set of three circles for the two sets of circles flown on Feb. 5. Also shown is the average over all circles over all days. The continuity of the divergence within a circle, and across two sets of circles – although on some flights vertical motion can change more markedly across sets of circles – gives confidence that the measurements are capturing a physical signal. It also shows, for the first time from measurements on this scale, how the mean $\omega$ reduces to the expected climatological profile, with a magnitude (of about $1\,\mathrm{hPa\,h^{-1}}$) similar to what is expected if subsidence warming is to balance radiative cooling.

The second target of the flight strategy was the measurement of the cloud fraction at cloud base through horizontal lidar-radar measurements by the ATR. In fields of optically-thin shallow cumuli (such as those associated with the cloud patterns observed on January 28), cloud droplets were too small to be detected by the radar but the lidar could detect the presence of many successive clouds along a roughly $10\,\mathrm{km}$ line of sight (i.e. half of its box-pattern width, Fig. 10). In the presence of larger cloud droplets, normally associated with larger or more water laden clouds, such as on February 11, the radar detected larger droplets and rain drops over a range of $10\,\mathrm{km}$ (Fig. 10). The lidar-radar synergy will provide, for each rectangle, the cloud fraction and the distribution of cloud geometric and optical properties at cloud base. The second, vertically pointing ATR cloud radar, allows a characterization of the aspect ratio of clouds, which may help infer the mesoscale circulations within the cloud field. These measurements, associated with new methods developed to estimate the cloud-base mass flux (Vogel et al., 2020), and to characterize the mesoscale cloud patterns from GOES-16, MODIS or ASTER satellite observations (Stevens et al., 2019b; Mieslinger et al., 2019; Bony et al., 2020; Denby, 2020; Rasp et al., 2020), will make it possible to test cloud



feedback mechanisms and advance understanding as to whether mesoscale cloud patterns influence the hypothesized feedback mechanisms.

## 3.2 Quantifying processes influencing warm rain formation

As highlighted by Bodenschatz et al. (2010), the range of scales, from micro to mega meters, that clouds encompass has long been one of their fascinating aspects. Measurements made during EUREC⁴A quantified, for the first time, the main processes that influence trade-wind clouds across this full range of scales. By doing so, long-standing questions in cloud physics were addressed, including: (i) whether microphysical processes substantially influence the net amount of rain that forms in warm clouds, and (ii) how important is the interplay between warm-rain development and the mesoscale organization of cloud fields. These questions identify precipitation development as the link among processes acting on different scales, and hence guided EUREC⁴A's measurement strategy.

On the particle scale, measurements were performed to characterize aerosols and to quantify how small scale turbulence mixing processes influence droplet kinematic interactions and activation. Aerosol properties and turbulence both imprint themselves on the cloud microstructure, and thereby affect the formation of precipitation (Broadwell and Breidenthal, 1982; Cooper et al., 2013; Li et al., 2018; Pöhlker et al., 2018; Wyszogrodzki et al., 2013). In most cases, not only the magnitude, but also the sign of the hypothesized effects can be ambiguous, if not controversial. For example, by acting as an additional source of CCN, Saharan dust may retard the formation of precipitation (Levin et al., 1996; Gibson et al., 2007; Bailey et al., 2013), but if present as giant CCN, it may have the opposite effect (Jensen and Nugent, 2017).

On the cloud scale, the intensity of rain and the evaporation of raindrops can lead to downdrafts, cold pools and mesoscale circulations which can lift air parcels, producing secondary and more sustained convection (e.g., Snodgrass et al., 2009). These cloud-scale circulations, which the EUREC⁴A-Circle measurements quantified, may also change the vigor and mixing characteristics of cloud. This could in turn influence precipitation formation, a process that Seifert and Heus (2013) suggest may be self reinforcing, consistent with an apparent link between precipitation and mesoscale cloud patterns such as 'Fish' or 'Flowers' (Stevens et al., 2019b).

On larger (20 km to 200 km) scales, horizontal transport which determines whether or not Saharan dust reaches the clouds, as well as factors such as the tropospheric stability, or patterns of mesoscale convergence and divergence, which influences cloud vertical development, may affect the efficiency of warm rain production. In addition to the characterization of the environment from the dropsondes, the positioning of surface measurements (R/V Meteor, R/V Ron Brown, and BCO) helped characterize the Lagrangian evolution of the flow, also in terms of aerosol and cloud properties.

Fig 11 shows an example of the cascade of measurements, spanning scales covering ten orders of magnitude. On the smallest $O(10^{-5} \text{ m})$ scale, a sample holographic image from an instrument mounted on the MPCK+ shows the spatial and size distribution of individual cloud drops. In-situ measurements and airborne remote sensing document the cloud microphysical structure and its relationship to the properties of the turbulent wind field. On scales of hundred meters to a few kilometers, vertically and horizontally pointing cloud radars and lidars characterize the geometry and the macrophysical properties of clouds. On yet



**Figure 11.** Composition of scales from different cloud sensing instrumentation, highlighting the test-section of the Tradewind Alley Cloud Chamber. Most measurements are co-located, with scanning C-band radar (POLIDRAD) scans are overlain on satellite imagery from GOES with segments of the Twin-Otter, HALO and ATR flight tracks. Radar images from the ATR (horizontal and zenith) and HALO (nadir) are shown (all radar imagery shares the same color scale) as well as cloud water and updraft velocity from a penetration of cloud (on the same day but at a different time) by the Twin-Otter. Visual image from specMACS with POLIDRAD reflectivity contours superimposed, shows the specMACS cloud visualization along flight track. MPCK+ hologram measurements (made in the southern portion of the circle – 12.25°N, 57.70°W at 1084 m – on 17 February) demonstrates the capability to measure the three dimensional distribution of individual cloud droplets colored by size.


larger $O(10^5\,\mathrm{m})$ scales, the spatial organization and clustering of clouds and precipitation features is captured by satellite, by high-resolution radiometry from high-altitude aircraft, and by the C-band scanning radar, POLDIRAD (Schroth et al., 1988).

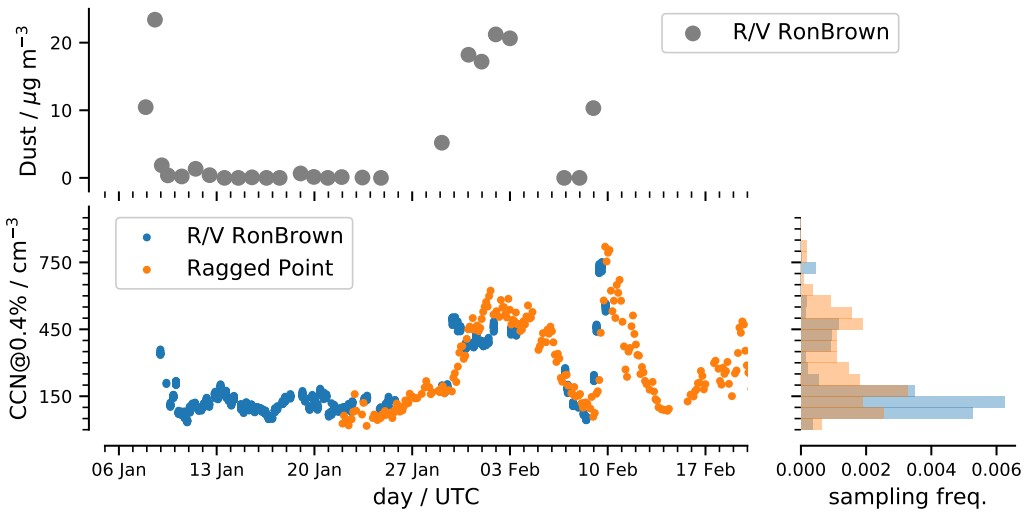

**Figure 12.** Aerosol characteristics in the Tradewind Alley over the period of measurements. Dust mass density from the R/V Ron Brown (upper left), which was mostly east of $55°W$. Normalized histogram showing the relative frequency of occurrence of different CCN concentration levels (lower right). Note that the periods of observation at the two locations are only partly overlapping.

An example of how the measurements upwind and downwind of the EUREC[4]A-Circle helped constrain its aerosol environment is shown in Fig 12. Two periods with larger CCN number concentration (near $450\,\mathrm{cm}^{-3}$), both associated with periods of elevated mineral dust, can be identified in measurements made aboard the R/V Ron Brown (East of $55°W$) and from the ground station at Ragged Point (Pöhlker et al., 2018). The slight lag of the Ragged Point measurements relative to those on the R/V Ron Brown is consistent with the positioning of the two stations and the westward dust transport by the mean flow. The episodes of elevated dust are believed to be from Saharan dust outbreaks, which are unusual in the (Boreal) winter months (J M Prospero and Carlson, 2020) and can greatly increase CCN number concentrations (Wex et al., 2016). In between these events, CCN number concentration are threefold smaller ($150\,\mathrm{cm}^{-3}$), which we take as representative of the clean maritime environment. Capturing such large perturbations in varying cloud conditions should aid efforts to untangle the relative role of different factors influencing warm-rain formation.

### 3.3 Subcloud mass, matter, energy and momentum budgets

Early field studies extensively and compellingly documented the basic structure of the lower atmosphere in the trades (Riehl et al., 1951; Malkus, 1958; Augstein et al., 1974; Brummer et al., 1974; Garstang and Betts, 1974). What remains poorly understood is the relative role of specific processes, particularly those acting at the mesoscale, in influencing this structure. A specific question that EUREC[4]A aims to answer is the importance of downdrafts, and associated cold pools (Rauber et al., 2007; Zuidema et al., 2012), in influencing boundary layer thermodynamic structure and momentum transport to the surface. A

related question is whether the links between the cloud and sub-cloud layer depend on the patterns of convective organization,

for instance as a result of differences in the circulation systems that may accompany such patterns.

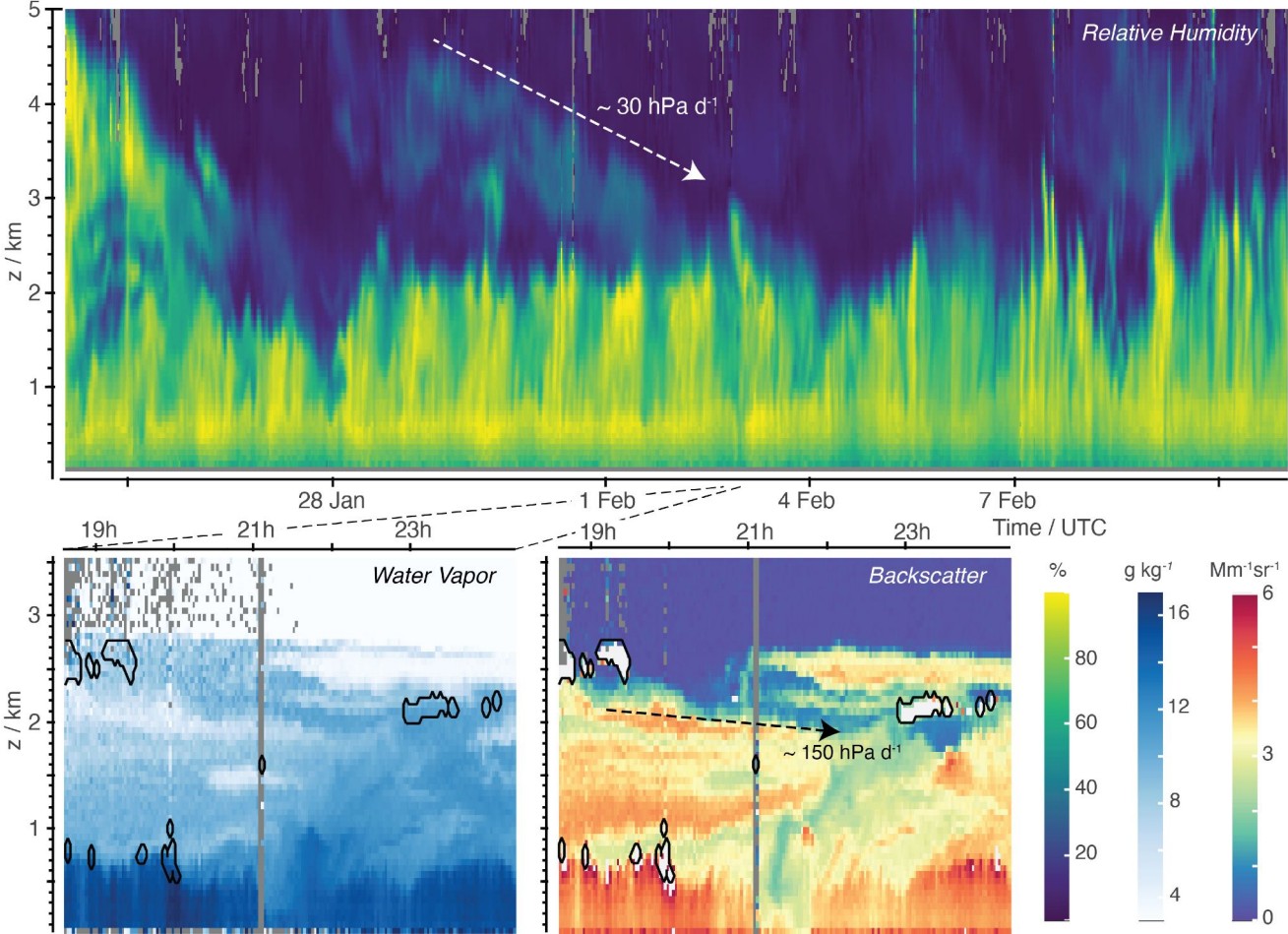

**Figure 13.** Lidar profiling of the lower atmosphere using the CORAL lidar at the BCO. Upper panel shows the relative humidity in the lower 5 km over the entirety of the campaign. Lower panels show the specific humidity over a four hour period marked by a large intrusion of cloud layer air on 2 February, and the associated aerosol/cloud backscatter. Also shown are the Lagrangian evolution of humidity, or backscatter features, as indicative of the magnitude of vertical velocity variations on different temporal scales.

For quantifying the sub-cloud layer budgets, as for many other questions, a limiting factor has been an inability to measure mesoscale variability in the vertical motion field. EUREC[4]A's measurements not only address this past short coming, but the ship-based sounding network additionally quantifies the mean vertical motion at different scales. The arrangement of measurements, particularly flight segments, was designed to quantify the Lagrangian evolution of air-masses, with legs repeated

on every mission at levels attuned to the known structure of the lower-troposphere i.e., near the surface, in the middle, near the top and just above the sub-cloud layer, as well as in, and just above, the cloud layer. Past studies using a single aircraft, albeit



in a more homogeneous environment, demonstrate that such a strategy can close boundary layer moisture and energy budgets (Stevens et al., 2003). Doing so also aids quantification of the vertical profile of turbulent transport and contributions associated with horizontal heterogeneity and sets the stage for estimating mass and energy budgets through the entire atmospheric column.

To address the measurement challenge posed by an environment rich in mesoscale variability, EUREC$^4$A made use of additional aircraft and a larger array of surface measurements (also from uncrewed platforms) as well as extensive ship and airborne active remote sensing, and a network of water stable-isotopologues (as presented in 2.3.2). At the BCO, aboard the R/V Meteor and on the R/V MS-Merian, advanced Raman lidars provided continuous profiling of water vapor, clouds, temperature and aerosols. The nadir staring WALES lidar on HALO likewise profiled water vapor, clouds and aerosols. As an

example of this capability, Fig. 13 presents relative humidity data (deduced from temperature and absolute humidity retrievals) from the BCO lidar. These measurements document the time-height evolution of water vapor in the boundary layer, something impossible to assess from in situ measurements, which measure at only a few levels, or soundings, which are sparse in time.

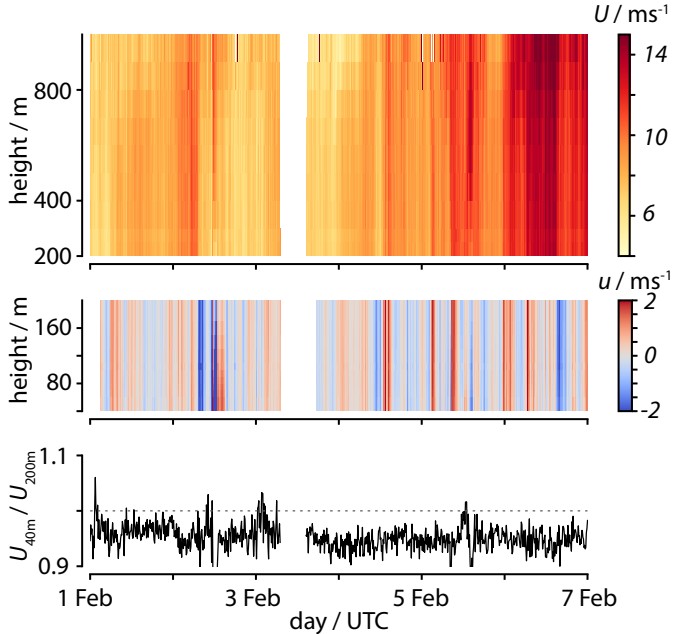

**Figure 14.** Wind lidar profiling of sub-cloud layer winds from the R/V Meteor. The upper panel shows the value of the wind speed in the sub-cloud layer, above 200 m. Fluctuations of the near surface wind speed from a three-hourly running mean value are shown in the zoom in the middle panel. Lower time-series shows the ratio of the wind at the lowest remotely-sensed level (40 m) from its value at 200 m.

The BCO lidar measurements quantify the structure of moist or dry layers in the free atmosphere, as well as variations in the cloud and sub-cloud layers, illustrating days of more nocturnal activity (centered on Feb 1), and also features presumed to

be the signature of mesoscale circulations. The latter is the focus of the zoom in the lower panels which shows the lower 3 km over five hour period late on February 2, 2020. It shows a period where aerosol-poor air appears to descend adiabatically into the cloud layer (near 2 km, coincident with a large-scale fold of cloud-layer air into the sub-cloud layer. This results in a sharp





contact discontinuity (aerosol front) near 21 UTC, which extends to the surface and is also evident in the water-vapor field. Typically the marine boundary (sub-cloud) layer is viewed as a turbulent layer that primarily interacts with the much larger-scale evolution of the free atmosphere through small scale entrainment at its top. Events such as the one shown in Fig. 13, suggest that in addition to downdrafts and the cold pools they feed, circulations on scales commensurate with and larger than the depth of the sub-cloud layer may be important for boundary layer budgets.

Similar considerations also apply to the momentum budget of the trades. In Dixit et al. (2020) idealized large-eddy simulations are shown to under-estimate the flux of momentum in the sub-cloud layer, something they hypothesize to arise from an absence of mesoscale circulations in the simulations. As an example of efforts to quantify such processes Fig. 14 shows the total wind speed measured in the sub-cloud layer by the long-range wind lidar aboard the R/V Meteor. The lower panel documents kilometer-scale wind speed variations on the order of $2\,\mathrm{m\,s^{-1}}$ that extend into the surface layer (derived from the short-range wind lidar, defined with respect to three-hourly running means). One question asked is whether at a given surface friction, convectively driven flows can sustain a relatively large near-surface wind, and weaker surface layer wind shear, than expected from shear-driven turbulence alone. The third panel shows that the ratio of 40 m to 200 m wind speeds, as a measure of surface layer wind shear, is regularly close to 1. Combined with surface heat and momentum fluxes measured by other platforms, the lidars provide a unique opportunity to identify the influence of (moist) convection on wind stress at the surface.

### 3.4  Ocean mesoscale eddies and sub-mesoscale fronts and filaments

Mesoscale eddies, fronts, and filaments – not unlike the mesoscale circulations that are the subject of increasing attention in the atmosphere – are coherent structures that may be important for linking surface mixed layer to the interior ocean dynamics (Carton, 2010; Mahadevan, 2016; McWilliams, 2016). By virtue of a sharp contrast with their surroundings, these structures can efficiently transport enthalpy, salt, and carbon through the ocean. Though satellite observations have enhanced knowledge of their occurrence and surface imprint Chelton et al. (2001), the sparsity of direct observations limits our ability to test our understanding of such structures, in particular subsurface eddies. Understanding of the role of these types of structures is further limited by their short lifespans (hours to days) and small spatial scales (0.1 km to 10 km), which make them difficult to observe. These facts motivated ocean observations during EUREC[4]A, as did recent work suggesting that such coherent structures, in particular localized upwelling, downwelling, straining, stratification variability, wave breaking, and vertical mixing, may couple with and influence atmospheric processes, including cloud formation (Lambaerts et al., 2013; Renault et al., 2016; Foussard et al., 2019).

To address these questions, measurements during EUREC[4]A attempted to quantify how near-surface currents, density, and waves varied across and within different dynamical regimes, e.g., for mesoscale eddies, fronts, and filaments. Such measurements aimed to answer specific questions not unlike those posed for the atmospheric boundary layer, namely to quantify the contribution of such structures to the spatial and temporal variability of the upper ocean. EUREC[4]A distinguished itself from past campaigns that have attempted similar measurements – LatMix: (Shcherbina et al., 2013); OSMOSIS: (Buckingham et al., 2016); CARTHE: (D'Asaro et al., 2018) – by virtue of the number and diversity of observing platforms deployed (Saildrones, underwater gliders, instrumentally enhanced surface and subsurface drifters, wave gliders, an Autonaut, and biogeochemical



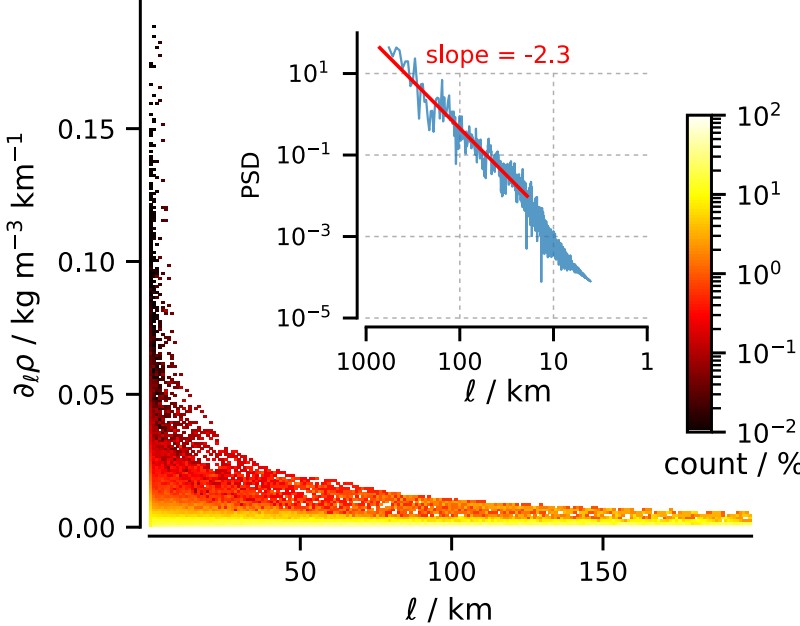

**Figure 15.** Percentage count of surface density gradients at different horizontal length scales (1 km to 200 km) measured by Saildrones; note the log scale colorbar. Inset is power spectral density of the surface density gradients, calculated by averaging Periodograms constructed for each vehicle after detrending the data and smoothing the data with a 2 km Gaussian filter. The red line shows the linear regression best fit slope of -2.3.

Argo floats). These mapped the ocean down to 1000 m or more, simultaneously across both the Tradewind Alley and the Boulevard des Tourbillons (Fig. 2). These measurements have resulted in an unprecedented view of a large spectrum of ocean temporal and spatial scales across different oceanic environments.

The richness of structure observed in the upper ocean during EUREC[4]A can be quantified by the distribution of surface temperature fronts. All seagoing platforms contributed to observing the upper-ocean temperature structure, surveying a wide region and a large spectrum of ocean scales, and thus can contribute to this measure of upper ocean variability. An example from one such platform, a Saildrone, is shown in Fig. 15. The sensitivity of frontal density gradients to spatial resolution was explored by subsampling data from 0.08 km to 100 km (Fig. 15). For each length scale, the percentage frequency of each

density gradient was calculated. This analysis demonstrates that smaller length scales yield larger density gradients. The largest gradients were found at spatial scales of only 1 km and were associated with strong, local freshening. These are believed to be associated with small-scale, but intense, rain showers, a potentially far-reaching idea given the importance of rain for linking processes at different scales in the atmosphere (e.g., §3.2). The analysis further documents self-similar (powerlaw) scaling between 19 km and 1900 km with a slope of $-2.3$. There is evidence of a scale break at around 25 km. Surface quasi-

geostrophic turbulence generally predicts a slope of $-5/3$ or steeper (Callies and Ferrari, 2013; Rocha et al., 2016; Lapeyre, 2017).



**Figure 16.** Measurement of a subsurface freshwater eddy near 58°W 10°N. Background map presents Absolute Dynamic Topography from satellite altimetry (SALTO/DUACS). This shows the remotely sensed surface eddy field (Pujol et al., 2016), with features moving toward the northwest through the Boulevard des Tourbillons, and to the West along Tradewind alley. Eddy contours as detected automatically by the TOEddies algorithm (Laxenaire et al., 2018). The position of subsurface eddies (200 m to 600 m deep) as identified from the eddy detection method (Nencioli et al., 2010) applied to vector currents measured by Ship Acoustic Doppler Current Profilers (SADCP) are shown by red circles. Overlain vertical transects show the zonal velocity component from the two SADCPs of the R/V Atalante, and the salinity from CTD soundings as measured across one of the sections (A-B) that sampled surface and subsurface eddies evolving in the region. The surface and subsurface eddies appear to be evolving independently. The subsurface eddy freshwater anomaly is indicative of South Atlantic origins.





A wide array of instruments deployed from all four ships (CTDs, underway CTDs, Mounted Vessel Profilers, microstructure profilers, XBTs, XCTDs, Doppler current meter profilers, 5 BGC Argo floats) and the seven ocean gliders (e.g., Fig. 5) profiled water properties and ocean currents. This array of measurements, guided by near-real time satellite data and real-time ship
profiling, revealed a surprisingly dense and diverse distribution of mesoscale eddies. All of the measured eddies captured by satellite data (Fig. 16) were shallow, extending to a depth of about $150\,\mathrm{m}$ (Fig. 16) and transporting warm and salty North Atlantic tropical water swiftly northward. Below but not aligned with the surface structures and separated by strong stratification, large subsurface anticyclonic eddies (and on some occasions cyclonic eddies) extended from $150\,\mathrm{m}$ to $800\,\mathrm{m}$ and carried large quantities of water from the South Atlantic northward. An example sampled by R/V Atalante along a Southwest
to Northeast aligned transect near 50°N and 58°W is illustrated in Fig. 16. Here a ca $200\,\mathrm{km}$ eddy characterized by a $0.2\,\mathrm{PSU}$ freshwater anomaly was measured carrying water, which likely subducted in the south Atlantic, northward. The anomaly was associated with a circulation of $\approx 1\,\mathrm{m\,s^{-1}}$ with maximum velocities near $300\,\mathrm{m}$ extending downward to a depth of about $800\,\mathrm{m}$. EUREC[4]A observations such as these will be essential for understanding the complex dynamics of the upper ocean, and the extent to which they can be captured by a new generation of $\mathrm{km}$ scale coupled climate models.

## 3.5 Air-sea interaction

What distinguished EUREC[4]A from the many previous campaigns focused on air-sea interaction was its interest in assessing how circulation systems, in both the ocean and the atmosphere, influence surface exchange processes. These interests extended to interactions with ocean biology and their impact on both $CO_2$ exchange and profligate amounts of seaweed (*Sargassum*) that have, in past years, developed into a regional hazard. To study these processes EUREC[4]A made use of a flotilla of uncrewed
devices, and a wealth of nadir staring airborne remote sensing, specifically designed to characterize the air-sea interface on a range of scales.

Ocean eddies, fronts and filaments, influence the atmosphere by perturbing air-sea surface fluxes (Chelton and Xie, 2010; O'Neill et al., 2012) a process that may also feedback on the ocean by causing a damping of the (sub)mesoscale activity (Renault et al., 2018). As an example, Sullivan et al. (2020) use large-eddy simulation to show how small scale ocean fronts
perturb the boundary layer through its depth, giving rise to circulations on scales much larger than that of the boundary layer, or of the front itself (their Fig. 12). These lead to large perturbations in vertical mixing and, one can speculate, on patterns of cloudiness. Similarly, clouds influence the downward longwave and shortwave irradiance, which influences both the sea-surface temperature, but also atmospheric temperatures directly, something that Naumann et al. (2019) have shown to commensurately power ($2\,\mathrm{km}$ to $200\,\mathrm{km}$) circulations.

In the area of intensive measurements near and within the EUREC[4]A-Circle (Region A), measurements sought to quantify how surface exchange processes vary with circulation (cloud pattern) regime. Measurements by Caravela (an AutoNaut) and three seagliders characterized the air-sea interface in a small, and spatially fixed, (ca $10\,\mathrm{km}$) region in this domain (Fig. 17). These measurements help untangle spatial from temporal variability, with both a secular (seasonal) cooling of surface waters over the course of the campaign and a variable, but at times pronounced, diel-cycle (Fig. 17). In addition, CTD casts, lower
atmospheric profiling (with a mini-MPCK and a Quadcopter), and eddy-covariance measurements from an outrigger mast, were



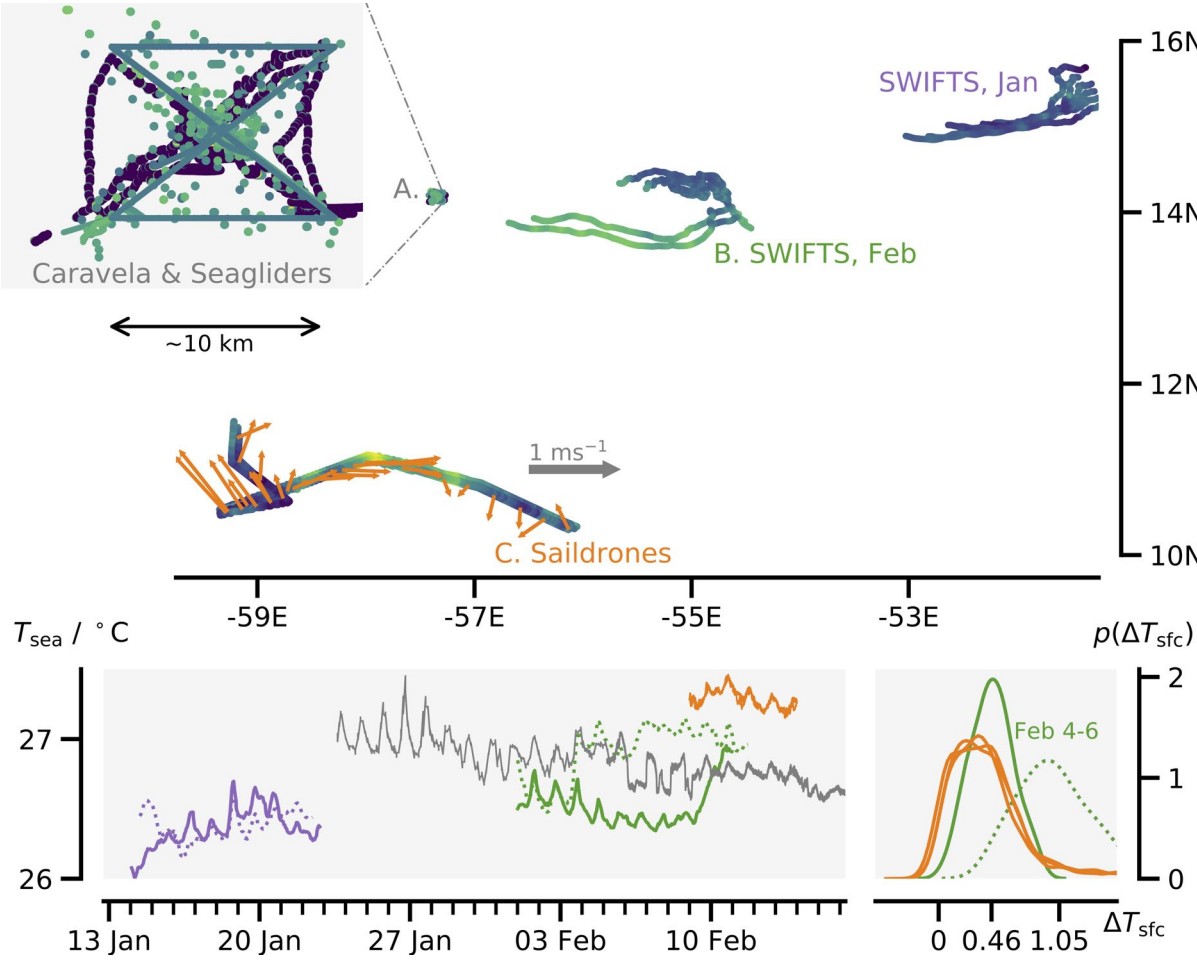

**Figure 17.** Uncrewed vehicles were used to study different aspects of air-sea interaction in all three of the EUREC[4]A study regions. Upper panel shows the tracks of the instruments colored by their measurements of near surface water temperature, $T_{\text{sea}}$. The zoom (upper left) expands the domain of the Caravela (and Seaglider) measurements in Region A (near 57°W). January and February SWIFT buoys ($T_{\text{sea}}$ at $-0.3\,\text{m}$) deployments in Region B. Saildrones ($T_{\text{sea}}$ at $-0.5\,\text{m}$) measurements across an eddy near 11°N, with anti-cyclonic currents (at $-5\,\text{m}$) shown by vectors, in Region C. Lower-left panel shows time-series of $T_{\text{sea}}$ measurements by the different instruments. Distribution of air-sea temperature differences measured by two SWIFT buoys between $4\,\text{UTC}$ Feb 4 and $14\,\text{UTC}$ Feb 6 (lower right).





performed by the R/V Meteor as it steamed up and down the $57.25°$W meridian bisecting the EUREC[4]A-Circle just upwind of Caravela's box. Rounding out the measurements in this region were low-level Twin-Otter, ATR (as part of its 'L' pattern) legs, and Boreal UAS measurements, as well as airborne remote sensing of sea-surface temperatures along the EUREC[4]A-Circle by HALO.

Effects of ocean sub-mesoscale processes on air-sea interactions were the focus of measurements in Region B (Fig. 1). On two occasions the R/V Ron Brown deployed six SWIFT drifters (spar buoys) in regions of surface heterogeneity; once in January near the NTAS buoy, and again in early February near $55°$W. The deployments were performed and coordinated with further measurements by the R/V Ron Brown, as well as by the P3, two wave-gliders and a saildrone. The P-3 (see also Fig. 4 and Fig. 2) dropped AXBTs around the SWIFTS, quantified air-sea exchange with near surface flight legs, and surveyed

the near surface wind and wave fields using remote sensing. Fig. 17 documents how, during the February deployment, the SWIFTS sampled large $0.5\,\mathrm{K}$ mesoscale (ca $30\,\mathrm{km}$) variability in SST features. This variability gives rise to air-sea temperature differences twice as large as the baseline, as inferred from the average of measurements over longer periods (i.e., as shown by the Saildrone data (orange lines) and is characteristic of the SWIFT data away from the local feature in surface temperatures (e.g., green-solid line in Fig. 17).

In the Boulevard des Tourbillons (Region C), coordinated sampling between Saildrones and two research vessels aimed to quantify mesoscale and submesoscale air-sea interaction. Submesoscale variability and strong near-surface currents, with a circulation indicative of an NBC ring, were measured by the Saildrones (Fig. 17). These measurements were coordinated with the activities of the R/V Atalante and R/V MS-Merian, and three seagliders (e.g., Fig. 4). Extensive vertical profiling, also by high-speed underway CTDs, aimed to quantify the impacts of submesoscale fronts and filaments and mesoscale eddies on

surface exchange processes, and vice versa. Being able to resolve the thermal structure of the upper ocean should also help quantify the importance of the O(0.3K) cool-skin effect and diurnal warming just below the skin-layer (Fairall et al., 1996) on ocean mixing and air-sea exchange.

Factors, including the role of meso and sub-mesoscale variability, influencing air-sea gas exchange were also studied. $pCO_2$ were made on the R/V Atalante, R/V MS-Merian and the R/V Ron Brown (Fig. 18). In addition, both the R/V MS-Merian and

R/V Meteor regularly sampled water at four different depths (selected based on chlorophyll concentrations) for $N_2$ fixation and primary production rates as well as potential aerobic methane production. DNA- and RNA-based sequencing will additionally be performed on these water samples to identify diazotrophic community members, potentially including so far unrecognized members. Furthermore, large floating mats of seaweed (genus *Sargassum*) were observed from all crewed platforms. On the R/V MS-Merian, to investigate if, and to what degree, this biomass and primary production can be supported by local $N_2$

fixation, incubation experiments including stable isotopes were conducted on seaweed samples that were collected underway. In addition to extending studies of air-sea interaction to incorporate chemical and biological processes, EUREC[4]A may also shed light on the role of meso and submeso-scale ocean circulations on these chemical and biological processes.

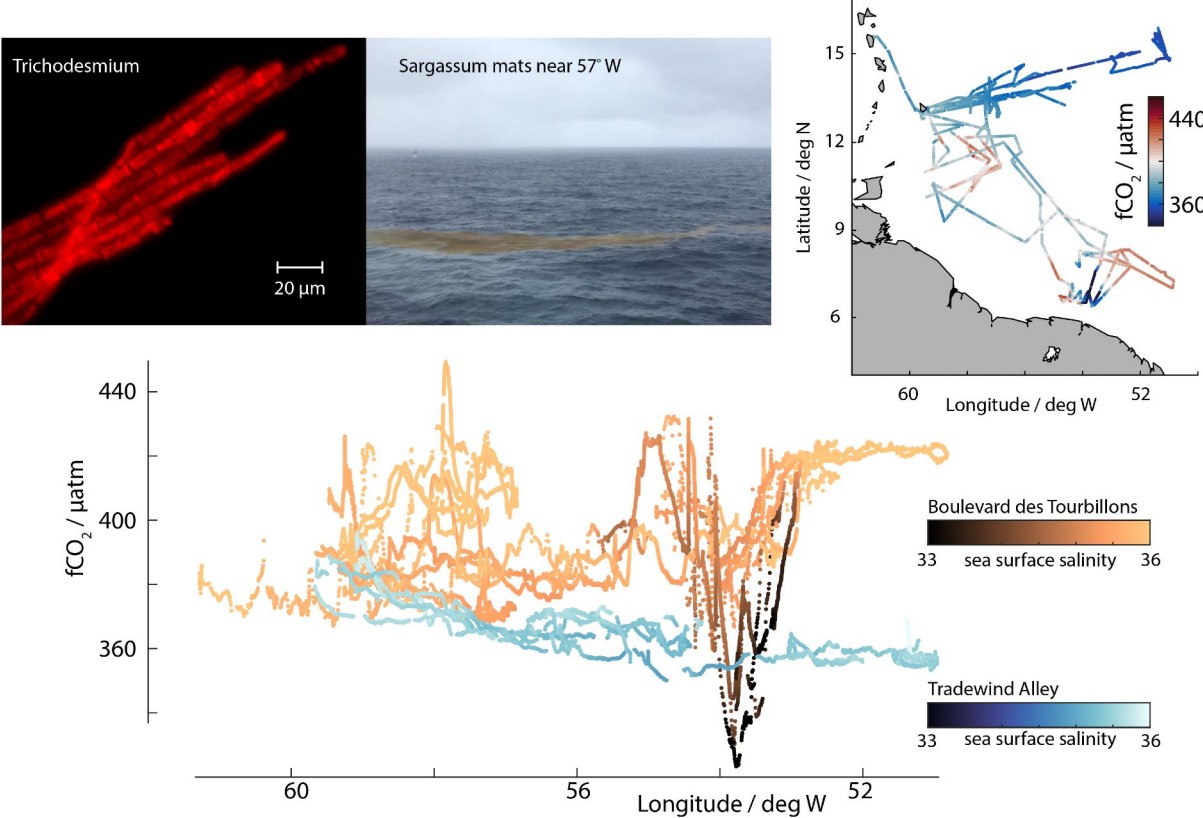

**Figure 18.** Measurements of $CO_2$ fugacity (f$CO_2$) from the different surface vessels shown versus longitude (lower panel). The presentation contrasts strong variability in f$CO_2$ in association with eddies and salinity variations along Boulevard des Tourbillons (oranges in lower panel, track in upper right) versus in the Tradewind Alley (blues). Microscopic/epifluorescence image of several filaments of *Trichodesmium*, an $N_2$-fixing cyanobacterium that is found in the region (upper left), and mats of seaweed (*Sargassum*, photo W. Mohr, upper center) which were frequently observed and difficult to navigate from some of the uncrewed surface vehicles.

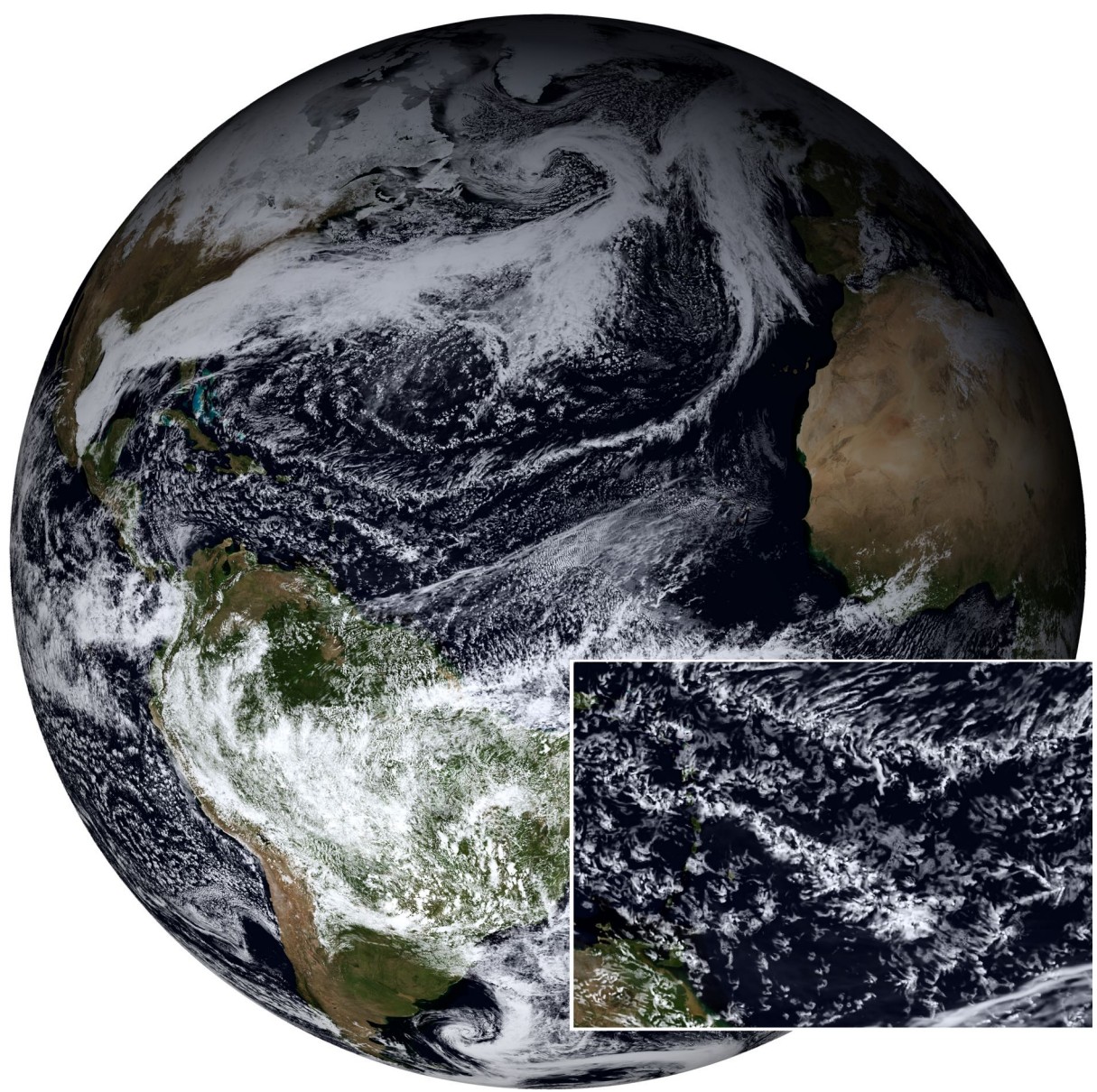

**Figure 19.** Global 2.5 km mesh coupled simulations performed by ICON as part of DYAMOND-Winter for the EUREC$^4$A period. The snapshot, with a zoom over the study region to show the degree of detail in the simulations, was taken from Feb 2 of a simulation initialized on January 20 and allowed to freely evolve thereafter.





### 3.6 Benchmarks for modelling and satellite retrievals

The range of scales and types of processes that can presently be captured by both satellites and models, and the extent to which
they were integrated into EUREC[4]A's experimental design (cf., Bony et al., 2017), allows EUREC[4]A to address questions
that could not be addressed with data from earlier field studies. For instance, what resolution is required for atmospheric
models with an explicit (fluid-dynamical) representation of clouds and convection to represent the vertical structure of the
lower troposphere, and its interaction with mesoscale vertical motion and upper ocean variability, within the observational
uncertainty? The fine-scale of the EUREC[4]A measurements also makes it possible to quantify satellite retrieval uncertainty,
for instance for measurements of small scale precipitation features, cloud microphysical properties, or column energy budgets
(Illingworth et al., 2015).

For these purposes EUREC[4]A was closely coordinated with efforts to develop and test a new generation of Earth-system
models. Recently, following the pioneering efforts of Japanese colleagues (Tomita et al., 2005), a number of groups in other
countries have demonstrated – within the DYAMOND[3] project (Stevens et al., 2019c) – the capability of performing km-
scale simulations on global (atmospheric) grids (Satoh et al., 2019). A follow up, called DYAMOND-Winter is extending this
capability to also include coupled global models and has been coordinated to simulate the EUREC[4]A period. DYAMOND-
Winter simulations (Klocke et al., in preparation) are being initialized from observational analyses on 20 Jan, and run for at
least forty-days. With grid-scales of a few km in both the atmosphere and ocean the simulations explicitly represent scales
of motion similar to those observed, all as part of a consistently represented global circulation. This enables investigations of
processes influencing the mesoscale organization of fields of shallow convection, including the possible role of surface ocean
features, as well as a critical evaluation of the simulations.

An example of a DYAMOND-Winter simulation using ICON is given in Fig. 19. The simulated cloud fields exhibit rich
mesoscale variability whose structure, while plausible, begs a more quantitative evaluation. The combination of the field mea-
surements and simulations with realistic variability on the mesoscale, will aid efforts to test retrievals of physical quantities
from satellite radiances. This should make it possible to establish a self consistent and quantitative understanding of controls
on cloudiness.

In addition to the global coupled modelling activities, coordinating modelling activities using much higher resolution (meters
to tens of meters) simulations of the ocean, atmosphere, and the coupled system, over a limited area are ongoing. These include
idealized simulations with doubly periodic boundary conditions, atmospheric simulations designed to track the Lagrangian
evolution of the flow, and simulations with open boundaries matched either to meteorological/oceanographic analyses or the
free running global simulations. Few if any field studies have benefited from such a rich complement of modelling activities.

Some of the challenges to evaluating these simulations are illustrated with the help of preliminary, but idealized, large-eddy
simulations with the forcing specified based on preliminary data in a manner similar to what has been adopted in past studies
(e.g., Stevens et al., 2005; vanZanten et al., 2011), albeit over considerably larger domains. Fig. 20 shows, with the help of
a satellite image, the degree of mesoscale cloud variability. This apparent whimsicality suggests that, given the imprecision

---

[3]the DYnamics of the Atmospheric general circulation Modeled On Non-hydrostatic Domains





in the forcing and the cloud retrievals, assessing the magnitude of systematic biases in the simulations will be a challenge. In this case, the simulations performed for the mean conditions in the vicinity of 'D' seem implausible. The challenge will be to assess to what extent this reflects imprecision in the forcing, of the sort that differentiates the different marked regions in the figure.

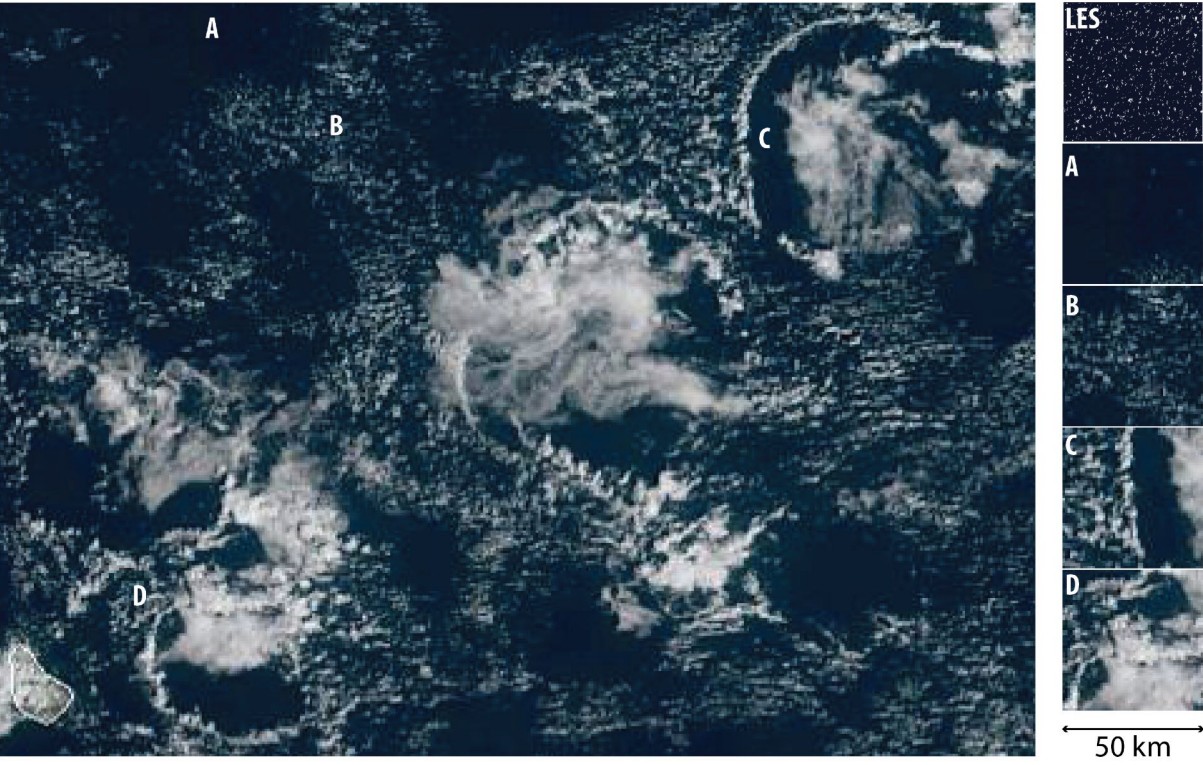

**Figure 20.** Geostationary satellite image showing the cloud field in the measurement area on 5 February, 2020. Snapshots of cloud fields over $50\,\mathrm{km} \times 50\,\mathrm{km}$ subdomains labeled 'A' to 'D' are compared to large-eddy simulation. The Large-eddy simulation employed a $100\,\mathrm{m}$ mesh and was configured in the traditional way with doubly periodic horizontal boundary conditions and with an applied horizontally homogeneous mean forcing estimated from measurements in the vicinity of subdomain 'D'.

Given a demonstration that fine-scale models can quantitatively represent the macro-structure of the observed clouds, EUREC$^4$A measurements are expected to provide benchmarks for the simulation of cloud microphysical process. This would allow the first ever evaluation of the ability of microphysical models, which depend on a variety of parameterized processes, to quantitatively represent precipitation formation processes in realistically simulated cloud fields. Previous attempts (Ackerman et al., 2009; vanZanten et al., 2011) at making such an evaluation have highlighted large differences in models, but it remains unclear to what extent these differences are due to the representation of cloud macrophysics, versus microphysics. Greater confidence in the fidelity of these simulation approaches will also greatly benefit their application to questions in remote sensing.





## 3.7 Scientific outreach and capacity building

A core, and hopefully sustainable, feature of the EUREC⁴A field campaign was the rich human and scientific interactions with the Barbadian public, the regional research community, and the larger community of scientists from outside of the region.
Activities that permitted these exchanges included operational support for flight planning, inclusion in flight-teams and on-ship data collection teams, weekly seminars, a larger symposium, as well as scientific outreach to schools and to the general public. A total of more than twenty-five researchers as well as representatives of regional governments contributed to the data collection. This participation was essential not just for meeting's EUREC⁴A's original objectives, it also expanded the scope of activities to include the development of regional climate resilience and capacity building in the use of advanced weather and
climate early warning systems.

### 3.7.1 Operational support

Daily operational meetings were hosted at a facility shared by the Barbados Meteorological Service and the Department of Civil Aviation. Scientists from National Meteorological Services across the region supported the effort by providing daily weather forecasts which helped to coordinate the measurements for the following days. The European Centre for Medium Range
Weather Forecasts and national weather services in France, Germany, Holland, the UK and US supported these activities by providing access to output from global models at $10\,\mathrm{km}$ resolution and regional forecasts made specially for the region on $\mathrm{km}$ scale grids. The daily weather discussions provided an opportunity for scientists from different teams to discuss and analyze the early results of the campaign and the perspectives ahead. For example, during one of these meetings it was learned that the mesoscale cloud pattern identified as 'Fish' in the recent literature, have long been termed 'Rope' clouds by the regional
forecast community.

### 3.7.2 Symposium and scientific seminars

Knowledge transfers of immense value were facilitated by the organization of regular scientific presentations that provided an opportunity for exchange among EUREC⁴A participants and researchers at the Caribbean Institute for Meteorology and Hydrology (CIMH). It is expected that such exchanges will sustain collaborations well beyond the campaign. Keynote presen-
545 tations at the Barbados Museum and Historical Society brought to a general audience the goals of the EUREC⁴A campaign, the very early history of meteorology on Barbados and issues or relevance to climate change and related adaptation.

Campaign participants also celebrated the 50 anniversary of the Barbados Oceanographic and Meteorological EXperiment (BOMEX) field campaign with a two-day public symposium entitled "From BOMEX to EUREC⁴A ". The symposium brought together a varied audience, including regional and international scientists, EUREC⁴A participants, and students from the Uni-
550 versity of the West Indies Cave Hill Campus, Fig. 21. The symposium provided an opportunity to reflect upon the evolution of climate research during the past 50 years. From fascinating speeches by BOMEX veterans, to presentations describing the state of present day understanding as expressed in EUREC⁴A's objectives, the symposium helped contextualize the efforts being made as part of EUREC⁴A.





In December 2019, prior to the start of the EUREC⁴A campaign, the Caribbean Meteorological Organization (CMO) Head-

555 quarters Unit and the University of Leeds, with the assistance of CIMH, organized a 'Caribbean Weather Forecasting Initiative'4 workshop at CIMH, to promote knowledge exchange between researchers and forecasters from CMO Members and other Caribbean States. This activity supported the EUREC⁴A Forecast Testbed, in which many of the regional forecasters who participated in the workshop, provided some of the daily forecasts previously discussed.

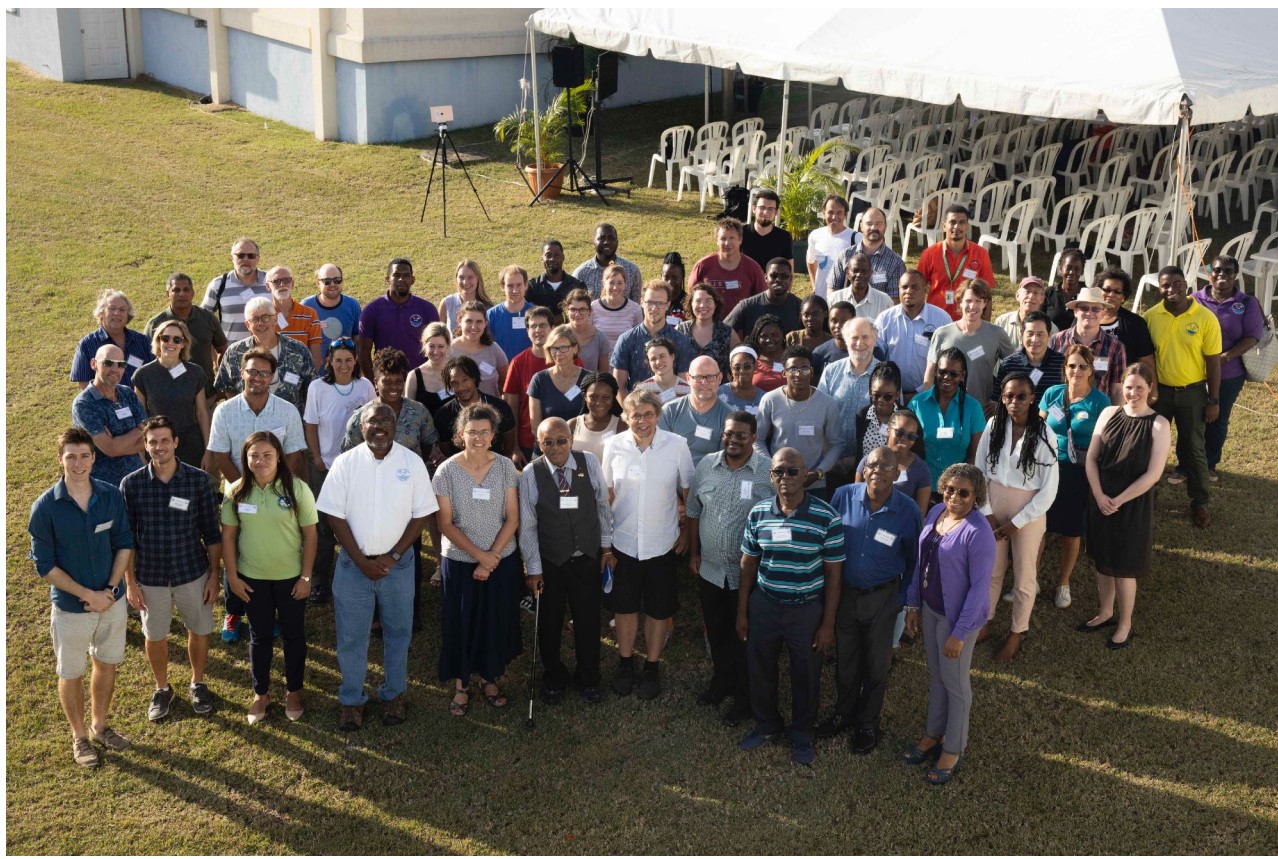

**Figure 21.** Group photo showing participants in the symposium "From BOMEX to EUREC⁴A" (Photo by F. Batier).

### 3.7.3 Scientific outreach in schools and facility visits

Scientific outreach activities, such as school visits and scientific open houses, sought to sensitize the public, in particular school children, to the EUREC⁴A programme and its important role in addressing issues of severe weather and climate change. Together with the CIMH, and the Barbados Ministry of Education, ten visits to primary and secondary schools in Barbados were arranged. Simple experiments were designed and performed with the children to help build intuition as to the underlying atmospheric and ocean processes relevant to EUREC⁴A's scientific objectives, not to mention the weather phenomena that

surrounds them on a daily basis. These outreach efforts aimed to raise awareness in ways that would increase the resilience of
the region to weather and climate extremes, to support citizen science, and to expose young people to scientific career paths.

The open houses, consisted of guided tours of many of the measurement platforms. This included: tours of the research
ships; visits to the BCO at Deebles Point, St. Philip (where visitors could help launch radiosondes); the aerosol measurement
facility on nearby Ragged Point; Boreal, Skywalker and CU-RAAVEN drone launches at Morgan Lewis, St. Andrew; and tours
of the POLDIRAD radar at Collecton, St. John. As an informal complement to the symposium, the outreach activities provided
a window into the daily life of the campaign and gathered a diverse audience, from local Barbadians to the scientists involved
in EUREC⁴A. The success of EUREC⁴A's outreach efforts are perhaps best exemplified by the ad hoc team of young engineers
(Fig. 22) that helped flight proof the drones before their launch from Morgan Lewis.

Further activities included members of the scientific team planting trees as part of the "Barbados We Planting" initiative.
In parallel to the scientific campaign, two French filmmakers also visited Barbados for the duration of the EUREC⁴A field
campaign to shoot a documentary combining scientific, cultural and historical elements of the island of Barbados. From their
material an additional short scientific documentary of the campaign was created. It is provided as an electronic supplement to
this manuscript.

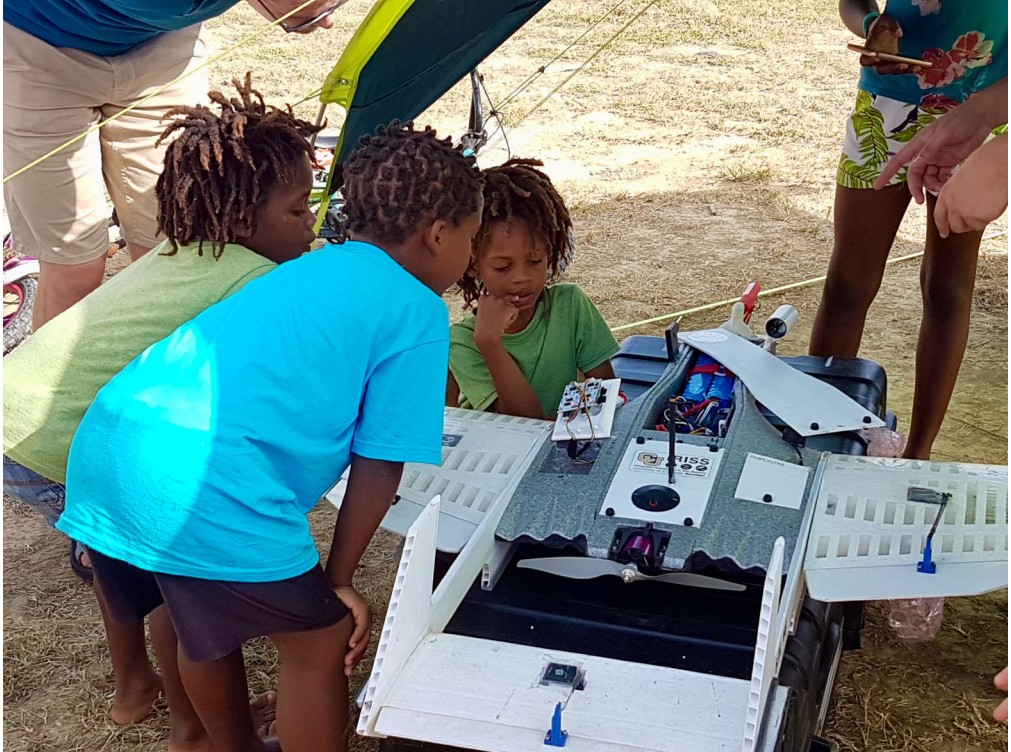

**Figure 22.** Local children helping to evaluate air-worthiness of CU-RAAVEN UAS prior to launch from Morgan Lewis Beach (Photo By
S. Bony).





## 4  Scientific Practice

EUREC⁴A advanced a culture of open and collaborative use of data. It did so by initiating a series of discussions, starting well before the field campaign and culminating in a document outlining principles of good scientific practice. In arriving at these principles emphasis was placed on understanding the differing cultural contexts in which data is collected. For instance, the degree to which measurements are made by individual investigators, or made for investigators by institutions, were often colored by different national practice. Differences in how measurements are made drive differences in expectations as to how the resultant should be made available and used, and thus reflect this national coloring. EUREC⁴A defined 'Good Scientific Practice' in terms of four principles, summarized below:

1. To actively support the initial dispersal of data by making (even preliminary) data available to *everyone* as quickly as possible through the AERIS archive.

2. To publish finalized data in ways that ensure open and long-term availability and bestow appropriate credit on those who collected it.

3. To actively attempt to meaningfully involve those who collected data in its analysis at the early stages of its use.

4. To provide clear timely and unprompted feedback on the use of the data, both by the analysis community for the instrument groups and vice versa.

In addition, examples of 'bad practice' were outlined. For example, using someone's data to write a paper and then sending the paper to the data-providers as it is about to be submitted and offering authorship was deemed bad practice. Or by assigning co-authorship on the basis of someone's status rather than through substantive contributions. 'Good practice' would have been to intellectually involve the data provider at an early stage of the study. 'Good practice' also recognized the importance of providing intellectual space for young scientists to independently develop their ideas, and have the time to appreciate and savour the low hanging fruit to be found along their path of research.

Authorship of the present paper recognizes all technical/scientific contributions to the data collection. The ways in which each author contributed to EUREC⁴A are summarized in an electronic supplement to this paper.

### 4.1  Data

The data collected during EUREC⁴A will, in different stages of development, be uploaded and archived on the AERIS datacenter. AERIS datacenter is part of the French Data Terra Research Infrastructure, it has the objective to facilitate and enhance the use of atmospheric data, whether from satellite, aircraft, balloon, or ground observations, or from laboratory experiments. It generates advanced products and provides services to facilitate data use, to prepare campaigns, and to interface with modelling activities.

In addition, emphasis is being placed on the publishing of datasets through a special collection of articles in *Earth System Science Data*. Many of these data papers will involve the construction of cross-platform datasets, for instance for the upper-air



network, or dropsondes, isotope measurements, or classes of remote sensors. At the end of the data collection phase, all data on AERIS will be mirrored by the Caribbean Institute for Meteorology and Hydrology in Barbados.

## 4.2  Environmental Impact

EUREC$^4$A was motivated by an interest to better anticipate how Earth's climate will change with warming. This makes it all the more relevant to ask how EUREC$^4$A exacerbates the problems it attempts to understand, or more pertinently, whether
it made appropriate use of scarce resources. The first step in answering this question is to estimate the magnitude of its environmental impact. We do so here mostly in terms of EUREC$^4$A's carbon footprint, which we estimate (see A for details) to be $5 \times 10^3$ t$CO_2$. The marginal increase – EUREC$^4$A took place at the expense of other campaigns – is of course much less. The main contribution to the carbon footprint was from fossil fuels (kerosene and diesel) used to power the research platforms. The travel of the participating scientists contributed non-negligibly ($5\,\%$) to EUREC$^4$A's carbon footprint, and provides context
for the total emissions.

   When learning about EUREC$^4$A many people become concerned about the environmental impact of the dropsondes – a concern shared by some of the present authors. The sondes have been designed to sink to the ocean floor after descending to the sea-surface. As elaborated upon in A, this along with the choice of materials (including batteries), and their small size (which with planned modifications may be reduced by a further factor of two), results in the environmental impact of the use
of even a very large number of sondes themselves being minimal. One hesitates to call any environmental impact negligible, but compared to many of the other activities – let alone the initial emotional response to the idea of throwing objects out of an aircraft – this is probably an apt description of the environmental impact.

   The potential of using yet smaller sondes, or smaller platforms in general, to further reduce environmental impacts was vividly illustrated by EUREC$^4$A's extensive use of small autonomous vehicles. In many cases these provided more agile and
less energy intensive ways of sampling the environment. For instance, the Saildrones, Wave Gliders and AutoNaut (Caravela) many make use of renewable energy sources for their propulsion (e.g., wind for Saildrones, waves for the Wave Gliders and AutoNaut), and for their scientific sensors (solar panels).

   Validation for EUREC$^4$A's use of the resources was experienced not just through the data collected, but also through the social interactions that the campaign enabled. This, as discussed in § 3.7, was expressed in bonds that were established and
the many opportunities that were presented through the numerous outreach and capacity building activities (Fig. 22). Further validation of EUREC$^4$A's use of resources depends on how the gained data advances scientific understanding to help humanity. This ultimately depends on what is done with the data; something over which we, the authors, have considerable influence and responsibility. We very much hope this realization will motivate a determination to learn as much as possible from the EUREC$^4$A measurements, and that the importance of supporting such efforts is recognized by funding agencies. We also
acknowledge the imperative this creates to make the hard won data easy to access and use, e.g., through data papers, and to also communicate what we learned from our efforts, as widely and freely as possible.





# 5  Conclusions

Field studies are commonplace, and each – by virtue of taking a snapshot of nature at a given point in time and space – is unique and unprecedented. This is of itself not particularly remarkable. Field studies involving such a large number of investigators

and such a large degree of coordination as was the case in EUREC[4]A are uncommon, but also this represents little more than an organizational achievement. Moreover, many of the questions EUREC[4]A attempted to address have been the focus of past field studies. For instance, air-sea interaction was at the heart of the original Barbados field study, BOMEX (Holland and Rasmusson, 1973). Likewise a great number of studies, most recently the Convective Precipitation Experiment (Leon et al., 2016) and Rain in Cumulus over the Ocean (RICO Rauber et al., 2007), had warm rain formation processes as a central focus.

The influence of boundary layer processes on cloud formation was already extensively studied by Malkus (1958), and again more recently by Albrecht et al. (2019). Field studies to measure aerosol-cloud interactions are myriad and include very large international efforts such as the Indian Ocean Experiment (Ramanathan et al., 2001). And with new insights from modelling, an increasing number of studies have begun to focus on ocean meso and submesoscale dynamics (Shcherbina et al., 2013; Buckingham et al., 2016; D'Asaro et al., 2018). What made EUREC[4]A an advance was neither its size, nor many of its specific

questions, rather its ability to quantify a single process – the link between circulation and cloudiness. Doing so opened the door to characterizing the totality of processes believed to influence the structure of the lower atmosphere and upper ocean in the region of the trades, and this is what made EUREC[4]A special.

The execution of EUREC[4]A was successful. All of the measurements we set out to make have been made. For some key quantities, such as the mean mesoscale vertical motion field, preliminary analyses (e.g., Fig. 9) suggest that the measurements

sampled substantial variability, which bodes well for testing hypothesized link between cloudiness and cloud based mass fluxes. The analysis of other measurements, such as those that aim to quantify clouds, is more delicate and ongoing. We anticipate that each step of the subsequent analysis of the EUREC[4]A data will teach us a great deal more about the ways of clouds, how they couple to circulation systems on different scales, how they influence and are influenced by the upper ocean, the extent to which they are susceptible to perturbations in the aerosol environment, and how precipitation links processes across scales. At

the very least a better quantification of these sensitivities should help us understand to what extent a warmer world will express the majesty of the clouds in the trades less markedly.



## Appendix A:  Estimates of environmental impact

With a specific density of $0.82\,\mathrm{kg\,L^{-1}}$ we can compute the total $CO_2$ emissions from aircraft operations as

$$629\,497\,\mathrm{L} \times 3\,\mathrm{kg_{CO_2}/kg_{fuel}} \times 0.82\,\mathrm{kg_{fuel}/L} = 1\,548\,562\,\mathrm{kg}$$

$CO_2$ emissions, or roughly $1500\,\mathrm{t}$.

| Aircraft | Total Fuel / liters | Notes |
|---|---|---|
| ATR | 92 348 | Includes cross-Atlantic transit from Toulouse France. |
| HALO | 235 493 | Includes ferry from Oberpfaffenhofen Germany. |
| TO | 33 850 | Not including ferry from Panama. |
| P-3 | 267 806 | Assuming 20 hours of ferry and 95 flight hours. |
| Sum | 629 497 | |

**Table A1.** Fuel consumption from EUREC[4]A crewed aircraft. Some fuel burn rates were reported in lbs, and were converted to liters using a conversion of $1.76\,\mathrm{lbs/L}$

Estimates of the fuel consumption for the Research Vessels are larger as they operate around the clock and support the life of a community of scientists and their laboratories at sea. They are also more roughly estimated. We begin with numbers from the R/V Meteor, which burns sulfate-reduced diesel. Its burn rate is estimated as $5000\,\mathrm{L\,d^{-1}}$ when stationary, and as much as twice that much when under way. Given that the ships were generally steaming but with station work mixed in, we adopt a burn rate of $8000\,\mathrm{L\,d^{-1}}$. Diesel is denser than kerosene, and produces more $CO_2$ per kilogram. We adopt a conversion of $3.15\,\mathrm{kg}$ of $CO_2$ for every $\mathrm{kg}$ of diesel, and a density of $0.85\,\mathrm{kg\,L^{-1}}$. Based on this we estimated that the R/V Meteor burnt $21\,420\,\mathrm{kg}$ per day. The reported fuel use for the R/V Ron Brown was $79\,922\,\mathrm{gallons}$ or $363\,333\,\mathrm{L}$, which included the Ferry to and from a home port. This fuel burn corresponds to $972\,823\,\mathrm{kg_{CO_2}}$. Assuming 35 days of operations, this corresponds to a burn rate of $10\,380\,\mathrm{L\,d^{-1}}$. For our estimates we adopt the $8000\,\mathrm{L\,d^{-1}}$ burn rate for all the ships, and estimate five days of ship ferry-time, so for four ships each with $30\,\mathrm{d}$ of ship time we end up with a total emission of $3000\,\mathrm{t}$ $CO_2$, about twice the direct emissions from the research aircraft. The ship numbers are not offset by the reduced personal emissions of those on the ship, i.e., who don't need hotels, or rental cars, or the operation of their home labs, and often have reduced travel, but this is likely minor.

We estimated that 200 people traveled to EUREC[4]A. If each is further assumed to have flown $15\,000\,\mathrm{km}$ (about the round-trip distance from Frankfurt to Grantley Adams International Airport in Barbados) in economy class, then we can adopt an emission estimate of $75\,\mathrm{g\,km^{-1}}$ per passenger (from Atmosfair for a non-stop flight with an Airbus 340-500). This

$$200 \times 15\,000\,\mathrm{km} \times 75\,\mathrm{g_{CO_2}/km} = 225\,000\,\mathrm{kg}$$

$CO_2$ emissions. More modern aircraft have substantially reduced emissions ($60\,\mathrm{g\,km^{-1}}$), then again carbon-offsetting schemes often estimate a three-fold larger equivalent emission due to the inclusion of other factors.





The environmental impact of the sondes is informed by life-cycle analyses life-cycle analyses that Vaisala has commissioned for their radiosondes, as well as our own analysis. The life-cycle analysis identified "the production of the printed circuit board

and the electricity used during the assembly [as having] the most significant effect on the environmental impacts." But this analysis did not consider the impact of the waste, beyond issues of things like battery toxicity. In this regard the lithium batteries used by the sondes had the least environmental impact of all available choices i.e., alkaline, or water-activated batteries. We estimated that $1.2\,\mathrm{kg}$ of lithium were deposited with the sondes, in the ocean – which is roughly equivalent to what would be found naturally in the seawater displaced by one of the EUREC$^4$A research vessels. Plastic sensor casings, parachutes, and/or

remainders of the latex balloon, add an additional impact. This is minimized by designing the sondes to sink to the ocean floor. Efforts are ongoing to identify different materials to further reduce the environmental impact of the sondes. One also questions whether the potential energy loss by the sonde could be used to power the instrumentation, and whether a different and smaller sonde could forgo the use of a parachute.





## Appendix B: Platforms

**BCO**

The Barbados Cloud Observatory (BCO) is located at the far East of Barbados (13°09′45.8″N 59°25′43.8″W) has started operation in April 2010 with a growing set of different instruments for cloud observing and recording. During the whole EUREC⁴A-campaign the BCO was staffed for radiosonde launching and maintenance. Measurements at Ragged Point listed below were made by the University of Miami, the University of Manchester and the Max Planck Institute for Chemistry. DLR 700 and CIMH staff operated POLDIRAD at St John (13°10′49″N 59°29′47″W, altitude 240 m).

Table B1: BCO

| Instrument | Brief Description |
|---|---|
| Doppler cloud radar (CORAL) | Metek MIRA-35 vertically-staring pulsed 35 GHz Doppler cloud radar measuring radar reflectivity, linear depolarization ratio, Doppler velocity, spectral width |
| Rain radar (POLDIRAD) | A scanning polarized Doppler C-band radar measuring radar reflectivity, linear depolarization ratio, Doppler velocity, spectral width, differential reflectivity, copolar correlation coefficient, differential propagation phase |
| Rain Radar | Metek micro rain radar operating at 24 GHz to measure rainrate, liquid water content and drop velocity |
| Raman lidar (CORAL) | A vertically-staring backscatter lidar with attenuated backscatter at 355 nm, 387 nm, 407 nm, 532 nm and 1064 nm measuring particle backscatter at 355 nm, water vapor mixing ratio, temperature and humidity |
| Wind lidar | HALO Phononics scanning Doppler lidar measuring Doppler velocity, vertical and horizontal winds |
| Ceilometer | OTT CHM 15k pulsed laser cloud height detector at 1064 nm used to detect cloud-base height and lifting condensation level |
| Isotopic analyzer | Picarro (L2130-i)cavity ring-down laser spectrometer to measure water stable isotopologues |
| Radiometer (BCOHAT) | RPG HATPRO-G5 scanning radiometer at K and V band used for retrievals of columnar water vapor content, liquid water path and rainrate |
| Pyrgeometer | Kipp and Zonen CGR4 sensor measuring hemispheric broadband downwelling longwave (4.5 μm to 42 μm) radiative fluxes |
| Pyranometer | Kipp and Zonen CMP21 sensor measuring hemispheric broadband downwelling and diffuse shortwave (0.285 μm to 2.8 μm) radiative fluxes |
| Pyrheliometer | Kipp and Zonen CHP1 sensor measuring direct normal shortwave (0.2 μm to 4 μm) radiative irradiance |
| All-sky camera | Vertically operating visible (hemispheric) and infrared camera to infer cloud coverage |
| Disdrometer | Eigenbrodt ODM470 optical sensor measuring droplet size distributions |





Table B1: BCO, continued

| Instrument | Brief Description |
| --- | --- |
| Particle counter | Size resolved CCN measurements (CCN-100, DMT vs. CPC 5.412 - GRIMM) |
| " | CPC (TSI 3750) measuring total aerosol/condensation nuclei (CN) concentration ($d > 10\,\mathrm{nm}$) |
| Particle sizer | SMPS (GRIMM 5.420) measuring aerosol/CN size distribution ($10\,\mathrm{nm}$ to $1000\,\mathrm{nm}$) |
| " | WRAS (GRIMM, EDM 180) Aeasuring aerosol/CN size distribution ($0.25\,\mathrm{\mu m}$ to $32\,\mathrm{\mu m}$ |
| " | SMPS (TSI 3080) measuring aerosol/CN size distribution ($15\,\mathrm{nm}$ to $650\,\mathrm{nm}$) |
| " | APS (TSI 3321) measuring aerosol/CN size distribution ($0.5\,\mathrm{\mu m}$ to $20\,\mathrm{\mu m}$) |
| " | PAS (GRIMM 1.109) measuring aerosol/CN size distribution ($0.25\,\mathrm{\mu m}$ to $32\,\mathrm{\mu m}$) |
| Aerosol mass spectrometer | AeroMegt LAAP-ToF mass spectrometer measuring single particle aerosol composition and mixing state |
| " | Aerodyne AMS, bulk chemical composition of non-refractory aerosol ($d < 1\,\mathrm{\mu m}$) |
| Bioaerosol sensor | WIde-Band integrated Bioaerosol Sensor (WIBS-4M) measuring aerosol size distribution, three-channel fluorescence and particle asymmetry |
| Aerosol filter measurements | Mass concentrations of dust and soluable ions collected daily using high volume sampling |
| Aerosol composition | Aerosol collection for offline analysis of mixing state and size-resolved composition |
| Anemometer | Gill R350 sonic anemometer to measure wind speed and wind direction |
| Weather sensors | Metpack (Vaisala WXT-510) to measure temperature, relative humidity, pressure, precipitation |
| " | Vaisala WXT-530 weather sensors measuring temperature, humidity, pressure, wind speed, wind direction, rain |
| GNSS | Dual TRIMBLE NetR9 GNSS receivers with ZEPHYR GEODETIC2 antennas for columnar water vapor content |
| Radiosondes | Vaisala RS41-SGP radiosondes measuring atmospheric profiles (during ascent and descent) of temperature, relative humidity, pressure, wind direction and wind speed |

*BCO*

**NTAS**

The Woods Hole Northwest Tropical Atlantic Station (NTAS) is a surface mooring maintained at approximately 15° N, 51° W since 2001 by means of annual mooring "turnaround", i.e., deployment of a refurbished mooring and recovery of the old mooring. The refurbished mooring has freshly calibrated sensors and is deployed first. A one to two-day period of overlap
before recovering the old mooring provides intercomparison data and allows consecutive data records to be merged. Meteorological variables suitable for estimation of air-sea fluxes from bulk formulas, as well as upper ocean variables and deep ocean temperature and salinity are measured. Data are available from the Upper Ocean Processes Group at Woods Hole and from OceanSITES.



Table B2: NTAS

| Instrument | Brief Description |
|---|---|
| ASIMET | Air-Sea Interaction METeorology system measuring air temperature, relative humidity, barometric pressure, precipitation, wind speed, wind direction, longwave broadband irradiance, shortwave broadband irradiance, sea-surface temperature, sea-surface salinity every 1 min (on surface buoy, sensors at $\sim$3 m height). |
| Weather sensors | Vaisala WXT-520 measuring air temperature, relative humidity, barometric pressure, precipitation, wind speed, wind direction (sensor at $\sim$3 m height) |
| Temperature sensor | Seabird SBE-39 adapted to measure air temperature every 5 min (sensor at $\sim$3 m height) |
| ” | Seabird SBE-56 fast-sampling temperature recorder measuring sea-surface temperature every 1 min (in buoy hull) |
| ” | Seabird SBE-39 recording seawater temperature every 5 min (on mooring line at 5, 15, 20, 30, 35, 45, 50, 60, 65, 75, 80, 90, 100, 110 m) |
| ” | Star-Oddi Starmon measuring seawater temperature every 10 min (on mooring line at 110-160 m, 10 m intervals) |
| Wave height sensor | Xeos Brizo measuring surface wave height and period (every 60 min, with a 20 min sample interval each hour) |
| Acoustic Doppler current meter | Nortek Aquadopp ADCM recording horizontal velocity every 20 min (on mooring line at 5.7 and 13 m) |
| Current profiler | Nortek Aquapro profiler measuring horizontal velocity profile every 60 min (on mooring line at 24 m, uplooking) |
| Acoustic Doppler current profiler | Teledyne RDI ADCP measuring horizontal velocity profile every 60 min (on mooring line at 85 m, uplooking) |
| CT sensor | Seabird SBE-37 measuring seawater temperature and conductivity/salinity every 10 min (on mooring line at 10, 25, 40, 55 and 70 m) |
| ” | Seabird SBE-37 measuring deep ocean temperature and salinity every 5 min ($\sim$38 m above bottom) |

*NTAS*

## ATR

The French ATR-42 aircraft, operated by SAFIRE (a national research infrastructure of Meteo-France/CNRS/CNES), is a turbo-prop aircraft that has the capability of flying in the lower troposphere (ceiling at about 8 km). During EUREC[4]A it flew 19 missions on 11 days, from January 25 to February 13 2020, totaling 91 flight hours. It generally flew two missions per day. Each mission was about 5 hour long, including a transit time from the airport to the EUREC[4]A circle of 20 min in each direction. Most missions were composed of a transit leg to the circle flown at an altitude of about 2.5 km, two or three box





patterns (rectangles of $15\,\mathrm{km}$ by $120\,\mathrm{km}$) near cloud-base, two L-patterns within the subcloud-layer (near the top and the middle of the layer, cross-wind and along-wind) and a surface leg (altitude of $60\,\mathrm{m}$) before a transit back to the airport at an altitude of about $4.5\,\mathrm{km}$.

Table B3: ATR

| Instrument | Brief Description |
|---|---|
| Backscatter lidar (ALIAS) | An horizontally-staring backscatter lidar operating at $355\,\mathrm{nm}$ and detecting polarization (Chazette et al., 2021) |
| Doppler cloud radar (BASTA) | An horizontally-staring bistatic FMCW $95\,\mathrm{GHz}$ Doppler cloud radar characterizing clouds, rain, horizontal Doppler velocity (Delanoë et al., 2016) |
| Doppler cloud radar (RASTA) | An upward-looking $95\,\mathrm{GHz}$ Doppler pulsed cloud radar characterizing clouds, rain, vertical Doppler velocity and Doppler spectrum |
| Isotopic analyzer | Picarro (L2130-i) cavity ring-down laser spectrometer measuring stable isotopologues in water vapor (including water vapor mixing ratio) |
| Humidity sensor | WVSS2 absolute humidity sensor (tunable diode laser absorption) |
| " | Licor-7500 hygrometer |
| " | Campbell krypton hygrometer KH-20 measuring humidity fluctuations |
| " | 1011 C Buck dew point hygrometer |
| " | Enviscope capacitive sensor measuring relative humidity |
| Visible cameras | High-resolution visible cameras (AV GT1920C) looking sideways and downward |
| Particle sizer | UHSAS, measuring aerosol particle sizes from $0.06\,\mu\mathrm{m}$ to $1\,\mu\mathrm{m}$ |
| Scattering spectrometer | FSSP 300, measuring particle sizes from $0.3\,\mu\mathrm{m}$ to $20\,\mu\mathrm{m}$ |
| Cloud droplet probe | CDP/FCDP, measuring particle sizes from $2\,\mu\mathrm{m}$ to $50\,\mu\mathrm{m}$ |
| Particle imager | 2D-S Stereo probe measuring particle sizes from $10\,\mu\mathrm{m}$ to $2000\,\mu\mathrm{m}$ |
| Liquid water sensor | LWC-300 hot-wire liquid water sensor to measure liquid water content |
| Pyrgeometer | Kipp and Zonen sensor measuring hemispheric broadband upwelling and downwelling longwave ($4.5\,\mu\mathrm{m}$ to $42\,\mu\mathrm{m}$) radiation |
| Pyranometer | Kipp and Zonen sensor measuring hemispheric broadband upwelling and downwelling shortwave ($0.2\,\mu\mathrm{m}$ to $3.6\,\mu\mathrm{m}$) radiation |
| Infrared radiometer | CLIMAT CE332, a downward staring infrared radiometer measuring irradiance at $8.7\,\mu\mathrm{m}$, $10.8\,\mu\mathrm{m}$ and $12.0\,\mu\mathrm{m}$ and used to infer SST |
| Core instrument. | The aircraft and in-situ data system (aircraft state, temperature, humidity and pressure). |

*ATR*





## HALO

The German research aircraft HALO (High-Altitude Long-Range Research Aircraft) is operated by DLR in the configuration described by Stevens et al. (2019a), with instruments developed, certified and operated by groups from around Germany. During EUREC[4]A it was stationed at the Grantley Adams International Airport (TBPB) on Barbados. It flew 13 missions out of GAIA plus 2 transfer flights from and to Germany, totalling in 130 flight hours. Its first local flight was on January 22 and last on February 15, 2020. HALO is a modified twin-engine business jet (Gulfstream 550) with an endurance of over $8000\,\mathrm{km}$ and a ceiling of $15.5\,\mathrm{km}$. Most flights were spent around $10\,\mathrm{km}$ altitude with a flight speed of about $190\,\mathrm{m/s}$ with a typical flight pattern of two sets of 3.5 circles (each lasting about an hour) on the EUREC[4]A circle separated by a 1 hour excursion toward the NTAS buoy.

Table B4: HALO

| Instrument | Brief Description |
| --- | --- |
| Cloud radar (HAMP) | A Downward-looking high-powered Ka-band polarized Doppler cloud radar |
| Microwave radiometers (HAMP) | An ensemble of microwave radiometers in K, V, W, F and G bands to estimate cloud properties, precipitation and integrated water vapor amount |
| DIAL lidar (WALES) | A differential absorption lidar near $935\,\mathrm{nm}$, with additional channels at $532\,\mathrm{nm}$ and $1064\,\mathrm{nm}$, to characterize aerosols, clouds and water vapor profiles |
| Hypersp. imager (specMACS) | A VNIR and a SWIR hyperspectral camera ($0.4\,\mathrm{\mu m}$ to $2.5\,\mathrm{\mu m}$ ) and two polarization cameras to characterize clouds, droplets and optical thickness (Ewald et al., 2016) |
| Solar radiometer (SMART) | A spectral solar radiometer measuring visible and near-infrared radiation to infer clouds presence and microphysics |
| Infrared imager (VELOX) | A spectral IR imager to characterize clouds and measure liquid water path and SST |
| Infrared thermometer (V-KT 19) | An infrared thermometer channel for the VELOX to measure SST |
| Broadband radiom. (BACCARDI) | A broadband radiometer for downwelling and upwelling irradiances in the LW and the SW. |
| Dropsondes | AVAPS dropsonde (Vaisala RD-41) system to measure temperature, humidity and wind |
| Core instrument. (BAHAMAS) | The aircraft and in-situ data system (aircraft state, temperature, humidity and pressure). |

*HALO*

## P-3

The NOAA WP-3D Orion (P-3) "Miss Piggy" was operated by NOAA's Aircraft Operations Center. During EUREC[4]A it flew 11 missions on 11 days totaling 95 flight hours. Its first flight was January 17 and last February 11, 2020. There were three night flights on February 9, 10 and 11, taking off between 22:00 and 23:30 local time on Feb 8, 9, and 10 respectively. The P-3, one of two "Hurricane Hunters", is a four engine turboprop, and operates with a maximum endurance of 10 flight hours and a maximum ceiling of $8.75\,\mathrm{km}$. Flight strategies varied over the course of the experiment but included circles at or above $7\,\mathrm{km}$ to





deploy dropsondes, slow profiles from 7 km to 500 ft/150 m for water vapor sampling, stacked straight and level legs for *in situ*
sampling cloud the sub-cloud and cloud layers, and "lawnmower" patterns at 2.75 km to 3.00 km for deployment of Airborne
eXpendable BathyThermographs and remote sensing of ocean surface state. Flight-level data includes aircraft navigation and
orientation and standard meteororological variables.

Table B5: P-3

| Instrument | Brief Description |
| --- | --- |
| Doppler cloud radar | Downward-looking PSL W-band (94 GHz) Doppler pulsed cloud radar to characterize clouds, precipitation and the ocean surface wave state |
| Isotope analyzer | Picarro (L2130-i) cavity ring-down laser spectrometer measuring stable isotopologues in water vapor (including water vapor mixing ratio) |
| Microwave radiometer | Stepped frequency microwave radiometer ProSensing Inc, measuring in six C-band frequencies to characterize surface wind speed and rain rate |
| Wave radar | Wide Swath Radar Altimeter ProSensing Inc, 80 beams to 30°, 16 GHz (Ku band) to characterize the ocean surface wave state |
| Infrared radiometer | Heitronics KT19.85, measuring infrared brightness temperature in the atmospheric window (9.6 μm to 11.5 μm) |
| Particle sizer | Measuring particle sizes from 0.5 μm to 50 μm |
| Cloud droplet probe | Measuring cloud droplets size from 2 μm to 50 μm |
| Particle imager | Measuring cloud and rain drops from 12.5 μm to 1550 μm |
| Rain imager | Measuring precipitation particles sizes from 0.1 mm to 6.2 mm |
| Core instrument. | The aircraft and in-situ data system (aircraft state, temperature, humidity and pressure). |

*P-3*

## TO

The Twin Otter (TO) was operated by the British Antarctic Survey (BAS). It flew 25 missions on 15 days totalling 90 flight
hours. The first flight was on 24 January and the last on 15 February, 2020. All flights were during daylight hours. The main
objective of the TO flights was to observe the subcloud and cloud layers at a number of altitudes from close to the sea surface
to the level of the detrainment region and above. As many clouds as possible were sampled by zig-zagging to catch the clouds.

Table B6: TO

| Instrument | Brief Description |
| --- | --- |
| Hygrometer | Buck 1011c Cooled-mirror hygrometer measuring humidity at 1Hz |
| Temperature sensor | Non de-iced and de-iced Rosemount probes measuring temperature at 0.7 Hz |
| Pyrgeometer | Eppley pyrgeometer measuring longwave irradiance 4-50 μm |





Table B6: TO, continued

| Instrument | Brief Description |
| --- | --- |
| Pyranometer | Eppley pyranometer measuring shortwave irradiance 0.295-2.8 $\mu$m |
| Infrared thermometer | Heimann IR thermometer measuring temperature at 10 Hz |
| Camera | Forward camera mounted in the cockpit |
| Turbulence probe | Best Aircraft Turbulence (BAT) probe measuring air motions and high-frequency wind (50 Hz) |
| Ultra-fast thermometer | Ultra-Fast Thermometer 2 (UFT2, developed from UFT-M) measuring temperature fluctuations with a sampling frequency of 20 kHz |
| Particle counter | TSI 3772 CPC measuring total ultrafine particle concentration (D > 10 nm) |
| Particle sizer | GRIMM Sky-OPC aerosol spectrometer measuring particle sizes from 0.3 to 10 $\mu$m |
| Particle sizer | TSI SMPS 3938 and 3775 measuring aerosol particle sizes from 25 nm to 1 $\mu$m |
| Particle counter | DMT PCASP measuring aerosol particle sizes from 0.1 to 3 $\mu$m |
| " | CCN-100 measuring cloud condensation nuclei as a function of supersaturation |
| Cloud droplet probe | DMT CDP-100 measuring cloud droplet sizes from 2 to 50 $\mu$m |
| Scattering spectrometer | SPEC FFSSP measuring cloud droplet sizes from 1.5 to 50 $\mu$m |
| Particle imager | SPEC 2D-S measuring cloud particle sizes from 10 to 1280 $\mu$m |
| Rain spectrometer | SPEC HVPS-3 measuring precipitation particles sizes from 150 $\mu$m to 19.2 mm |
| Core instrument. | The aircraft and in-situ data system (aircraft state, temperature, humidity and pressure). |

*TO*

## Atalante

The R/V L'Atalante (Atalante) belongs to the French oceanographic research fleet national infrastructure and is operated by IFREMER. This 85 m long vessel is the first modern vessel of the French open-ocean fleet. The R/V Atalante ship time has

745 been provided by the French operator to the EUREC[4]A _OA project within the EUREC[4]A umbrella. It sailed from Pointe-Pitre, Guadaloupe on January 20 and started its operations in Barbados water on January 21, 2020. It navigated for more than 3000 NM collecting ocean and atmosphere data from 6°N to 15°N and 60°W to 52°W surveying the Tradewind Alley and the North Brazil Current eddy corridor (Boulevard des Tourbillons) in international waters and in the ZEEs of Barbados, Trinidad and Tobago, Guyana, Suriname, and French Guyana. During the cruise, an underwater electric glider (Kraken) and surface drifters

(including the Ocarina and Piccolo platforms) were deployed (see respective subsubsections).



Table B7: Atalante

| Instrument | Brief Description |
|---|---|
| Underway data | DSHIP navigational data, base meteorology data, bathymetry: UTC-time, lat, lon, heading, heave, pitch, roll, pressure, air temperature, water temperature dew point, relative humidity, solar broadband radiation, infrared broadband radiation, UV radiation, visibility, wind-direction, wind-speed, relative wind-direction, relative windspeed |
| Radiosondes | Vaisala RS41-SGP radiosondes measuring atmospheric profiles (during ascent and descent) of temperature, relative humidity, pressure, wind direction and wind speed |
| Radiosondes | MeteoModem M10 radiosondes measuring atmospheric profiles (during ascent only) of temperature, relative humidity, wind direction and wind speed |
| Ceilometer | Lufft CHM 8K ceilometer measuring profiles of attenuated backscatter at 905 nm |
| Cloud cameras | Upward all-sky visible images and upward 30° field of view thermal images, both at 0.1 Hz |
| Sun photometer | MICROTOPS sunphotometer capturing at sun-views the solar attenuation at 380, 440, 670, 870 and 940 nm to derive column properties of aerosol optical depth (AOD) at these wavelengths and the atmospheric water vapor content |
| Atmospheric mast | A meteorological station WXT-500, a 3D wind anemometer Gill HS-50, a CNR4 radiometer for direct and indirect radiation, an inertial station to filter the ship motion, a refractometer for humidity, a LICOR LI-7500DS for fast $CO_2$ and humidity measurements, fast measurements of wind, temperature, humidity and pressure, two GPS antennas and a motion pack |
| Isotope analyzer | Picarro (L2140-i7) cavity ring-down laser spectrometer measuring stable isotopologues in water vapor (including water vapor mixing ratio) |
| Isotope sampler | Sampling system for measuring stable isotopologues in rain and seawater |
| Pyranometer | SPN1 Delta-T SNC radiometer measuring global and diffuse radiation |
| Hypersp. radiometer | ROX JBC hyperspectral radiometer measuring global, reflected and diffuse spectral radiation from 350 to 950 nm |
| GNSS | Global Navigational Satellite System (GNSS) antennas and receivers for measuring underway columnar water vapor content |
| Aethalometer | Aethalometer AE33 Magee Scientific measuring aerosol attenuation at seven wavelenghts 370 nm, 470 nm, 520 nm, 590 nm, 660 nm, 880 nm and 950 nm and used to infer black carbon concentrations |
| Particle sizer | NanoSCAN TSI inc measuring particle number concentrations from 10 nm to 400 nm |
| Particle counter | GRIMM 1.107 optical particle counter measuring number concentrations from 0.25 μm to 32 μm |
| Volume sampler | ECHO-PUF TCR Tecora measuring particle mass concentrations and chemical composition |
| Thermosalinograph | SBE-21 and SBE38 thermosalinographs measuring near-surface seawater temperature, conductivity and fluorescence (to infer salinity and chlorophyll) |
| $pCO_2$ | LICOR LI-7000 measuring system for $pCO_2$ and $fCO_2$ |





Table B7: Atalante, continued

| Instrument | Brief Description |
|---|---|
| CTD casts | SBE-911 Plus CTD for conductivity, temperature and pressure measurements, a SBE-43 dissolved oxygen sensor, a Chelsea Aquatracka III fluorimeter for in situ detection of Chlorophyll-a, a Wetlabs Cstar transmissometer for underwater measurements of beam transmittance and a Tritech PA500 altimeter, a QCP2350 (Cosinus collector) and a PAR sensor. |
| Moving vessel profiler | A MVP30-300 AML Oceanographic measuring water column temperature, salinity, pressure and high-frequency vertical profiles of seawater temperature and salinity within the upper (0 m to 200 m) |
| uTCD | Teledyne underway Conductivity Temperature Depth is a profiler device measuring vertical profiles within the upper ocean (0 - 400 m) of salinity (through conductivity), temperature (through a thermistor) and depth (through pressure), deployed. It is deployed over a moving vessel. It has the convenience of an expendable device like the XBT, but it is recoverable and reusable |
| Microstructure sonde | Rockland Scientific VMP-500 measuring vertical profiles of microstructure and dissipation-scale turbulence in the upper 300m of the ocean |
| Acoustic Doppler current profiler | Teledyne RDI 38 kHz and 150 kHz acoustic Doppler current profilers measuring subsurface (30 m to 1000 m and 30 m to 150 m, respectively) water velocity |
| " | Teledyne RDI 300 kHz duo acoustic Doppler current profiler measuring full water column water velocity |
| XBTs | XBT-T7 SIPPICAN measuring vertical profiles of seawater temperature within the upper ocean (0 m to 800 m). |
| XCTDs | XCTD-2 SIPPICAN measuring vertical profiles of seawater temperature and conductivity (salinity) within 0 m to 1000 m. |
| Argo floats | CTO-NKE Argo Floats measuring vertical profiles of seawater temperature, salinity and dissolved oxygen within the upper-ocean (initially 0-1000m, parking depth 200 dbar, daily; after 0-2000, parking depth 1000 dbar, every 10 days), |

*L'Atalante*

**MS-Merian**

The R/V Maria Sybille Merian is a (95 m) long German research vessel, owned and funded jointly by the German Science Foundation (DFG) and the German Ministry for Education and Science (BMBF). The MSM89 cruise started (January 17, 2020) and ended (February 20, 2020) in Bridgetown, Barbados (Karstensen et al., 2020). During the cruise various ocean and
atmosphere profile measurements were carried out from the ship. In addition, underwater electric gliders and the Cloudkite were deployed but are described separately (see respective subsubsections).



Table B8: MS-Merian

| Instrument | Brief Description |
| --- | --- |
| CTD casts | Seabird SBE-911 CTD mounted on a rosette system, operated up to full water depth, measuring vertical profiles of seawater temperature, salinity, fluorescence, oxygen and PAR |
| Rosette bottle sampler | Samples used for ocean-microbiology observations, including biogeochemical parameters (e.g. nutrients, chlorophyll a, particulate organic carbon and nitrogen, trace greenhouse gases), biogeochemical process rates (e.g. primary production), and molecular analyses (e.g. microbial community structure) |
| Cloud cameras | Upward all-sky visible images and upward 30° field of view thermal images, both at 0.1 Hz |
| Underwater Vision Profiler | The Underwater Vision Profiler UVP5 is an underwater camera system mounted on the CTD rosette that can take images of particles and plankton down to 6000 m water depth |
| UV spectral sensor | The TriOS OPUS spectral sensor was mounted on the CTD rosette to acquire full depth profiles of nitrate and nitrite |
| Acoustic Doppler current profiler | Teledyne RDI 300kHz lowered ADCP mounted on the CTD rosette to survey full depth ocean currents and acoustic backscatter |
| ” | Teledyne RDI ADCP 38 kHz, 75kHz were mounted in the hull of the ship (75kHz) and in the ship moon pool (38kHz) to provide water currents and backscatter information from close to surface to more than 1000 m depth. Surveyed in an underway mode |
| Thermosalinograph | SeaBird SBE-45/SBE-38 measuring temperature and salinity underway from about 6.5 m water depth |
| Microstructure sonde | Sea &Sun Technology Microstructure probe MSS 90 measuring the micro-scale temperature, pressure, conductivity and shear to determine the micro-scale water stratification and the strength of small scale turbulence in the water column |
| FerryBox system | OceanPack FerryBox underway system that acquires $pCO_2$ data from 6.5 m water depth |
| Radiosondes | Vaisala RS41-SGP radiosondes measuring atmospheric profiles (during ascent and descent) of temperature, relative humidity, pressure, wind direction and wind speed |
| Moving vessel profiler | An AML Oceanographic MVP30-350 measuring vertical profiles of temperature, salinity, pressure and fluorescence in the upper ocean, operated as underway system at selected segments |
| uCTD | Teledyne Oceanscience underway CTD measuring vertical profiles of temperature, pressure, salinity in the upper ocean, operated as underway system at selected cruise segments |
| Wave radar | A 9.4 GHz X-band wave radar measuring both phase and intensity of the radar backscatter and Doppler speed (sampled up to a maximum range of about 3 km around the ship), to determine surface currents, wind field, and surface signatures of internal waves. |
| Sun photometer | MICROTOPS sunphotometer capturing at sun-views the solar attenuation at 380, 440, 670, 870 and 940 nm to derive column properties of aerosol optical depth (AOD) at these wavelengths and the atmospheric water vapor content |



Table B8: MS-Merian, continued

| Instrument | Brief Description |
|---|---|
| Weather sensors | German Weather Service (DWD) automatic weather station measuring underway wind speed, wind direction, temperature (air, ocean), humidity and barometric pressure |
| GNSS | Global Navigational Satellite System (GNSS) antennas and receivers for measuring underway columnar water vapor content |
| Particle sizer | SMPS (TSI 3080) measuring aerosol/CN size distribution (10 nm to 420 nm) |
| Micro Rain Radar | A 24 GHz meteorological radar profiler for Doppler spectra of hydrometeors and multiple parameters related to rain |
| Cloud radar | RPG FMCW94 cloud radar to characterize the atmospheric state and clouds (incl. precipitation, and turbulent structure of the atmospheric boundary layer) through reflectivity, Doppler velocity, spectral width and skewness |
| Raman lidar (ARTHUS) | University of Hohenheim Raman lidar at 355 nm providing underway vertical profiles of temperature, water-vapor mixing ratio, aerosol particle backscatter and extinction coefficients at up to 10 s and 7.5 m and range resolution, respectively |
| Wind lidar | Doppler lidar (1 s temporal- and 30 m range-resolution) for the profiling of vertical wind and particle backscatter coefficient as well as boundary layer depth and cloud base height |
| ” | Scanning Doppler lidar operated in the 6-beam staring mode for the profiling of horizontal wind, TKE, and momentum flux. Horizontal wind profiles are delivered with 1 min as well as the TKE and momentum flux profiles with 30 min to 60 min resolution. |
| Ocean microbiology | Measurements in the upper water column of biogeochemical parameters (e.g. nutrients, chlorophyll A, particulate organic carbon and nitrogen, trace greenhouse gases), biogeochemical process rates (e.g. primary production) and molecular analyses (e.g. microbial community structure) |
| Underway data | Data acquired at high temporal resolution along the ship track: latitude, longitude, motion, course/heading (multiple), water depth, solar global radiation, Infrared radiation, PAR radiation, rain (vertical/lateral) |

*MS-Merian*

**Meteor**

The R/V Meteor is a 98 m long German research vessel, which is funded jointly by the German Science Foundation (DFG) and the German Ministry for Education and Research (BMBF). The M161 cruise started in Bridgetown (Barbados) on January 760 17 2020, and R/V Meteor spent the following month in the trade wind alley east of Barbados (between $12^oN$ to $15^oN$ and $54^oW$ to $60^oW$). During this time, the atmosphere above and the ocean underneath were repeatedly probed using a wealth of instruments that were either running continuously or sampled discrete stations. At the end of the campaign, R/V Meteor headed





for its next working area. Additional measurements were carried during this transit, and the vessel arrived in Ponta Delgada (Azores, Portugal) on March 3rd, 2020 after a total of 6250 NM of sailing (Rollo et al., 2020). The autonomous platforms
(such as CloudKite or ocean glider) deployed from the R/V Meteor are described in their respective subsubsections.

Table B9: Meteor

| Instrument | Brief Description |
| --- | --- |
| Underway data | Data acquired along the ship track: time, position (latitude, longitude), motion (heave, pitch, roll), atmospheric state (pressure, air temperature, water temperature, dew point, relative humidity), wind (wind-direction, wind-speed, relative wind-direction, relative windspeed), downward broadband solar radiation (200 - 3600 nm), downward broadband infrared radiation(4.5 - 42 $\mu$m), ocean currents at different depth (ADCP), surface water seasalt content and water temperature (thermosalinograph) |
| Raman lidar | Lidar providing vertical profiles of attenuated backscatter at 355, 532 and 1064 nm, volume depolarization ratio at 532 nm, particle linear depolarization ratio and vertical profiles of water vapor mixing ratio. |
| Cloud radar | 94GHz cloud radar operating on a balanced platform measuring radar reflectivity , linear depolarization ratio, Doppler velocity and spectral width |
| Microwave radiometer | HATPRO microwave radiometer measuring brightness temperatures in K and V Bands |
| Solar spectrometer | Measuring downward spectral solar irradiance between 300 and 2200 nm for droplet effective radius and liquid water path retrievals |
| Ceilometer | Jenoptic system measuring vertical profiles of attenuated backscatter at 1064 nm to infer cloud-base as a function of altitude and aerosol |
| Cloud cameras | Upward all-sky visible images and upward 30° field of view thermal images, both at 0.1 Hz |
| Sun photometer | MICROTOPS sunphotometer capturing at sun-views the solar attenuation at 380, 440, 670, 870 and 940 nm to derive column properties of aerosol optical depth (AOD) at these wavelengths and the atmospheric water vapor content |
| Solar spectrometer | MAX-DOAS (Multi-AXis Differential Optical Absorption Spectroscopy) measuring scattered sky solar radiances at different elevation angles to infer $NO_2$, $SO_2$, $SO_2$, $H_2O$, oxygen dimer (near the suface, vertical profiles and column data) |
| GNSS | Global Navigational Satellite System (GNSS) antennas and receivers for measuring underway columnar water vapor content |
| Particle sizer | WRAS (GRIMM, EDM 180) Measuring aerosol/CN size distribution ($0.25\,\mu$m to $32\,\mu$m |
| Photoacoustic extinctiometer | PAX photoacoustic nepholometer measuring the scattering of aerosol (size information) and the photo-acoustic absorption (black carbon concentration). |
| Air sampler | Locomotive air sampler measuring concentrations of larger (PM10) aerosol particles |
| Wind lidar | Windcube WLS70 wind-lidar measuring vertical wind profiles for the 100–2000 m range |
| " | Windcube V2 wind-lidar measuring vertical wind profiles for the 40–250 m range |





Table B9: Meteor, continued

| Instrument | Brief Description |
|---|---|
| Eddy covariance | A 4 m outrigger in front of the Meteor and one high up in the main mast close to the operational wind measurements providing high resolution (20Hz) measurements of vertical wind speed and gas concentrations to derive surface momentum, enthalpy and $CO_2$ fluxes. |
| Isotope analyzer | Picarro (L2140-i7) cavity ring-down laser spectrometer measuring stable isotopologues in water vapor (including water vapor mixing ratio) |
| Isotope sampler | Sampling system for measuring stable isotopologues in rain and seawater |
| Rain collector | Palmex Rain collector avoiding re-evaporation used to sample precipitation |
| Radiosondes | Vaisala RS41-SGP radiosondes measuring atmospheric profiles (during ascent and descent) of temperature, relative humidity, pressure, wind direction and wind-speed |
| UAVs | Light ship-based (quad-copter) UAVs were operated (during stations only) for measuring temperature and wind-speed profiles in the lower atmosphere (up to 300 m), and temperature profiles within the upper 20 m of the ocean. |
| CTD casts | Vertical profiles, within the upper 800 m of the ocean, of temperature, salinity, dissolved oxygen and fluorescence |
| Ocean microbiology | Measurements in the upper water column of biogeochemical parameters (e.g. nutrients, chlorophyll A, particulate organic carbon and nitrogen, trace greenhouse gases), biogeochemical process rates (e.g. primary production) and molecular analyses (e.g. microbial community structure) |

*Meteor*

## Ron Brown

Sampling onboard the NOAA R/V Ronald H. Brown took place from January 7 to February 13, 2020 and focused on the region between $57^oW$ and $51^oW$ east of Barbados and north of $12.5^oN$ in Trade Wind Alley. The overarching strategic goal of ATOMIC was to provide a view of the atmospheric and oceanic conditions upwind of the EUREC[4]A study region. Opera-
770 tions of the R/V Ron Brown were coordinated with the Wave Gliders and SWIFTS deployed from the ship, the P-3 aircraft, SailDrone 1064, and BCO (these platforms are described in their respective subsubsections). An additional logistical objective included recovering the NTAS-17 mooring and replacing it with the NTAS-18 mooring. A third objective was to triangulate and download data from a Meridional Overturning Variability Experiment (MOVE) subsurface mooring and related Pressure Inverted Echo Sounders (PIES). MOVE is designed to monitor the integrated deep meridional flow in the tropical North At-
775 lantic. Additional information on shipboard sampling strategies, measurements, and data availability can be found in Quinn et al. (2020).



Table B10: Ron Brown

| Instrument | Brief Description |
|---|---|
| Anemometer | Metek uSonic-3 anemometer measuring wind vector, stress, and heat flux (bow mast) |
| Rain gauge | Optical precipitation sensor ORG-815 DA measuring the rain rate at 1Hz (bow mast) |
| Humidity sensor | Licor-7500 gas analyzer measuring water vapor density (bow mast) |
| Weather sensors | Vaisala HMT335 transmitter measuring air temperature and humidity, and Vaisala PTB220 barometer to measure atmospheric pressure (bow mast) |
| Pyranometer | Eppley PSP measuring shortwave 295–2800 nm irradiance (bow mast) |
| Pyrgeometer | Eppley PIR measuring downward longwave irradiance $4 - 50$ $\mu$m (bow mast) |
| Inertial system | Mast and ship motions (bow mast) |
| Ceilometer | Vaisala CL31 ceilometer measuring vertical profiles of backscatter from refractive index gradients, cloud base height, cloud fraction (O3 deck) |
| Disdrometer | Two Parsivel optical rain gauges (650 and 780 nm) measuring the size, fall speed and precipitation intensity, radar reflectivity, particle number and cumulated precipitation, for rain droplets with sizes between 0.2 and 8 mm (O3 deck) |
| Camera | StarDot NetCam XL camera pointed to starboard (field of view of 50ft , image captured every 4 sec, O3 deck) |
| Doppler lidar | For measuring atmospheric velocities and backscatter to infer cloud base height, and boundary-layer turbulence (O2 deck) |
| Cloud radar | Vertically pointing W-band (95 GHz) Doppler cloud radar measuring vertical profiles of non-precipitating and lightly-precipitating clouds up to 4.2 km with 30 m resolution (O2 deck) |
| Radon detector | Dual-flow loop two-filter radon ($^{222}$Rn) detector (O3 deck) |
| Weather sensors | Vaisala WXT-536 weather sensors measuring air temperature, humidity, pressure, rainfall and wind (O2 deck) |
| Isotope analyzer | Picarro (L2130-i) cavity ring-down laser spectrometer measuring stable isotopologues in water vapor (including water vapor mixing ratio) |
| Isotope sampler | Sampling system for measuring stable isotopologues in rain and seawater |
| Radiosondes | Vaisala RS41-SGP radiosondes measuring atmospheric profiles (during ascent and descent) of temperature, relative humidity, pressure, wind direction and wind speed (every 4 hr, main deck) |
| Ozone analyzer | Thermo Environmental Model 49C measuring ozone concentration (aerosol inlet) |
| In-situ aerosol instrumentation | Collection with multi-jet cascade impactors and analysis by ion chromatography, thermal-optical, gravimetric, and XRF analysis to measure size segregated concentrations of Cl-, NO3-, SO4-, methanesulfonate (MSA-), Na+, NH4+, K+, Mg2+, Ca2+, organic carbon, elemental carbon, trace elements (aerosol inlet) |
| Particle sizer | DMPS and TSI 3321APS measuring number and size distributions 0.02 to 10 $\mu$m (aerosol inlet) |
| " | TSI 3025A, 3760A, 3010 measuring number concentrations $> 3$, 13, 13 nm (aerosol inlet) |



Table B10: Ron Brown, continued

| Instrument | Brief Description |
| --- | --- |
| Nephelometer | TSI 3563 nephelometer measuring aerosol light scattering and backscattering at 60% relative humidity by splitting light into 450, 550, 700 nm wavelengths (aerosol inlet) |
| " | TSI 3563 nephelometer measuring aerosol light scattering and backscattering as a function of relative humidity by splitting light into 450, 550, 700 nm wavelengths at dry and 80% relative humidities (aerosol inlet) |
| Photometer | Radiance Research Particle Soot Absorption Photometer (PSAP) measuring light absorption at 467, 530 and 660 nm (aerosol inlet) |
| Particle counter | DMT cloud condensation nucleus counter (CCNC) measuring the number of particles which can activate cloud droplets at a specific supersaturation (0.1 to 0.6% supersaturation) |
| Sun photometer | MICROTOPS sunphotometer capturing at sun-views the solar attenuation at 380, 440, 670, 870 and 940 nm to derive column properties of aerosol optical depth (AOD) at these wavelengths and the atmospheric water vapor content |
| Spectrometer | Marine-Atmospheric-Emitted Radiance Interferometer (M-AERI) spectrometer to infer skin sea surface temperature (O2 deck) |
| Radiometer | Remote Ocean Surface Radiometer (ROSR) measuring sea surface skin temperature (O2 deck) |
| Thermistor (Sea Snake) | Floating YSI 46040 Thermistor (deployed off port side with outrigger) measuring near-skin sea surface temperature |
| Altimeter | Riegl 1-D laser altimeter measuring wave height (bow mast) |
| CTD casts | Seabird 9+ CTD at station measuring conductivity (salinity), temperature, depth (pressure), PAR, fluorescence, and dissolved oxygen (deployed off starboard side, main deck) |
| Thermosalinograph | Seabird SBE-45 and SBE-38 thermosalinographs measuring seawater temperature and conductivity near the surface and 5 m below the surface |
| CTD/uCTD | ADCP measuring ocean currents to 600 – 750 nm using an RBR Concerto underway CTD and a Tuna Brute winch (uCTD) to measure conductivity (salinity), temperature and depth (pressure) in the upper 60 to 130 m (deployed off starboard aft quarter) |

*Ron Brown*

**Boreal**

The Boreal UAV was operated by Boreal SAS (Toulouse, France) and the National Center for Meterological Research (CNRM; Toulouse, France). During EUREC[4]A it flew 9 missions totalling 32 research flight hours. Its first flight was on 26 January and last on 9 February 2020. The Boreal UAS has a $4.2\,\mathrm{m}$ wingspan and a maximum take-off weight of $25\,\mathrm{kg}$. The Boreal UAV can fly up to 6 hours, covering more than $600\,\mathrm{km}$, with a $5\,\mathrm{kg}$ payload. During EUREC[4]A, the Boreal UAV flew a roughly $20\,\mathrm{km}$



radius circles between 80 m and 1000 m above sea level centered at a distance of roughly 50 km upwind of the Barbados Cloud Observatory (see 2).

Table B11: Boreal

| Instrument | Brief Description |
|---|---|
| Particle Counter | CPC measuring aerosol concentrations ($D > 11\,\text{nm}$) at 1 Hz |
| " | Optical particle counter measuring aerosol size distribution ($0.3\,\mu\text{m} < D < 3\,\mu\text{m}$) at 1 Hz |
| Infrared thermometer | Measuring infrared brightness temperatures to infer SST |
| Turbulence probe | Multi-hole probe measuring vertical and horizontal winds at 30 Hz to characterize turbulence and fluxes |
| Radar Altimeter | To characterize the sea state, including wave height, speed and direction; 1 Hz |
| Weather sensors | Measuring temperature, pressure and relative humidity |
| Solar radiometer | Solar irradiance and broadband solar flux (400 nm to 1100 nm); 1 Hz |
| Camera | High-resolution visible camera looking forward and down to characterize clouds and the sea state |

*Boreal*

## CU-RAAVEN

The RAAVEN was operated from Morgan Lewis Barbados by the University of Colorado Boulder, under support from the National Oceanic and Atmospheric Administration (NOAA). During EUREC[4]A it flew 38 missions spanning 20 days and totaling 77 flight hours. Its first flight was on 24 January and its final flight was on 15 February 2020. It typically flew two flights per day, with one occurring in the mid-morning and one in the mid-afternoon. It is an electric, fixed-wing remotely-piloted aircraft system and operates with an endurance of approximately 2.5 hours. Flights were primarily focused on the 790 boundary layer, with all measurements collected below 1000 m altitude, and majority of observations occurring in the lowest 750 m of the atmosphere. Additional information on the RAAVEN, its capabilities, and the data collected during EUREC[4]A can be found in de Boer et al. (2020).

Table B12: CU-RAAVEN

| Instrument | Brief Description |
|---|---|
| Weather sensors | Vaisala RSS-421 PTH sensors (two body-mounted RSS-421s, similar to sensors in RS41 radiosondes and RD41 dropsondes) measuring pressure, temperature and relative humidity |
| Multihole probe | Black Swift Technologies multihole pressure probe measuring static and dynamic pressure, airspeed, angle of attack, sideslip, attitude and acceleration, temperature, humidity, primarily for wind estimation |
| Inertial system | VectorNav VN-300 INS to measure aircraft position, attitude, acceleration, inertial velocities, static pressure and inertial motion unit, with integrated GPS |





Table B12: CU-RAAVEN, continued

| Instrument | Brief Description |
|---|---|
| Infrared thermometer | Melexis IR thermometers measuring surface and sky brightness temperatures in a narrow field-of-view for sensing surface temperature and cloud presence |
| Turbulence probe | Custom finewire array (three-wire array with two cold wires and one hot wire) for fast temperature, humidity and velocity sensing |

*CU-RAAVEN*

## MPCKs

The Max Planck CloudKites were operated on R/V MS-Merian and R/V Meteor. Two different instrument boxes were flown
with the CloudKites, namely the MPCK+ (only on R/V MS-Merian) and mini-MPCK (on both R/V MS-Merian and R/V Meteor). The CloudKites (expect for the aerostats and winches that are produced by the Allsopp Helikites Ltd) are designed and produced by the Mobile Cloud Observatory at the Max Planck Institute for Dynamics and Self-Organization (http://www.lfpn.ds.mpg.de/MCO/home.html). The MPCK+ and mini-MPCK aboard R/V MS-Merian together flew 18 missions on 17 days (between January 26 and February 18, 2020) totalling 135 measurement flight-hours. Most of the flight times
was spent at 1000 m with the maximum altitude being at 1500 m. The mini-MPCK aboard the R/V Meteor flew 9 successful missions on 9 days (between January 24 and February 6, 2020) totalling 51 measurement flight-hours. Most of the flight time was spent between 200 m and 1000 m.

Table B13: MPCK+

| Instrument | Brief Description |
|---|---|
| Weather sensors | Aosong relative humidity (2% RH accuracy) and temperature (0.1-1 K accuracy) sensor (0.2Hz). |
| Anemometer | 1D static tube with PT100 measuring wind speed and temperature at 100Hz, wind speed range 3-20 m/s with <5% accuracy, temperature -35...55 °C with accuracy of $\pm0.3$ K |
| " | 5-hole static tube; 100Hz, 4...15 m/s, <5% wind-speed accuracy, 1° wind direction accuracy to measure 3D wind velocity |
| Particle image velocimetry | PIV at 15Hz with a pulsed 532 nm laser and a high-speed camera, max. probing. vol. 150 mm x 100 mm x 5 mm, to characterize droplet 2D velocity and spatial distribution |
| Holography | In-line holography at 75Hz with a pulsed 355 nm laser and 25 Mpx camera, max. probing. vol. 15 mm x 15 mm x 250 mm, to characterize 3D droplet size and spatial distribution |
| Cloud droplet probe | Spec Fast-CDP using forward scattering to measure droplet size and number concentration of $2\,\mu$m–$50\,\mu$m particles up to 2000 particles/cm$^3$ |

*MPCK+*





Table B14: Mini-MPCK

| Instrument | Brief Description |
|---|---|
| Weather sensors | Aosong relative humidity (2% RH accuracy) and temperature (0.1-1 K accuracy) sensor (0.2Hz). |
| Anemometer | 1D static tube with PT100 measuring wind speed and temperature at 100Hz, wind speed range 3-20 m/s with <5% accuracy, temperature -35...55 °C with accuracy of ±0.3 K |
| ” | Metek uSonic-3 Class A ultrasonic anemometers operating at 30Hz, 0-40 m/s with minimal flow shadowing from of the arms, 1° wind direction accuracy |
| Cloud droplet probe | DMT CDP-2 UAV, a forward-scattering optical spectrometer to determine size and number concentration of $2\,\mu m$–$50\,\mu m$ particles |
| Weather sensors | Vaisala HMP7 heated humidity and temperature probe; response time 15 s, ±0.8 % RH accuracy, ±0.1 K temperature accuracy |
| ” | Bosch BMP388 sensor measuring pressure and temperature (±50 Pa absolute pressure accuracy, ±0.3 K temperature accuracy) |
| Hot wire | Dantec mini-hotwire measuring air velocity fluctuations at 8 kHz |
| Ultra-fast thermometer | Ultra-Fast Thermometer M (UFT-M, provided by University of Warsaw) measuring temperature fluctuations with a sampling frequency of 8 kHz |

*Mini-MPCK*

## Skywalker

The Skywalker X6 UAVs were operated by the National Center for Meteorological Research (CNRM; Toulouse, France), the
French Civil Aviation University (ENAC; Toulouse, France), and the Laboratory for Analysis and Architecture of Systems
(LAAS; Toulouse, France) as part of the NEPHAELAE project (ANR-17-CE01-0003). During EUREC[4]A, the team flew more
than 50 flights over the ocean up to 15 km off the eastern coast of Barbados. The first flight was on 25 January and last on 9
February 2020. The Skywalker UAV has a 1.5 m wingspan and a maximum take-off weight of 2.5 kg. The Skywalker UAVs
flew up to one hour and utilized sensor feedback to map an individual cloud autonomously and adaptively.

Table B15: Skywalker

| Instrument | Brief Description |
|---|---|
| Cloud sensor | Cloud extinction measured at three wavelengths (1 Hz) |
| Cloud active mapping system | Adaptive sampling and real-time mapping of clouds |
| Weather sensors | Measurements of temperature and relative humidity |
| Camera | High-resolution visible camera looking forward and down to characterize clouds and the sea state |
| Inertial system | Paparazzi Apogee autopilot providing UAV flight parameters (latitude, longitude, altitude, pitch, roll, yaw, horizontal wind speed) |



Table B15: Skywalker, continued

| Instrument | Brief Description |
| --- | --- |
| *Skywalker* | |

## Caravela

The Autonaut (Caravela) was operated by the University of East Anglia (UEA). Caravela is an unmanned surface vehicle with continuously sampling meteorological and oceanographic sensors, and can also be used to tow and deploy a Seaglider. During EUREC[4]A she was deployed on 22nd January from the coastguard station on Barbados towing Seaglider SG579, which was released on 28th January. Caravela continued to her study location centered at $14.2°$ N, $57.3°$ W, and spent 11 days repeating a butterfly pattern around a square with 10 km sides, before returning to Barbados for recovery on 24th February.

Table B16: Caravela

| Instrument | Brief Description |
| --- | --- |
| Pyranometer | CS301 Apogee SP-110-SS Pyranometer measuring downwelling shortwave radiation (360 - 1120 nm) |
| Pyrgeometer | Apogee SL-510-Pyrgeometer measuring downwelling longwave radiation (5 - 30 $\mu$m) |
| Weather sensors | Airmar 120WX weather station measuring wind velocity, air temperature (measurement height 1.5m) |
| " | Rotronic HC2A-S3, Rotronic MP402H 082000, Rotronic AC1003 measuring air temperature and humidity (measurement height 1m) |
| Ocean state | Valeport MiniCTD measuring sea surface temperature and conductivity (mounted through Caravela's hull, measurement depth 0.2m) |
| Acoustic Doppler current profiler | Nortek Signature1000 1 MHz Acoustic Doppler Current Profiler (ADCP) with a 5 beam set up for estimating velocity shear and biomass in the water column, range 30 m. |
| *Caravela* | |

**Underwater electric gliders**

Three Seagliders were operated by UEA. SG579 was deployed from Caravela on 28th January and completed 76 dives to 1000 m along the transect to the gliders' study location, followed by 219 dives to 250 m while traversing the same butterfly pattern as Caravela. She was recovered onto R/V Meteor on 16th February. SG637 was deployed from R/V Meteor on 23rd January, completed 155 dives to 750 m around the same 'butterfly', all with ADCP data, and was also recovered onto R/V Meteor on 16th February. SG620 was deployed from the Meteor on 23rd January and completed 131 dives to 750 m in virtual mooring mode at the centre of the butterfly, before being recovered on 5th February onto R/V Meteor.





Three Teledyne Slocum G2 underwater electric gliders were operated by GEOMAR. All three surveyed down to 1000 m depth. IFM03 and IFM12 were deployed from the R/V MS-Merian on January 24 to survey the edge of a mesoscale eddy. IFM03 was recovered by the R/V Meteor on February 3rd, and IFM12 was recovered with R/V MS-Merian on February 17, 2020 after a complete survey. IFM09 was operated within the EUREC[4]A circle as a virtual mooring mission from January 20 and recovered by R/V MS-Merian on February 9, 2020.

One SeaExplorer X2 underwater glider (Kraken) was deployed and recovered from the R/V Atalante and operated by DT INSU CNRS in connection with the ship chief scientist. Kraken was deployed on January 25 at $10°8'45''$N $57°29'37''$W and was recovered on February 13, 2020 at $10°19'24''$N $57°53'30''$W. It accomplished 472 profiles down to 700 m across two different mesoscale eddies, both anticyclonic, a North Brazil Current ring limited to the upper 150 m of depth, and a thicker subsurface (200–600 m) eddy.

Table B17: Humpback (SG579)

| Instrument | Brief Description |
| --- | --- |
| CT sail | Seabird unpumped CT sail measuring temperature and conductivity |
| PAR sensor | Biospherical Instruments QSP2150 measuring photosynthetically active radiation |
| Fluorometer | Wetlabs Triplet Ecopuck measuring chlorophyll fluorescence, chromophoric dissolved organic matter (CDOM) fluorescence and optical backscatter at 650 nm |

*Humpback*

Table B18: Omura (SG637)

| Instrument | Brief Description |
| --- | --- |
| CT sail | Seabird unpumped CT sail measuring temperature and conductivity |
| Acoustic Doppler current profiler | Nortek Signature1000 1 MHz Acoustic Doppler Current Profiler (ADCP) measuring vertical shear of horizontal current velocity |

*Omura*

Table B19: Melonhead (SG620)

| Instrument | Brief Description |
| --- | --- |
| CT sail | Seabird unpumped CT sail measuring temperature and conductivity |
| Microstructure sonde | Rockland Scientific Micropods microstructure system measuring shear and temperature and thus ocean turbulence |



Table B19: Melonhead (SG620), continued

| Instrument | Brief Description |
| --- | --- |

*Melonhead*

Table B20: IFM03

| Instrument | Brief Description |
| --- | --- |
| CTD sensor | Measuring temperature and salinity |
| Oxygen sensor | Optode measuring dissolved oxygen |
| Fluorometer | Wetlabs Ecopuck measuring chlorophyll fluorescence and turbidity |
| Microstructure sonde | Rockland Scientific MicroRider turbulence sensor |

*IFM03*

Table B21: IFM09

| Instrument | Brief Description |
| --- | --- |
| CTD sensor | Measuring temperature and salinity |
| Oxygen sensor | Optode measuring dissolved oxygen |
| Fluorometer | Wetlabs Ecopuck measuring chlorophyll fluorescence and turbidity |
| Microstructure sonde | Rockland Scientific MicroRider turbulence sensor |

*IFM09*

Table B22: IFM12

| Instrument | Brief Description |
| --- | --- |
| CTD sensor | Measuring temperature and salinity |
| Oxygen sensor | Optode measuring dissolved oxygen |
| Fluorometer | Wetlabs Ecopuck measuring chlorophyll fluorescence and turbidity |
| Nutrient analyzer | Seabird Submersible UV Nitrate Analyzer (SUNA) |
| Optical sensor | Measuring dissolved organic matter (CDOM) |

*IFM12*

Table B23: Kraken

| Instrument | Brief Description |
| --- | --- |
| CTD sensor | Seabird CTD (GPCTD) measuring temperature, salinity and pressure |
| Oxygen sensor | Optode measuring dissolved oxygen |
| Fluorometer and backscatter sensor | Wetlabs Eco Puck BB2FLVMT measuring chlorophyll-A fluorescence and turbidity |
| Optical sensor | Wetlabs Eco Puck measuring dissolved organic matter (CDOM) |

*Kraken*

## Saildrones

Continuous measurements of air-sea interaction by five Uncrewed Surface Vehicle (USV) Saildrones were led by Farallon
Institute, NOAA/PMEL and CICOES/University of Washington during EUREC[4]A and ATOMIC from January 12 to March 3,
2020. Three NASA funded Saildrones (SD1026, SD1060, SD1061) and one NOAA funded Saildrone (SD1063) were dedicated
to the ocean eddy corridor southeast of Barbados, where large North Brazil Current Rings migrating northwestward. The
NOAA funded Saildrone SD1064 was dedicated to the Trade Wind Alley between NTAS buoy and the HALO flight circle.
The two NOAA funded Saildrones continued their observation after the EUREC[4]A/ATOMIC intensive observation period until
July 15, 2020.

Table B24: Saildrones

| Instrument | Brief Description |
| --- | --- |
| Anemometer | WindMaster Gill 1590-PK-020 sonic anemometer 10-Hz at $5.2\,\text{m}$ above water line, measuring wind speed and direction |
| Weather sensors | Rotronic Hygroclip 2 measuring air temperature and relative humidity at 1 Hz, 1-minute measurements every 5-minute at $2.3\,\text{m}$ above water line |
| Pyrometer | Heitronics CT 15.10 infrared pyrometer to measure skin SST (experimental) |
| Cameras | Four visible cameras looking sideways, upward and downward |
| PAR sensor | LICOR LI-192SA sensor measuring photosynthetically active radiation at $2.6\,\text{m}$ above water line |
| CT sensor | SBE-37-SMP-ODO Microcat measuring seawater temperature, salinity, conductivity and dissolved oxygen (pumped CTD at $0.5\,\text{m}$ depth) |
| " | RBR C-T-ODO.chl-a measuring seawater temperature, salinity, conductivity, dissolved oxygen, Chlorophyll-a (experimental); inductive CTD at $0.53\,\text{m}$ depth |
| Fluorometer | Wetlabs ECO-FL-S G4 and Turner Cyclops measuring Chlorophyll-a (experimental) at $0.5\,\text{m}$ depth |
| Barometer | Vaisala PTB 210 A1A1B measuring barometric pressure at $0.2\,\text{m}$ above water line |





Table B24: Saildrones, continued

| Instrument | Brief Description |
| --- | --- |
| Inertial system | VectorNav VN300 Dual GPS-aided inertial measurement unit deriving estimates of significant wave height and dominant period from wave spectrum |
| Thermistors | Seabird SBE-57s thermistors of seawater temperature (on keel at 0.3 m, 0.6 m, 0.9 m, 1.2 m, 1.4 m and 1.7 m depths) |
| Acoustic Doppler current profiler | Teledyne Workhorse 300kHz ADCP measuring ocean current profiles between 6 m and 100 m depths |
| Pyranometer | SPN1 Delta-T Sunshine Pyranometer measuring total and diffuse solar radiation (equipped on NOAA funded Saildrones, 2.8 m above water line, sampling at 5Hz for tilt correction, Zhang et al. (2019)) |
| Pyrgeometer | Eppley Precision Infrared Radiometer (PIR) measuring downward longwave radiation (equipped on NOAA funded Saildrones, 0.8 m above water line) |

*Saildrones*

## SWIFTs and wave gliders

A total of six SWIFT (Surface Wave Instrument Floats with Tracking) drifting platforms were deployed from the R/V Ron Brown during EUREC[4]A. There was an initial deployment for two weeks in January 2020 and a second deployment for another two weeks in February 2020. SWIFTs are produced and operated by the Applied Physics Laboratory at the University of Washington. Two of the SWIFTs were version 3 models, as described in Thomson (2012), and four of the SWIFTs were version 4 models, as described in Thomson et al. (2019). SWIFTs collect data in a wave following reference frame, with burst sampling and statistical products available hourly.

Table B25: SWIFTs

| Instrument | Brief Description |
| --- | --- |
| Inertial system | SBG Ellipse GPS measuring measuring drift speed from 5 Hz positions and velocities |
| ” | SBG Ellipse IMU measuring surface waves from 5 Hz displacements (processed for hourly wave spectral parameters) |
| Weather sensors | Vaisala WXT-536 measuring air temperature, humidity, pressure, rainfall and wind at 1 m |
| Weather sensors | Airmar PB200 measuring wind speed and direction, air tempature and air pressure at 1 m above surface |
| CT sensors | Aanderaa CT sensor measuring surface conductivity and temperature to infer ocean salinity and temperature at 0.5 m below surface |





Table B25: SWIFTs, continued

| Instrument | Brief Description |
|---|---|
| Acoustic Doppler current profiler | Nortek Aquadopp 2 MHz ADCP used in pulse-coherent (HR) mode to measure ocean turbulence and current profiles with high resolution |
| Acoustic Doppler current profiler | Nortek Signature 1 MHz ADCP used in pulse-coherent (HR) mode to measure ocean turbulence and current profiles with high resolution |
| Camera | Low-resolution sky cameras providing pictures every 4 sec |

*SWIFTs*

Two Wave Glider ASVs (Autonomous Surface Vehicles) operated from the R/V Ron Brown during EUREC[4]A. The missions each spanned four weeks from January to February 2020. The Wave Gliders are built by Liquid Robotics, Inc, with additional
850 sensors integrated by the Applied Physics Laboratory at the University of Washington for air-sea flux calculations (Thomson and Girton, 2017).

Table B26: Wave Gliders

| Instrument | Brief Description |
|---|---|
| Inertial system | Microstrain GPS measuring vehicle speed from 5 Hz positions and velocities |
| ” | Microstrain IMU measuring surface waves from 5 Hz displacements (processed for hourly wave spectral parameters Thomson et al. (2018)) |
| Weather sensors | Vaisala WXT-536 measuring air temperature, humidity, pressure, rainfall and wind at 1 m |
| ” | Airmar PB200 measuring wind speed and direction, air tempature and air pressure at 1 m above surface |
| CT sensors | Aanderaa CT sensor measuring surface conductivity and temperature to infer ocean salinity and temperature at 0.3 m below surface |
| Glider payload | Seabird GPCTD measuring conductivity, temperature and depth to infer ocean salinity and temperature at 8 m below surface |
| Acoustic Doppler current profiler | 300 kHz ADCP measuring ocean current profiles from 0 to 120 m depth |

*Wave Gliders*

**Surface drifters, Ocarina and Piccolo**

Ten SVP-BSW (Barometer-Salinity-Wind) drifters from Pacific Gyre were deployed from R/V Atalante from January 23 to February 4, 2020. Each drifter had temperature and salinity measurements at 3 levels (0.50 m, 5 m, and 10 m) as well as
barometric sea level pressure and wind.





Five SC40 SVP-BRST drifters (from Eumetsat grant TRUSTED to Meteo-France/CLS) manufactured by NKE were deployed for three short periods of time from the R/V Atalante before being finally released at sea between February 10 and 14, 2020. Each drifter provided high-precision measurements of temperature at 20 cm depth and of barometric sea level pressure.

Two SC40 SVP-BSC drifters (from Meteo-France/LOCEAN with CNES/SMOS support) manufactured by NKE were deployed for three short periods of time from the R/V Atalante before being released at sea between February 10 and 14, 2020. Each drifter measured temperature and salinity at 20 cm depth and barometric sea level pressure.

Two SURPACT wave riders manufactured by SMRU (University of St-Andrew) and LOCEAN (with CNES/SMOS support) were deployed for short times from the R/V Atalante before being released at sea between February 10 and 14, 2020. These drifters do not have a drogue to follow the currents at 15 m, and thus are deployed tethered to another drifter. Each float measured temperature and salinity at 5 cm depth, as well as vertical acceleration and thus wave spectra. Moreover, the noise recorded by a microphone under a cupola was spectrally analyzed to estimate the noise of the wind/waves and rainfall.

Five barometer-temperature drifters from Pacific Gyre were deployed from R/V MS-Merian from January 21 to February 4, 2020. Each drifter measured temperature at 20 cm depth and barometric sea level pressure. Each drifter had a 6 m long drogue centered at a depth of 15 m. The drifters used Iridium SBD Telemetry and 50-channel GPS system (including WAAS correction), and data were transmitted every 30 minutes.

Table B27: SVP-BSW drifters

| Instrument | Brief Description |
| --- | --- |
| Thermistor | 30K thermistor measuring seawater temperature at a depth of 20 cm |
| CT sensors | Seabird SBE-37-SI measuring seawater temperature and conductivity (salinity) at a depth of 20 cm |
| ” | Seabird SBE-37-IM measuring seawater conductivity and temperature to infer ocean temperature and salinity at depths of 5 and 10 m |
| Pressure sensor | Honeywell Integrated Pressure Transducer (IPT) measuring atmospheric pressure at a height of 50 cm |
| Tilt compensated compass | PNI TCM-2.5 measuring wind direction at a height of 50 cm |
| Anemometer | Gill WindSonic 2-axis ultrasonic anemometer measuring wind speed at a height of 50 cm |
| GPS | To monitor the current at sea surface |

*SVP-BSW drifters*

Table B28: SVP-BRST drifters

| Instrument | Brief Description |
| --- | --- |
| HRSST thermistor | Digital high-resolution thermistor measuring seawater temperature at a depth of 15 cm |
| Pressure sensor | Vaisala PTB110 barometer measuring atmospheric pressure at a height of 50 cm |





Table B28: SVP-BRST drifters, continued

| Instrument | Brief Description |
| --- | --- |
| GPS | To monitor the current at sea surface |

*SVP-BRST drifters*

Table B29: SVP-BSC drifters

| Instrument | Brief Description |
| --- | --- |
| CT sensors | Seabird SBE-37-SI measuring seawater temperature and conductivity (salinity) at a depth of 15 cm |
| Pressure sensor | Vaisala PTB110 barometer measuring atmospheric pressure at a height of 50 cm |
| GPS | To monitor the current at sea surface |

*SVP-BSC drifters*

Table B30: SURPACT drifters

| Instrument | Brief Description |
| --- | --- |
| CT sensors | Valeport CTD probe measuring seawater temperature and conductivity (salinity) at depth of 4 cm |
| Pressure sensor | Honeywell Integrated Pressure Transducer (IPT) measuring atmospheric pressure |
| Audio analyzer | MSGEQ7 Graphic Equalizer rain sensor to infer rain intensity from sound spectra |

*SURPACT drifters*

Table B31: Pacific Gyre Barometer-temperature drifters

| Instrument | Brief Description |
| --- | --- |
| Thermistor | 30K thermistor measuring seawater temperature at a depth of 20 cm |
| Pressure sensor | Honeywell Integrated Pressure Transducer (IPT) measuring atmospheric pressure at a height of 50 cm |
| GPS | monitors the current at sea surface |

*Barometer-temperature drifters*

The R/V Atalante also deployed two drifting platform prototypes, Ocarina (Ocean Coupled to Atmosphere, Research at the Interface with a Novel Autonomous platform) and Piccolo (Profiling Instrument to Check if the wind Curvature is Only





Logarithmic on the Ocean), to measure air-sea fluxes (momentum, sensible heat, latent heat, radiative shortwave and longwave up and down) very close to the sea surface. The platforms were deployed on Januuary 25, 2020 from 15:00 UTC to 21:00

UTC (Start position 57.50°W 10.11°N, End position 57.45°W 10.15°N), on February 3rd from 10:00 UTC to 21:00 UTC (Start position 54.14°W 6.83°N, End position 54.34°W 6.83°N), on February 10 from 10:00 UTC to 19:00 UTC (Start position 59.13°W 10.38°N, End position 59.23°W 10.53°N), and on February 17 from 09:00 UTC to 18:00 UTC (Start position 59.75°W 13.13°N, End position 59.82°W 13.10°N).

Table B32: Ocarina

| Instrument | Brief Description |
|---|---|
| Anemometer | Gill R3-50 3-axis sonic anemometer measuring wind speed, horizontal and vertical velocities at 50 Hz at a height of 1.6 m |
| Weather sensors | Vaisala WXT-520 measuring air temperature, barometric pressure, relative humidity at 1 Hz at a height of 1 m |
| Pyranometer | Campbell CNR4 Kipp and Zonen radiometer measuring broadband downward solar irradiance at a height of 1 m |
| Pyrgeometer | Campbell CNR4 Kipp and Zonen radiometer measuring upward and downward broadband infrared irradiance at a height of 1 m |
| Inertial platform | X-sens MTI-G IMU measuring time and accelerations at 50 Hz through gyroscopes, accelerometers, magnetometers, a GPS and a barometer at a height of 0.1 m |
| CT sensor | Seabird SBE-37 SI measuring sea surface temperature and conductivity/salinity at 1 Hz at a height of −0.2 m |
| Acoustic Doppler current profiler | Nortek Signature 1.2 GHz ADCP measuring vertical profiles of ocean current, horizontal and vertical velocities at 25 Hz from −0.3 m to −17.3 m every 0.5 m of water. |

*Ocarina*

Table B33: Piccolo

| Instrument | Brief Description |
|---|---|
| Anemometers | 5 home-made cup anemometer measuring the horizontal wind speed close to the surface (at 0.40 m, 0.60 m, 0.85 m, 1.15 m and 1.6 m) to infer the friction velocity $u_*$, the wave peak period and the roughness aerodynamic length $z_0$ |
| Inertial platform | X-sens MTI-G IMU measuring time and accelerations at 50 Hz through gyroscopes, accelerometers, magnetometers, a GPS and a barometer at a height of 0.1 m |

*Piccolo*





*Author contributions.* Author contributions are grouped (alphabetically) in different categories of contribution. BS and SB conceived of and led EUREC[4]A and the preparation and writing of this manuscript. DF co-led the coordination as the local lead-PI. FA, AB, CF, JK, PKQ and SS conceived of and led a major component of EUREC[4]A. AB, DF and SS additionally contributed substantially to the preparation of the manuscript. The next grouping of authors, beginning with C. Acquistapace, made major contributions to the broader coordination and execution of scientific and/or dissemination activities. The final group of authors, beginning with N. Alexander, served as instrument PIs and/or made substantial contribution to the scientific preparation, execution, and/or dissemination activities. The groupings are subjective and imperfect, but were motivated in part by the desire to better highlight the initiative of early career scientists in taking on responsibility and contributing to both the intellectual and organizational foundations of EUREC[4]A. Specific contributions, as self statements, are recorded as part of the scientific documentation.

*Competing interests.* The authors declare no competing interests

*Acknowledgements.* The authors thank the people and government of Barbados for their support of EUREC[4]A. Without the hospitality and scientific interest of both Barbadian citizens and their institutions (governmental, private and non-profit), our efforts would not have borne fruit. The Max Planck Society supported the preparation of the EUREC[4]A campaign, and BCO and HALO operations, measurements on the R/V Meteor and R/V MS-Merian, including cloud-kite operations and the addition of an upper air sounding network. Supporting members of the Max Planck Society provided funding for the POLDIRAD deployment and outreach activities. The preparation of the EUREC[4]A campaign, the ATR and Boreal operations, and support for dropsondes and outreach activities, were funded by the European Research Council (ERC) Advanced Grant EUREC[4]A (grant agreement No 694768) under the European Union's Horizon 2020 research and innovation programme (H2020), with additional support from CNES (the French space agency) through the EECLAT proposal, Meteo-France, the CONSTRAIN H2020 project (grant agreement No 820829), and the French AERIS Research Infrastructure. Additional support for HALO was provided by the DFG Priority Programme 1294. The cruises were funded by the German Research Foundation (DFG) and the German Federal Ministry of Education and Research (grant numbers: GPF18-1_69 and GPF18-2_50). The UK work associated with the Twin Otter was supported by the Natural Environment Research Council (NE/S015868/1, NE/S015752/1and NE/S015779/1). The deployment of the AutoNaut Caravela and the three UEA seagliders was supported by funding from the ERC under the European Union's H2020 programme (COMPASS, Advanced Grant agreement No. 74110). Funding for the ocean-atmosphere EUREC[4]A-OA effort has been provided by the French national program LEFE INSU, by IFREMER, the French Research Fleet, CNES, the French Research Infrastructures AERIS and ODATIS, IPSL, the Chaire Chanel of the Geosciences Department at ENS and the European Union's Horizon 2020 research and innovation program under grant agreement No. 817578 TRIATLAS. Measurements onboard the R/V Ron Brown and two saildrones were supported by NOAA's Climate Variability and Prediction Program within the Climate Program Office (Grants GC19-305 and GC19-301). This is PMEL contribution number 5173. The Joint Institute for the Study of the Atmosphere and Ocean (JISAO) supported this study under NOAA Cooperative Agreement NA15OAR4320063. The barometer-salinity-wind drifters were supported by NOAA's Climate Program Office and base funds to NOAA/AOML's Physical Oceanography Division. Support was also provided by the Swiss National Science Foundation Grant No. 188731. CU-RAAVEN observations were supported by NOAA through funding from the UAS Program Office, Climate Program Office and Physical Sciences Laboratory and by the US National Science Foundation (NSF) through grant AGS-1938108. Additional support for HALO flight operations was provided by the Deutsche Forschungsgemeinschaft (DFG, German Research Foundation) under Germany's





Excellence Strategy – EXC 2037 'CLICCS - Climate, Climatic Change, and Society' – Project Number: 390683824, contributing to the Center for Earth System Research and Sustainability (CEN) of Universität Hamburg. Universität Hamburg. We thank the captains and crews
of the R/V Atalante, the R/V Meteor (cruise M161), the R/V MS-Merian (cruise MSM89) and the R/V Ron Brown for their excellent support. Deployment of the Ultra Fast Thermometers and quad-copter drones was possible due to Poland's National Science Centre Grant No. UMO-2018/30/M/ST10/00674 and Foundation for Polish Science Grant No. POIR.04.04.00-00-3FD6/17-02. We likewise thank the pilots, technicians and engineers of the ATR-42, HALO, the Twin Otter, and the WP-3D. Mr Richard Hoad is thanked for his support and access to his property for UAV operations. Ms Alissandra Cummins and the Museum of Barbados are thanked for their enduring support and
their cooperation in the establishment and maintenance of the BCO. Mr William Burton is thanked for a guided hiking tour for the on-island scientists. We also gratefully acknowledge Aurélien Bourdon (director of SAFIRE) as well as all the SAFIRE and Airplane Delivery staff, for overcoming extraordinary last minute challenges to make the participation of the ATR in EUREC[4]A possible. Gaelle Bruant is thanked for her administrative support, likewise, and also for an admirable ability to manage people and bureaucracies, Angela Gruber.



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
