# Peer review of "EUREC4A"

_Earth System Science Data, 2021_

## Author Response (AR1)

**Author response to editor decision, and reviews, of EUREC⁴A (manuscript: essd-2021-18)**

Bjorn Stevens, Sandrine Bony, David Farrell, et al.,
14 May, 2021

We thank the editor and the reviewers for the attention they devoted to our manuscript. EUREC⁴A was a special experience for many of us, and we are glad that the emotional attachment that we had for this campaign was apparent in the manuscript devoted to its description.

The reviewer comments raised a few issues that motivated some substantial changes as outlined below:

1. We redrafted most figures and figure captions: Figures were changed to improve the consistency across figures, to better conform to standards (e.g., for labels), and to address mistakes or lack of clarity as identified in the reviewer comments. Figure captions were modified to better describe the figures, particularly more complex figures as suggested by Reviewer 2.
2. We analyzed a possible island influence on the BCO measurements, and were able to identified a heightened diel cycle (1.27 K) in both the near-surface insitu, and 400 m lidar-retrieved, temperatures as compared to offshore measurements (0.53 K). This quantification, along with a few words of speculation as to its possible origin, is included in the revised manuscript.
3. We analyzed the 3hr changes in CCN measurements at Ragged Point to evaluate how steady the CCN might be expected to remain over a 3hr period of circling. Despite the large variations across the campaign the changes over a three hour period are generally small, half the time less than 10% and only 20% of the time does the CCN increase or decrease by more than 30%. This analysis is now presented in the manuscript.
4. The standardizing of the track data (which was used for many of the figures) and its publishing as a data asset (with doi) with this paper. This was not requested by the reviewers but was suggested at the time of initial submission by the chief editor, and given that we have found these data to be useful we thought to standardize, quality control and include them with the manuscript.
5. Inclusion of the EUREC⁴A Film as a video supplement.

In addition we addressed the concern raised by the comments in the first review regarding data policy and the adequacy of the measurements to measure diel cycles. All the minor and editorial comments were addressed as requested. Small editorial changes were also introduced based on our own proofing of the manuscript, none of these changes address issues of substance, or change the basic content of the manuscript.